# Automated Classification of Model Errors on ImageNet

**Momchil Peychev**,* **Mark Niklas Müller**,* **Marc Fischer, Martin Vechev**
Department of Computer Science
ETH Zurich, Switzerland
{momchil.peychev, mark.mueller, marc.fischer, martin.vechev}@inf.ethz.ch

## Abstract

While the ImageNet dataset has been driving computer vision research over the past decade, significant label noise and ambiguity have made top-1 accuracy an insufficient measure of further progress. To address this, new label-sets and evaluation protocols have been proposed for ImageNet showing that state-of-the-art models already achieve over $95\%$ accuracy and shifting the focus on investigating why the remaining errors persist. Recent work in this direction employed a panel of experts to manually categorize all remaining classification errors for two selected models. However, this process is time-consuming, prone to inconsistencies, and requires trained experts, making it unsuitable for regular model evaluation thus limiting its utility. To overcome these limitations, we propose the first automated error classification framework, a valuable tool to study how modeling choices affect error distributions. We use our framework to comprehensively evaluate the error distribution of over 900 models. Perhaps surprisingly, we find that across model architectures, scales, and pre-training corpora, top-1 accuracy is a strong predictor for the *portion* of all error types. In particular, we observe that the portion of severe errors drops significantly with top-1 accuracy indicating that, while it underreports a model's true performance, it remains a valuable performance metric. We release all our code at `https://github.com/eth-sri/automated-error-analysis`.

## 1 Introduction

IMAGENET (Deng et al., 2009; Russakovsky et al., 2015) has established itself as one of the most influential and widely used datasets in computer vision, driving progress in object recognition (Krizhevsky et al., 2012), object detection (Tan et al., 2020), and image segmentation (Minaee et al., 2022). As state-of-the-art models have come close to, and by some metrics exceeded, human performance, the significance of further progress in top-1 and top-5 accuracy has, been questioned in the face of label errors and systematic biases (Beyer et al., 2020; Tsipras et al., 2020).

The most severe such bias is the lack of multi-label annotations in the original IMAGENET dataset, with recent studies finding that roughly a fifth of images show multiple entities. While state-of-the-art models have learned to exploit labeling biasses on these images (Tsipras et al., 2020), best practices have shifted to reporting multi-label accuracy (MLA) computed using new multi-label annotations (Beyer et al., 2020; Shankar et al., 2020). Further, many IMAGENET and especially organism classes, are hard to distinguish even by trained humans (Horn et al., 2015; Northcutt et al., 2021a; Lee et al., 2017), leading to persistent labeling errors.

In the face of these challenges and with state-of-the-art models exceeding $95\%$ MLA, the focus has increasingly shifted towards analyzing and understanding the remaining model errors instead of blindly pursuing improvements in headline accuracy numbers. To this end, Vasudevan et al. (2022) review all remaining errors of two state-of-the-art models using a panel of experts and classify them

---

*Equal contribution

37th Conference on Neural Information Processing Systems (NeurIPS 2023).

with regards to both error category and severity. While they find that many of the remaining errors are minor or can be attributed to fine-grained class distinctions, they also find major classification errors that are not so easily explainable. We believe that tracking and analyzing the distribution of these error types over a large number of models can not only help us to better understand the impact of novel training techniques and architectures but also identify where the biggest challenges lie and thus how to address them. However, the manual review process employed by Vasudevan et al. (2022) has several issues, preventing it from being employed for a large scale or repeated study of model errors: (i) it is time-consuming even for precise models, making it infeasible to repeat for a large number of potentially less precise models, (ii) it requires (a panel of) expert reviewers which need to be trained on the fine-grained class distinctions, and (iii) it is inconsistent as different reviewers or even the same reviewers at different times might classify the same error differently.

**This Work**  To overcome these challenges, we propose an automated error classification pipeline that we use to study the distribution of different types of errors across 962 models of different scales, architectures, training methods, and pre-training datasets. Our pipeline allows us to automatically detect all four error categories identified by Vasudevan et al. (2022): (i) fine-grained classification errors are detected using a set of 161 manually defined superclasses, (ii) fine-grained out-of-vocabulary errors are detected using a visual similarity based criterion for prediction quality and confirmation of their out-of-vocabulary nature using an open-world classifier, (iii) non-prototypical examples are identified using the exhaustive annotations by Vasudevan et al. (2022), and (iv) spurious correlations are detected using a co-occurrence-frequency based criterion.

**Main Findings**  Our automated error classification pipeline allows us to, for the first time, study the *distribution of different error types* across a large number of models, leading to the following insights: (i) even MLA is a pessimistic measure of model progress with the *portion* of severe model failures quickly decreasing with MLA, (ii) this reduction of model failure rate with MLA is more pronounced for larger (pre-)training corpora, *i.e.*, models trained on more data make less severe errors even at the same top-1 or multilabel accuracy, and (iii) organism and artifact classes exhibit very different trends and prevalences of error types, *e.g.*, fine-grained classification errors are much more frequent for organisms than for artifacts, while artifacts suffer much more from spurious correlations and out-of-vocabulary errors. We believe that these insights can help guide future research in computer vision and that studying the effects of new methods on the resulting error distribution can become an important part of the evaluation pipeline.

## 2   Related Work

**Multi-Label Annotations**  While the IMAGENET dataset is annotated with a single label per image, many images have been found to contain multiple entities (Beyer et al., 2020; Shankar et al., 2020; Tsipras et al., 2020; Yun et al., 2021). Thus, multi-label accuracy (MLA) with respect to new multi-label annotations has been established as a more meaningful metric. Yun et al. (2021) generate pixel-wise labels for IMAGENET by directly applying a classification layer to image embeddings before spatial pooling. Beyer et al. (2020) collect Reassessed Labels (ReaL) for the whole IMAGENET validation set, by first identifying 6 models with high prediction coverage and accuracy, before manually annotating all images where these 6 models disagree using non-expert labelers. They discard 3 163 images where these labelers disagreed. Tsipras et al. (2020) collect multi-label annotations for 10 000 validation set images, and find that top-1 accuracy is 10% lower for multi- compared to single-label images while MLA accuracy is identical. Shankar et al. (2020) collect multi-label annotation for 40 000 IMAGENET and IMAGENETV2 validation images by manually reviewing all predictions made by a diverse set of 72 models, using human expert labelers.

**Label Errors**  Northcutt et al. (2021a) study label errors across 10 commonly used datasets, including IMAGENET, using a confident learning framework to identify potential errors before validating them with Mechanical Turk (MTurk). Studying the whole validation set, they report an error rate of over $5.8\%$. Lee et al. (2017) manually review 400 randomly selected classification errors of an ensemble model and find the IMAGENET label for a substantial portion to either be incorrect or not describe the main entity in the image. Vasudevan et al. (2022) reviewed all remaining errors for two state-of-the-art models with a panel of experts and found that of 676 reviewed model mistakes, 298 were either correct, ambiguous, or the original ground truth incorrect or problematic.

**Error Analysis**   Recent work has focused on understanding the types of errors models still make on IMAGENET. To this end, one strand of work analyses the differences in errors between models. Mania et al. (2019) find that independently trained models make errors that correlate well beyond what can be expected from their accuracy alone. Geirhos et al. (2020) analyze the consistency between errors made by humans and CNNs on a 16 class version of IMAGENET. They observe that while CNNs make remarkably similar errors, the consistency between humans and CNNs barely goes beyond chance. Nguyen et al. (2021) analyze wide and deep ResNets and find that these exhibit different error patterns. Mania and Sra (2020) find that more precise models typically dominate less precise ones, *i.e.*, their error set is a subset of that of less precise models. Lopes et al. (2022), in contrast, find that different pretraining corpora and training objectives can significantly increase error diversity with Andreassen et al. (2021) showing that this diversity reduces significantly during finetuning.

Vasudevan et al. (2022) focus on the errors current state-of-the-art models make, identifying four categories: (i) *fine-grained* errors describe a model's failure to distinguish between two very similar classes, (ii) *fine-grained out-of-vocablary* errors occur when an image shows an entity not contained in the IMAGENET vocabulary and the model instead predicts a similar class, (iii) *spurious correlations* cause the model to predict a class that is not shown in the image in the presence of strongly correlated features, and (iv) *non-prototypical* instantiations of a class are not recognized by the model.

**Datasets for Error Analysis**   In addition to work analyzing model errors on fixed datasets, there has been a growing interest in datasets specifically designed to highlight and thus analyze specific error types. Singla and Feizi (2022) and Moayeri et al. (2022) find that, for some classes, models rely heavily on correlated or spurious features suffering severely reduced accuracy if these are absent or removed. To study this effect, they introduce SALIENT IMAGENET (Singla and Feizi, 2022) and HARD IMAGENET (Moayeri et al., 2022), providing soft masks for causal and correlated features and segmentation masks, respectively. Hendrycks et al. (2021) propose IMAGENET-A, a dataset of 7 500 single-entity, natural adversarial examples, which induce high-confidence misclassifications in a set of `ResNet50` and belong to 200 IMAGENET classes that were chosen to minimize class overlap and the potential for fine-grained errors (see App. C for an analysis). Vasudevan et al. (2022) collect images where state-of-the-art models fail in an unexplained manner in IMAGENET-MAJOR. Taesiri et al. (2023) investigate the effect of zoom and crop on classifier performance and introduce IMAGENET-HARD as the set of images (from a range of IMAGENET-like datasets) that none of their considered models classified correctly for any crop. Idrissi et al. (2023) annotate IMAGENET images with respect to how they differ from what is prototypical for their class, introducing IMAGENET-X. This allows them to study the effect of these image-specific variations such as atypical pose, background, or lighting situation, on accuracy over a large number of models. In contrast, our (and Vasudevan et al. (2022)'s) work focuses more on systematic errors caused (partially) by labeling (set) choices rather than on what makes an individual image hard to classify.

## 3   Categorizing ImageNet Errors

In this section, we introduce our automated error classification pipeline, which aims to explain model errors by assigning it one of six error types. We consider errors that can not be explained in this way to be particularly severe *model failures*. While the definitions of our error types are heavily inspired by Vasudevan et al. (2022), we address three main issues of their manual approach with our automated pipeline: (i) it is time-consuming even for precise models and intractable for imprecise ones, (ii) it requires a panel of expert reviewers, typically not available, and (iii) it introduces inconsistencies due to human expert error, disagreement, or ambiguity.

Below, we first provide a brief overview of our pipeline (illustrated in Fig. 1), before discussing the different error types and how we identify them in more detail.

We consider errors due to overlapping class definitions (discussed in §3.1) and missing multi-label annotations (§3.2) to be the least severe, as they can be interpreted as labeling errors rather than classification errors. When a model predicts a class that is closely related to one of the labels, we call this a fine-grained classification error (§3.3), as the model succeeds in recognizing the type of entity but fails during the fine-grained distinction. When an image contains an entity that does not belong to any of the IMAGENET classes and a model predicts a label that is closely related to this (out-of-vocabulary) class, we call this a fine-grained out-of-vocabulary error (§3.4). We consider both types of fine-grained errors to be minor. If a model fails on an image that shows a very

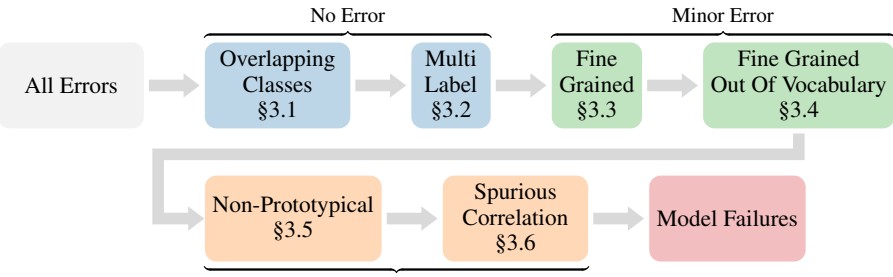

Figure 1: We first remove errors w.r.t. the original IMAGENET labels caused by overlapping class definitions or missing multi-label annotations, yielding multi-label accuracy (MLA). We then, in this order, identify fine-grained misclassifications, fine-grained misclassifications where the true label of the main entity is not included in the IMAGENET labelset, non-prototypical examples of a given class, and spurious correlations. This leaves us with severe model failures that are unexplained by our categorization.

non-prototypical instance of a given class, we call this a non-prototypical error (§3.5). If a model predicts a class that commonly co-occurs with a ground-truth class but is not shown in the image, we attribute this error to spurious correlation and discuss it in §3.6. We denote both of these errors as explainable errors. If an error can be attributed to multiple causes, we assign the least severe category and thus design our pipeline to consider error types in order of increasing severity (see Fig. 1).

Throughout this section, we include examples and trends for the kinds of errors we discuss. We generally describe these separately for images with a ground-truth class describing an organism (410 of 1000 classes) and artifacts (522 classes) (in line with previous work (Shankar et al., 2020)), as we observe that these two groups of classes exhibit different error patterns. These two groups account for almost all IMAGENET classes, with the remaining 68 classes being assigned the group "other". Where we find interesting trends, we further distinguish models by their (pre-)training dataset ranging in size from one million (IMAGENET) to multiple billion images (INSTAGRAM, LAION-2B), their architecture, including a broad range of MLPs, CNNs, transformer-based, and hybrid models, and their (pre-) training method. For the full details on all 962 models we consider, please refer to App. F. We provide more examples of every error type in App. E and more detailed trends in App. B.

## 3.1 Class Overlap

Prior work has established that a small number of IMAGENET classes exhibit extreme overlap (Northcutt et al., 2021b; Vasudevan et al., 2022). We illustrate one such example in Fig. 2: `tusker` is defined as "an animal with tusks"[2], which describes a strict superset of `african elephant` and has a significant overlap with `indian elephant` (females have short and sometimes no tusks). To avoid penalizing a model for correct predictions that do not match the ground truth, we consider all predictions describing a *superset or equivalent* of the ground-truth class to be correct. For example, we accept `tusker` for an image labeled `african`

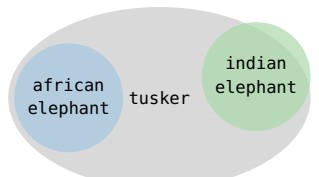

Figure 2: Venn-Diagram of the `tusker`, `indian elephant`, and `african elephant` classes.

`elephant`, but not vice-versa, as the latter might actually show a boar. We follow the mappings of equivalence and containment from Vasudevan et al. (2022) and refer to their App. C for full details.

In Fig. 3, we visualize the portion and number of top-1 errors identified to be correct, separating images with a ground-truth class describing an organism (green hues) and artifacts (red hues) and encoding the pre-training dataset size by assigning darker colors to larger datasets. While trends for artifacts and organisms seem very similar for portions of top-1 errors, we observe a clear difference when looking at their absolute numbers. There, two competing effects are at play. With increasing accuracy, more samples get classified correctly, in-

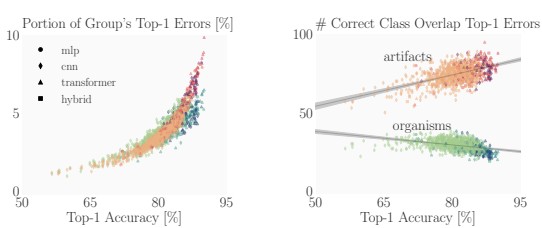

Figure 3: Portion (left) and number (right) of top-1 errors caused by *class overlap* by group – organisms (green) and artifacts (red). A 95% confidence interval linear fit is shown on the right.

---

[2]According to the Merriam-Webster Dictionary.

cluding as overlapping classes, thus increasing the number of such errors. On the other hand, more accurate models leverage labeling biases to pick the "correct" among equivalent classes (Tsipras et al., 2020), thus reducing the number of such errors. While the former effect seems to dominate for artifacts, the latter dominates for organisms.

## 3.2 Missing Multi-Label Annotations

While the IMAGENET dataset is annotated with a single label per image, many images contain multiple entities (Beyer et al., 2020; Shankar et al., 2020; Tsipras et al., 2020). Consequently, a model predicting the class of any such entity, different from the original IMAGENET label, is considered incorrect when computing top-1 accuracy. For example, the image shown in Fig. 4 contains ox, barn, and fence while only being labeled ox. To remedy this issue, multi-label accuracy (MLA) considers all shown classes, according to some multi-label annotation, to be correct. In this work, we use the annotations collected by Shankar et al. (2020) and improved by Vasudevan et al. (2022) combined with the mean class-wise accuracy definition of MLA (Shankar et al., 2020).

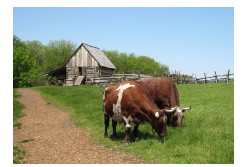

Figure 4: Image with label ox, but also showing the classes barn and fence. Example from Yun et al. (2021)

We visualize the portion and number of top-1 errors that turn out to be correct under multi-label evaluation in Fig. 5. When looking at the portion of errors (Fig. 5 left), we observe a similar trend for organisms and artifacts with missing multi-label annotations accounting for an increasing portion of top-1 errors as MLA increases and artifacts consistently more affected than organisms. When looking at the absolute numbers, we again observe the number of errors explained by multi-label annotations to first increase with ac-

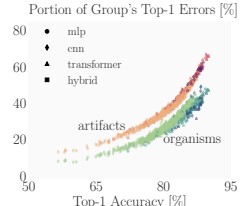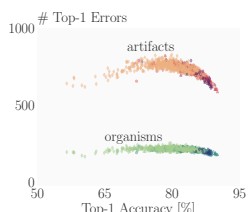

Figure 5: Portion (left) and number (right) of top-1 errors caused by *missing multi-label annotations* by group – organisms (green) and artifacts (red).

curacy as models become more precise, before decreasing again as models start to leverage labeling biases (Tsipras et al., 2020). As missing multi-label annotations explain up to 60% of model errors, we henceforth use MLA instead of top-1 accuracy as the reference for model comparison.

## 3.3 Fine-Grained Classification Errors

Many of the IMAGENET classes and especially the organism classes are very similar, making their fine-grained distinction challenging even for trained humans. For example, Fig. 6 is classified to cornet while actually showing a french horn, both brass wind instruments. While these errors can be corrected for the relatively small validation set using expert human judgment (Horn et al., 2015; Vasudevan et al., 2022), the much larger training set remains uncorrected. In this light, we believe a comprehensive model evaluation and comparison, should consider the failure to distinguish between very similar classes to be less severe than the failure to distinguish very different classes.

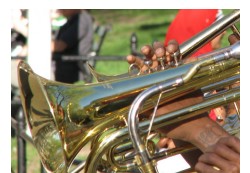

Figure 6: Image labeled french horn, but predicted to show a cornet.

To automatically detect such fine-grained classification errors, we manually review all 1000 IMAGENET classes and define semantically similar superclasses guided by whether an untrained human could reasonably confuse them. We obtain 161 superclasses, containing between 2 and 31 classes with an average of 6.7 and a median of 4. An additional 74 classes are not part of any superclass. 50 superclasses contain organisms (9.8 on average) and 101 contain artifacts (5.3 on average). We visualize the portion and num-

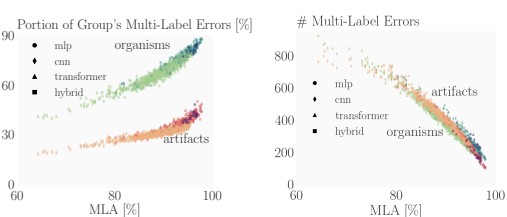

Figure 7: Portion (left) and number (right) of MLA errors caused by *fine-grained misclassifications* by group – organisms (green) and artifacts (red).

ber of fine-grained errors in Fig. 7. As expected, we observe significantly more fine-grained errors for organisms than for artifacts, explaining up to 88% and 50% of multi-label errors, respectively.

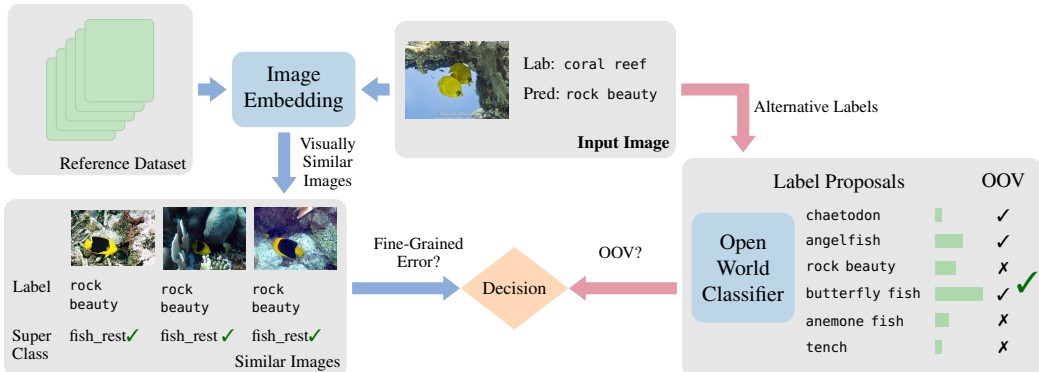

Figure 8: To confirm a fine-grained out-of-vocabulary error we proceed as follows. Given a misclassified input image, we first retrieve the 10 most visually similar images from the IMAGENET train set. If the superclass of any of these images matches the model prediction, we conclude that an entity with a class similar to our prediction is shown in the image. To confirm that this entity is indeed out-of-vocabulary, we first collect a set of proposal labels from WordNet (Miller, 1992) which are partially in and out of vocabulary. Then, we use an Open World Classifier to score each of the proposed labels. If the highest scored among these is not included in the IMAGENET labelset, we consider the analyzed error to be OOV.

## 3.4 Fine-Grained Out-of-Vocabulary Errors

Often images contain entities that do not belong to any IMAGENET class, we call these out-of-vocabulary. For example, Fig. 9 shows an image of two blue-cheeked butterflyfish, which is a class not part of the IMAGENET labelset. Instead, the target label is `coral reef`, describing the background of the image. The classifier predicts `rock beauty`, which is a WORDNET child of `butterflyfish` (despite being part of the angelfish and not the butterflyfish family) and the closest class in the IMAGENET labelset to the blue-cheeked butterflyfish. While an optimal classifier could be expected to always predict a class from the intersection of the contained entities and the IMAGENET labelset, the

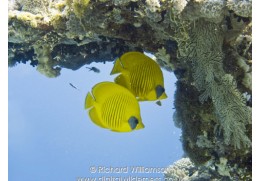

Figure 9: Image labeled `coral reef`, but prediction `rock beauty` (fish).

following is often observed in practice (Vasudevan et al., 2022). The classifier tries to classify a prominent entity in the image, but as it is not part of the IMAGENET labelset, it predicts a closely related class instead. We argue, that these cases should be treated at least as leniently as a fine-grained classification error, but they are harder to detect. As the entity the model tried to classify can not be assigned any IMAGENET label, it will not be included in any multi-label annotation and can thus not be captured by defining groups of closely related classes.

To still detect fine-grained out-of-vocabulary (OOV) errors, we propose the following approach, illustrated in Fig. 8 for the above example. We first retrieve the 10 most visually similar images from the IMA-GENET training set using cosine similarity in the CLIP embedding space (Radford et al., 2021), in this case all labeled `rock beauty`. If a label of any of these images belongs to the same superclass as the model's prediction

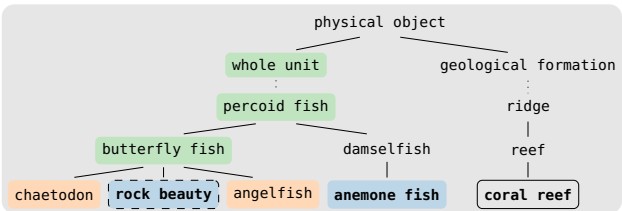

Figure 10: Illustration of proposal labels ▢▢. In-vocabulary **labels** are shown in bold. Classes in the same superclass as the prediction ⬚ are shown in blue boxes ▢, direct WORDNET siblings in orange ▢, and ancestors not shared with the label ▢ in green ▢.

(as defined in §3.3), we conclude that an entity similar to our prediction is shown in the image and proceed with our analysis. Otherwise, we conclude that the analyzed error is not fine-grained. To confirm that the shown entity is indeed out-of-vocabulary, we first collect a set of proposal labels from the following (illustrated in Fig. 10): all IMAGENET labels in the same superclass as the model's prediction (IV), all direct WORDNET siblings of the model's prediction (IV and OOV), and all WORDNET ancestors of the model's prediction, up to but excluding the first common ancestor with the IMAGENET label (IV and OOV). Finally, we use CLIP as an open-world classifier to score each of the proposed labels. If the highest scoring class is not included in the IMAGENET label set, we consider the shown entity to be OOV and conclude that this error was indeed fine-grained OOV.

We visualize the portion and number of multi-label errors categorized as fine-grained OOV in Fig. 11. Interestingly, we observe that fine-grained OOV errors are not only much more prevalent in artifacts but also that their portion increases much more quickly with MLA for artifacts. We hypothesize that this is due to the IMA-GENET labels covering organisms occurring in the validation set much more comprehensively than artifacts, and many im-

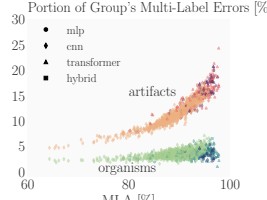 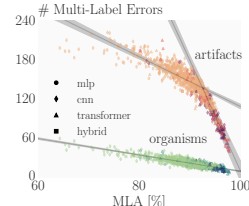

Figure 11: Portion (left) and number (right) of MLA errors identified as *fine-grained OOV* by group – organisms (green) and artifacts (red). 95% confidence interval linear fit is shown on the right. For artifacts, models are divided at 93% MLA.

ages of artifacts being cluttered and full of other (OOV) objects. We note that this might be a reason why trained humans still outperform even state-of-the-art models on artifacts but not organisms (Shankar et al., 2020). Interestingly, we observe that, across pretraining datasets and model sizes, the number of fine-grained OOV errors for artifact drops much quicker with MLA above around 93% MLA (see confidence intervals in Fig. 11 right).

## 3.5 Non-Prototypical Instances

Many concepts described by any one IMAGENET class label are broad, depend on social and geographic factors, and change over time. Combined with the search-based data collection process, this has led to a biased dataset with skewed representations of the concepts described by the contained classes. Therefore it is unsurprising that models perform worse on non-prototypical instances or pictures of any given object, *i.e.*, instances that, while clearly belonging to a given class, can be considered outliers in the IMAGENET distribution of that class.

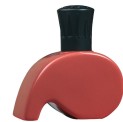

Figure 12: Image with label whistle, but predcition perfume.

For example, in Fig. 12 we see a non-prototypical whistle. We believe it is interesting to track progress on these hard, non-prototypical instances as they are likely to be a good indicator to what extent a model has learned to recognize the underlying concepts of a class. However, defining what constitutes a non-prototypical instance of any class is hard and to a large extent subjective. Fortunately, this error

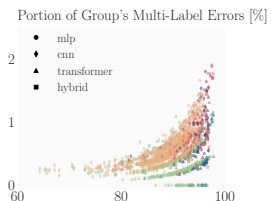 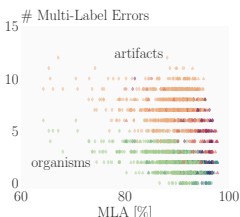

Figure 13: Portion (left) and number (right) of MLA errors identified as *non-prototypical sample*.

type is independent of the (incorrect) prediction made by the model, allowing us to directly leverage the manual categorization of non-prototypical images by Vasudevan et al. (2022). We thus implicitly decide that all images that are classified correctly by the state-of-the-art models (ViT-3B and Greedy Soups) they considered are not sufficiently non-prototypical to explain an error by another model.

We visualize the number of errors caused by non-prototypical instances in Fig. 13. Interestingly and in contrast to all other error types, there is no strong correlation between performance on non-prototypical examples and overall MLA. This suggests that these non-prototypical examples are not inherently hard to classify for all models. Further surprisingly, non-prototypical examples account for a very similar portion of errors for artifacts and organisms, despite the appearance of artifacts of the same class varying significantly more, which we expected would lead to a larger portion of artifact errors being explained by their non-prototypical appearance.

## 3.6 Spurious Correlations

Entities of many classes frequently co-occur with features that have no causal relationship to that class. For example, oxen are frequently observed on green pastures (such as in Fig. 4). We call these correlated features (Neuhaus et al., 2022). Inherently this is not a problem, however, it has been observed that models frequently rely on these correlated features to make predictions (Choi et al., 2012; Beery et al., 2018; Neuhaus et al., 2022; Moayeri et al., 2022), for example, by predicting ox when shown only a green pasture or camel when shown

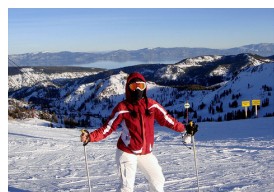

Figure 14: An image with label ski mask (and also multi-class label alp), but prediction ski.

a cow in the desert. When correlated features lead to prediction errors, we call these spurious correlations. Here, we focus on the case where the presence of a correlated spurious feature causes a misclassification to the correlated class, despite no corresponding entity being shown. In Fig. 14, we show an example of such a spurious correlation where the presence of a `ski mask` and `alps` causes a model to predict `ski`, despite the image containing no skis. We do not consider the error mode, where the absence of a correlated feature causes the model to *not* predict the correlated class, despite a corresponding entity being shown, investigated by Singla and Feizi (2022); Moayeri et al. (2022).

To identify errors caused by spurious correlations, we identify pairs of commonly co-occurring classes and then categorize errors as spurious correlations if an incorrect model prediction and a multi-label form such a co-occurrence pair. More concretely, we first extract all pairs of co-occurring labels from the ReaL multi-label annotations (Beyer et al., 2020), excluding samples we evaluate on. We then filter out pairs that either belong to the same superclass as defined in §3.3 or only co-occur once. Using

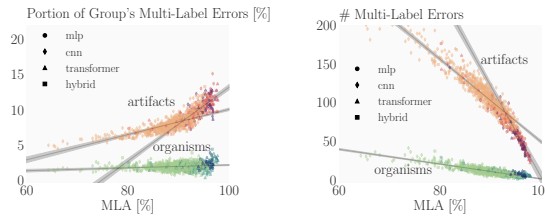

Figure 15: Portion (left) and number (right) of MLA errors caused by *spurious correlations* by group. A 95% confidence interval linear fit is shown on the right. For artifacts, models are divided into those with more and less than 93% MLA.

this process, we extract 13 090 label pairs from 6 622 images with more than one label, yielding 1019 unique co-occurrence pairs after filtering, which indicate a spurious correlation if they occur.

We visualize the portion and number of errors caused by spurious correlation in Fig. 15. We observe that artifacts and organisms follow notably different trends. Not only is the portion of errors caused by spurious correlations much larger for artifacts than organisms, but it also increases with MLA for artifacts (at an increased rate for higher MLA), while it stays roughly constant for organisms. For state-of-the-art models, spurious correlations explain up to 15% of errors on artifacts making them the second largest error source we identify.

# 4 Analysis of Model Errors on ImageNet

In this section, we discuss global trends, analyze their interaction with architecture and training choices, and validate our automatic pipeline against the manual analysis of Vasudevan et al. (2022).

**Model Failures** After removing all minor (see §3.3 and §3.4) and explainable (see §3.5 and §3.6) errors, from the multi-label errors (MLE), we are left with a set of particularly severe, unexplained model failures (MLF). In Fig. 16 we visualize the portion of these unexplained model failures over multi-label accuracy (MLA) and standard top-1 accuracy, again split by artifact (red) and organism (green). A ratio of 1 corresponds to none of the MLEs being explainable by our pipeline, while a ratio of 0 corresponds to all MLEs being explainable,

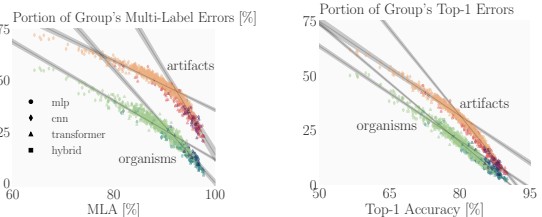

Figure 16: Portion of MLA (left) and top-1 (right) errors that can not be explained and are thus considered *model failures* – organisms (green) and artifacts (red). 95% confidence interval linear fit is shown shaded. Models are divided by MLA (at 93%) and top-1 accuracy (at 80%) for linear fits.

thus consisting only of less severe error types. Surprisingly, both top-1 accuracy and MLA are pessimistic when it comes to reporting model progress, with unexplained model failures decreasing at a much quicker rate than both top-1 errors and MLE.

Further, we observe different error distributions for organisms and artifacts. Firstly, the portion of model failures is much higher for artifacts than for organisms. Secondly, while the portion of unexplained errors decreases for both organisms and artifacts with MLA and top-1 accuracy, indicating that as models get more accurate they make not only less but also less severe errors, the rate of this change differs. For artifacts, this decrease is initially slower but then at around 93% MLA or 80% top-1 error, the portion of severe model failures starts to drop rapidly (roughly three times as fast as before), while the decrease is roughly linear for organisms. This phase change becomes particularly apparent when viewing the trend over MLA, where it is even visible for organisms.

## 4.1 (Pre-)training Dataset

Throughout §3, we have illustrated the pretraining dataset size with the marker hue. Generally, we observe that conditioned on identical MLA, the effect of pre-training dataset size is rather modest. Here, we divide the 12 datasets we consider into 4 categories from "small" (<5M) to "xlarge" (> 500M) depending on the number of included images, illustrating separate fits in Fig. 17, with a darker shade corresponding to a larger dataset. We observe that across a wide range of accuracies, larger pretraining datasets lead to a faster reduction of model failures with MLA for both artifacts and organisms. While this effect is partially explained by larger datasets more frequently leading to higher MLA, it is also observed for narrow MLA ranges. Considering individual error types, we observe that in particular fine-grained errors are more frequent for larger pretraining datasets. We hypothesize that this is due to these larger datasets leading to better underlying representations, allowing the model to succeed in the coarse classification for harder samples, while still failing on the fine-grained distinctions.

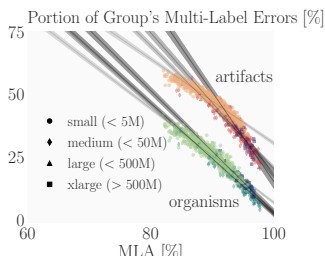

Figure 17: Portion of model failures (MLF) over MLA depending on pre-training dataset size with 95% confidence intervals for models with at least 82% MLA. Confidence intervals are shown darker for larger pretraining dataset sizes.

## 4.2 Model Architecture

In Fig. 18, we again show the portion of unexplained model failures over MLA, this time broken down by architecture type and colored by the number of parameters (larger models have darker hues). To investigate whether modern convolutional architectures (ConvNeXts (Liu et al., 2022)) and vision transformers (ViT (Dosovitskiy et al., 2021), Swin (Liu et al., 2021), EVA (Fang et al., 2023)) exhibit different trends in terms of model failures, we focus on state-of-the-art models with at least 92% MLA that were trained using a dataset with at least 50M images, leaving 28 CNN-based and 79 transformer-based models. While we observe clear trends of larger models performing better, the rate of model failures seems to be largely independent of model size when conditioned on MLA. We do observe a slight trend where the model failure rate of transformers decreases faster with MLA for organisms but slower for artifacts when compared to ConvNeXts. We speculate that this might be due to the ConvNeXts leveraging textures more heavily (Geirhos et al., 2019), which could be particularly helpful for fine-grained class distinctions on organisms leading to a larger portion of multi-label errors being model failures at the same MLA.

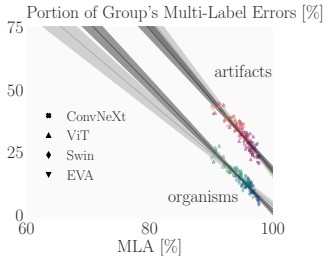

Figure 18: Portion of model failures (MLF) over MLA depending on model architecture. 95% confidence interval for transformer-based models is shown darker and for CNN-based models lighter.

## 4.3 Comparison to Vasudevan et al. (2022)

To evaluate the alignment of our automated error analysis with a panel of human experts, we compare its results to those of Vasudevan et al. (2022) on both networks they consider, here for `ViT-3B` and in App. A for `Greedy Soups`.

Comparing the results in Table 1, we confirm that our pipeline is conservative, classifying 16.4% (62 / 378) of the errors assigned an explanation by Vasudevan et al. (2022) as model failures. On the remaining errors, our pipeline agrees with the panel of human experts in 73.1% of the cases. By manually inspecting the 85 differently classified errors, we determine that for only 32 of these (8.4% of all errors), the categorization by Vasudevan et al. (2022) is clearly preferable to ours. Furthermore, our pipeline encodes a minimal-severity bias, absent in Vasudevan et al. (2022)'s categorization. That is, when multiple error explanations would be valid, our pipeline consistently chooses the least severe one. This highlights a further advantage of our approach. While any human judgment is inherently subjective and thus prone to differ between expert panels or even for the same panel over time, as acknowledged by Vasudevan et al. (2022), our approach is consistent and repeatable.

Table 1: Comparison of our automated analysis of the errors of a `ViT-3B` model with the manual annotations from (Vasudevan et al., 2022). Rows correspond to the error categories assigned by a human expert while columns correspond to the error types assigned by our automated pipeline.

| Error categories | FG | FG OOV | Non-prot. | Spur. Corr. | Model failures | Total (row) |
|---|---|---|---|---|---|---|
| Fine-grained | 192 | 15 | 0 | 10 | 25 | 242 |
| Fine-grained OOV | 9 | 20 | 0 | 11 | 14 | 54 |
| Non-prototypical | 13 | 2 | 12 | 3 | 0 | 30 |
| Spurious Correlation | 10 | 12 | 0 | 7 | 23 | 52 |
| Total (col) | 224 | 49 | 12 | 31 | 62 | 378 |

Observing similar trends for `Greedy Soups` in App. A, we confirm that on all models for which human expert error categorizations are available, our automated pipeline is well aligned with their judgment while providing consistent, repeatable, and conservative error categorizations.

## 5   Limitations

In this section, we briefly reflect on the limitations of our work.

**Personal Bias**   Our choices in the implementation of the error analysis pipeline reflect our personal biases and decisions, such as the superclasses we picked in §3.3. This is not only limited to our personal biases, but, as our pipeline relies on prior work in several places, it also encodes the biases of their authors. For example, we rely on the class overlap mappings from Vasudevan et al. (2022) and multi-class labels by Shankar et al. (2020) and Vasudevan et al. (2022).

**Extension to New Datasets**   To be applied to a new dataset, our pipeline requires multi-label and non-prototypicality annotations and superclass definitions. While less precise than using a human-annotated gold standard, multi-label annotations could be sourced using a Re-Label (Yun et al., 2021) approach, allowing (given sufficient scale) spurious correlation pairs to be extracted from co-occurrence frequencies. After defining superclasses manually as required by their very definition, this would allow all error types except for the rare non-prototypical instances to be categorized. With the rest of the pipeline in place, non-prototypical instances could be annotated by reviewing the uncategorized errors shared by multiple well-performing models. However, while feasible these steps still require a significant amount of manual work. We showcase a (partial) adaption to the similar IMAGENET-A dataset in App. C.

## 6   Conclusion

As state-of-the-art models come close to and exceed human performance on IMAGENET, focus is increasingly shifting towards understanding the last remaining errors. Towards this goal, we propose an automated error categorization pipeline that we use to study the distribution of different error types across 962 models, of different scales, architectures, training methods, and pre-training datasets. We distinguish between minor errors, constituting failures on fine-grained class distinctions both when the ground truth was in and out-of-vocabulary, explainable errors, constituting failures on non-prototypical examples and due to spurious correlations, and unexplained model failures, constituting all remaining errors. We find that even MLA is a pessimistic measure of model progress with the portion of severe errors quickly decreasing with multi-label-accuracy. Further, we find that organism and artifact classes exhibit very different trends and prevalences of error types. For example, we observe that fine-grained class distinctions are a much bigger challenge for organisms than for artifacts, while artifacts suffer much more from spurious correlations and out-of-vocabulary errors, with these trends becoming increasingly pronounced as models become more accurate. We believe that such an analysis of a new method's effects on model error distributions can become an important part of the evaluation pipeline and lead to insights guiding future research in computer vision.

## Acknowledgements

We thank our anonymous reviewers for their constructive comments and insightful feedback.

This work has been done as part of the EU grant ELSA (European Lighthouse on Secure and Safe AI, grant agreement no. 101070617) and the SERI grant SAFEAI (Certified Safe, Fair and Robust Artificial Intelligence, contract no. MB22.00088). Views and opinions expressed are however those of the authors only and do not necessarily reflect those of the European Union or European Commission. Neither the European Union nor the European Commission can be held responsible for them.

The work has received funding from the Swiss State Secretariat for Education, Research and Innovation (SERI).

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

# A  Extended Comparison to Vasudevan et al. (2022)

Similarly to the analysis in Sec. 4.3, we now evaluate our automated error classification pipeline on `Greedy Soups`, the only other model for which manual error categorizations exist (Vasudevan et al., 2022), showing the results in Table 2. We do not assign an explanation to $11.6\%$ (29 / 249) of the errors, classifying them as model failures. On the remaining errors, our pipeline agrees with the human expert annotation in $74.1\%$ (163 / 220) of the cases. After a manual review of the remaining 57 differently classified errors, we determine that the categorization provided by Vasudevan et al. (2022) is clearly preferable to ours for only 26 samples ($10.4\%$ of all errors). Overall, the results for `Greedy Soups` are very similar to those for `ViT-3B` in §4.3, thus further validating our modeling choices. Therefore, we believe that our automatic categorization is indeed aligned with the opinion of human experts and note that an efficient and automatic classification pipeline is valuable even if it is imperfect, as it enables the study of models and trends at scale.

Table 2: Comparison of our automated analysis of the errors of a `Greedy Soups` model with the manual annotations from (Vasudevan et al., 2022). Rows correspond to the error categories assigned by a human expert while columns correspond to the error types assigned by our automated pipeline.

| Error categories | FG | FG OOV | Non-prot. | Spur. Corr. | Model failures | Total (row) |
|---|---|---|---|---|---|---|
| Fine-grained | 139 | 14 | 1 | 7 | 11 | 172 |
| Fine-grained OOV | 7 | 8 | 0 | 5 | 6 | 26 |
| Non-prototypical | 8 | 1 | 8 | 2 | 0 | 19 |
| Spurious Correlation | 7 | 5 | 0 | 8 | 12 | 32 |
| Total (col) | 161 | 28 | 9 | 22 | 29 | 249 |

# B Additional Results

In addition to the analysis presented in the main paper, which focused on the portion of errors due to the different error categories for artifacts and organisms, we provide more detailed results in this section, considering absolute error counts as well as portions for artifacts, organisms, the remaining classes, and all classes jointly.

Figures 19-24 contain these statistics grouped by error type:

- Fig. 19 – top-1 errors caused by class overlap (§3.1);
- Fig. 20 – top-1 errors due to missing multi-label annotations (§3.2);
- Fig. 21 – fine-grained multi-label errors (§3.3);
- Fig. 22 – fine-grained out-of-vocabulary multi-label errors (§3.4);
- Fig. 23 – multi-label errors due to non-prototypical instances (§3.5);
- Fig. 24 – multi-label errors due to various spurious correlations (§3.6).

We show the number of errors in relative and absolute terms. In each figure, subfigures (a) and (b) show the results for each group of classes separated in "artifacts", "organisms" and "others" (the rest of the labels), where for (a) the relative portions of top-1 or multi-label errors are computed *per group*, *i.e.*, by diving by the number of top-1 or multi-label errors in the respective group. Red hues again correspond to artifacts, green hues – to organisms, and blue hues– to the remaining classes (others). Subfigures (c) and (d) present the relative and absolute statistics for all classes jointly.

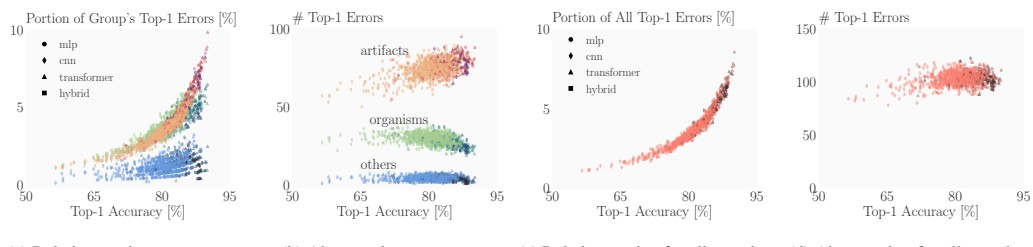

(a) Relative portion per group.    (b) Abs. number per group.    (c) Relative portion for all samples.    (d) Abs. number for all samples.

Figure 19: Top-1 errors caused by class overlap. The relative portion of these errors on "others" is the smallest and the trend is noisier compared to "artifacts" and "organisms". This is possibly due to the low absolute number of classes in this group (68 out of 1000). In absolute terms, the number of this kind of error remains relatively constant.

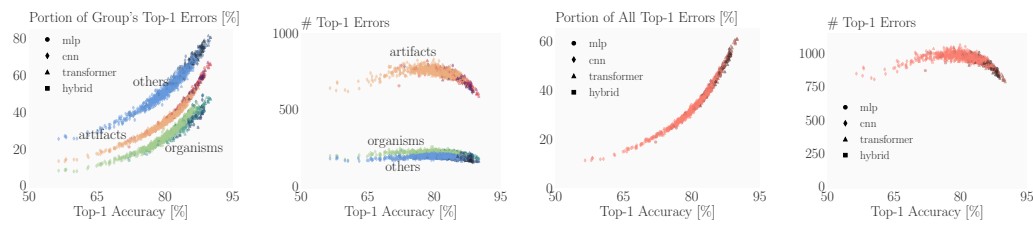

(a) Relative portion per group.    (b) Abs. number per group.    (c) Relative portion for all samples.    (d) Abs. number for all samples.

Figure 20: Top-1 errors caused by missing multi-label annotations. Surprisingly, the portion of these errors is the highest on the "others" group. In absolute terms, "organisms" and "others" result in roughly the same constant amount of errors, despite the much larger number of organism classes (410 vs. 68). For "artifacts", it first increases and then starts dropping, possibly due to more accurate models learning to leverage IMAGENET labeling biases.

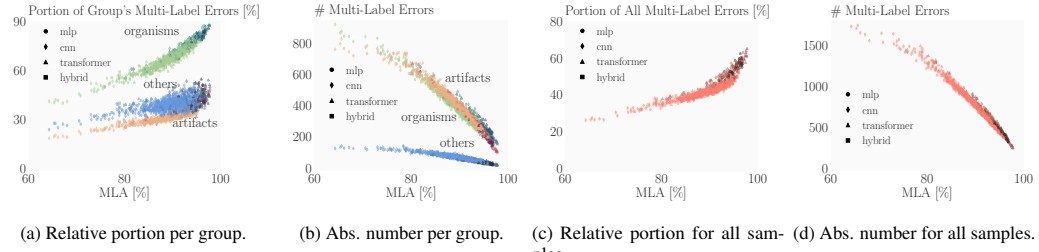

(a) Relative portion per group.  (b) Abs. number per group.  (c) Relative portion for all sam- (d) Abs. number for all samples.
ples.

Figure 21: Multi-label errors due to fine-grained misclassifications. The portion of fine-grained errors on "others" is larger than those on "artifacts" but smaller than "organisms".

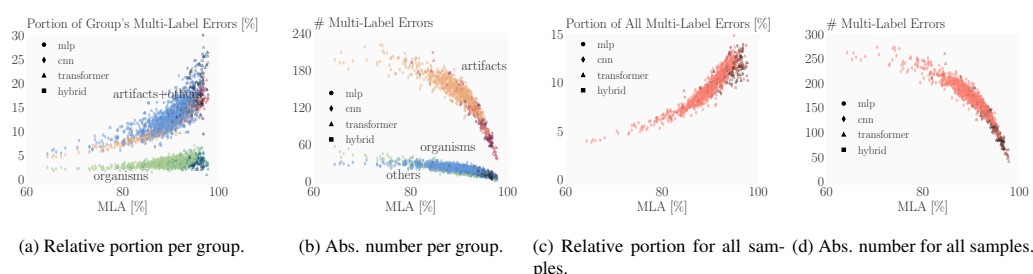

(a) Relative portion per group.  (b) Abs. number per group.  (c) Relative portion for all sam- (d) Abs. number for all samples.
ples.

Figure 22: Fine-grained out-of-vocabulary multi-label errors. The portion of errors follows a similar trend for "artifacts" and "others", while the sudden sharp drop in absolute numbers for "artifacts" is absent for "others".

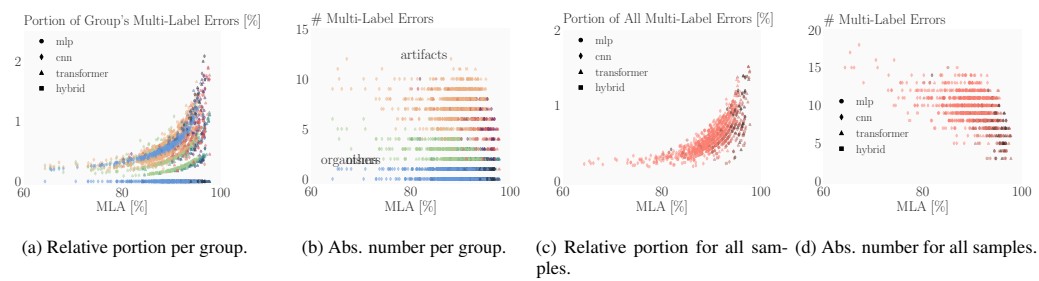

(a) Relative portion per group.  (b) Abs. number per group.  (c) Relative portion for all sam- (d) Abs. number for all samples.
ples.

Figure 23: Multi-label errors identified as non-prototypical instances. The number of samples identified as non-prototypical by Vasudevan et al. (2022) is inherently small (36) and the majority of them are artifacts, leading to noisy results. As the models get better at other types of errors, the relative portion of multi-label errors due to non-prototypical images increases.

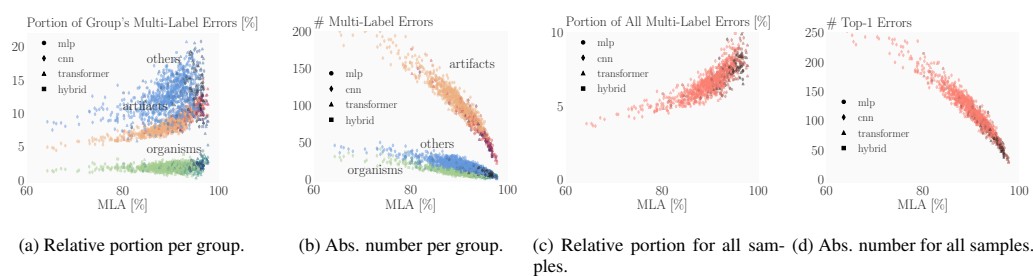

(a) Relative portion per group.  (b) Abs. number per group.  (c) Relative portion for all sam- (d) Abs. number for all samples.
ples.

Figure 24: Multi-label errors caused by spurious correlations. In relative terms, "artifacts" and "others" follow a similar trend which is noisier for the "others" group.

Finally, in Fig. 25 we present the ratio of unexplained model failures (MLF) over multi-label accuracy (MLA) and standard top-1 accuracy. Again, subfigures (a) and (b) are organized per group, while (c) and (d) show the results for all samples.

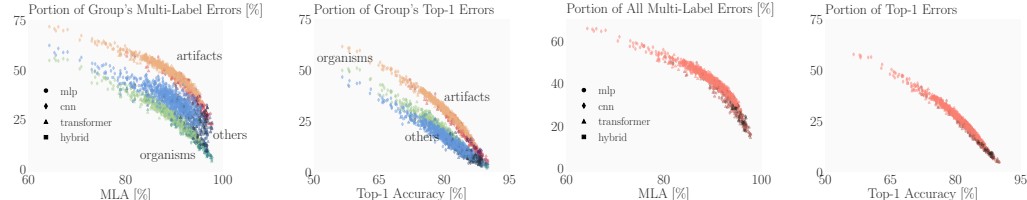

(a) Portion of unexplained model failures per group.

(b) Portion of unexplained model failures per group.

(c) Portion of unexplained model failures for all samples.

(d) Portion of unexplained model failures for all samples.

Figure 25: Ratio of unexplained model failures (MLF) over multi-label accuracy (MLA – (a) and (c)) and standard top-1 accuracy ((b) and (d)). In general, our pipeline explains the biggest portion of errors made on "organisms" and the lowest portion of errors on "artifacts". However, better models quickly close this gap. For the best models, our pipeline explains up to 85% of all multi-label errors and 95% of all top-1 errors.

# C Analysis of IMAGENET-A

To demonstrate the applicability of our automated error analysis pipeline to other datasets, we now apply it to IMAGENET-A. IMAGENET-A was introduced by Hendrycks et al. (2021) as an IMAGENET-like test set of "Natural Adversarial Examples". It contains 7500 images from 200 IMAGENET classes, picked to fool a set of ResNet-50 classifiers, while still constituting high-quality, single-entity images, clearly depicting an entity of

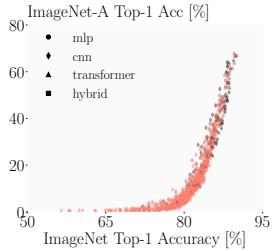 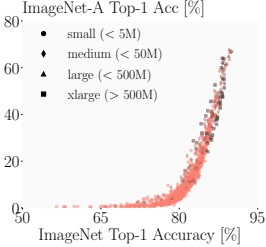

Figure 26: IMAGENET-A top-1 Accuracy vs IMAGENET top-1 Accuracy grouped by architecture (left) and training data size (right).

the target class. A comparison of IMAGENET and IMAGENET-A accuracy is shown in Fig. 26. In this section, marker color encodes model parameter counts, i.e., larger models have darker markers.

As IMAGENET-A images were selected to be challenging for IMAGENET classifiers at the time, we see that many classifiers with IMAGENET accuracies below 80% have almost 0 accuracy on IMAGENET-A. In fact, 801 of the 962 models we consider have an accuracy of less than 30% on IMAGENET-A. As classifiers improved (> 80% top-1 IMAGENET accuracy) IMAGENET-A accuracy catches up rapidly. Among other improvements, the main driver seems to be the model parameter count. Note that due to the much lower IMAGENET-A accuracy (of some models), the x-axis of all further plots starts at 0% rather than 60%, as before.

As IMAGENET-A uses IMAGENET classes, we keep the split into organisms (77 out of 200 classes, 38.5%), artifacts (104 / 200, 52%), and other classes (19 / 200, 9.5%).

**Class Overlap** Due to the shared class structure, we can use the same analysis as for IMAGENET and show the results in Fig. 27. The 200 classes in IMAGENET-A were selected from the 1000 IMAGENET classes to avoid class overlap and label ambiguity. Thus, as expected, we see a much lower overall amount of such errors (below 0.8% instead of up to 8.5% for IMAGENET). In particular, for artifacts and other classes, we see no

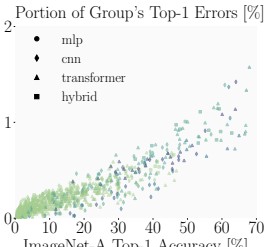 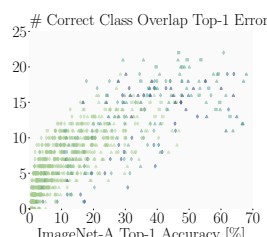

Figure 27: Portion (left) and number (right) of top-1 errors caused by *class overlap* for organisms (green). For the artifacts and other groups we observe no errors.

class overlap errors at all. Interestingly, for organisms, we still observe a small number (≤ 22) of errors caused by class overlap. We observe that errors become more frequent (albeit still rare) for models with higher accuracy. As with IMAGENET this is due to more accurate models classifying more images correctly, including to overlapping classes.

**Multi-Label Annotations** In contrast to IMAGENET, IMAGENET-A was designed to only include single-entity images. Thus, we skip the multi-label analysis for IMAGENET-A and use top-1 accuracy/error rather than MLA accuracy/error for all further analyses.

**Fine-Grained Classification Errors** Due to the shared class structure, we can apply the same analysis including superclasses for fine-grained classification errors as on IMAGENET. We show results in Fig. 28.

As with IMAGENET, the group for which we observe most of these errors are organisms (explaining up to 47% of the group's errors). However,

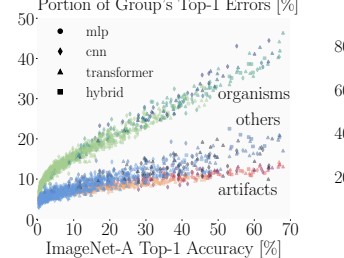 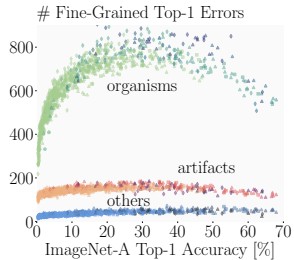

Figure 28: Portion (left) and number (right) of top-1 errors caused by *fine-grained misclassifications* by group – organisms (green), artifacts (red), and others (blue).

the portion of explained errors is much lower than for IMAGENET, where up to 88% of MLA errors for organisms can be explained by fine-grained classification errors. We see the percentage of errors increase linearly with accuracy. Looking at absolute counts, especially for organisms, we see that fine-grained errors initially become more prevalent as model accuracy increases before dropping off as models more frequently also succeed in the fine-grained class distinction.

**Fine-Grained Out-of-Vocabulary Errors** Again, we can use the same superclass definition, CLIP embedding and open world classifier as for IMAGENET to identify fine-grained OOV errors. We generally observe slightly larger portions of OOV errors than on IMAGENET and a much clearer linear dependence on accuracy (see Fig. 29). Notably, the superlinear increase observed on IMAGENET (see Fig. 11) is absent here. Looking at absolute counts, after an initial increase and stagnation, we observe a linear decrease with model accuracy. In contrast to IMAGENET, we again do not observe a slope change.

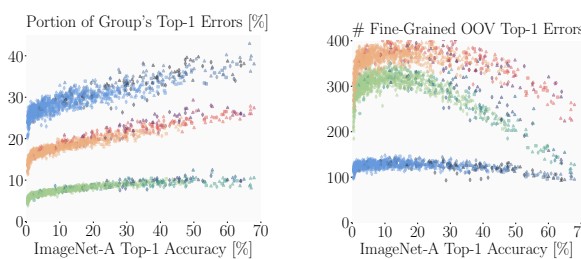

Figure 29: Portion (left) and number (right) of top-1 errors identified as *fine-grained OOV* by group – organisms (green), artifacts (red) and other (blue).

**Non-Prototypical Instances** Our analysis pipeline for IMAGENET uses the manual annotation of non-prototypical instances from (Vasudevan et al., 2022). As we do not have these available for IMAGENET-A we skip this analysis step here. However, for standard IMAGENET we only observe a small number of such errors ($\leq 2\%$ of MLE) and note that the IMAGENET-A images were manually selected to only include high-quality images (Hendrycks et al., 2021). Thus, we expect their portion to be even smaller here.

**Spurious Correlations** Again following our IMAGENET approach for spurious correlation errors we obtain the results in Fig. 30. At first glance, we observe much less spurious correlations than for IMAGENET (6.2% compared to 15% for artifacts). However, we observe a more linear instead of super-linear increase in the portion of errors explained by spurious correlations, leading to this reduction being smaller for less accurate models.

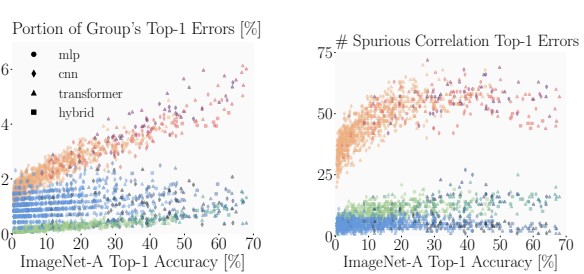

Figure 30: Portion (left) and number (right) of top-1 errors identified as *spurious correlations* by group – organisms (green), artifacts (red) and other (blue).

**Model Failures** Looking at the remaining failures, we see that we can only explain up to half of the errors for the best-performing models. Not only are top-1 and MLA error rates much higher than for IMAGENET but also a much larger portion of the remaining errors constitutes severe model failures, highlighting that IMAGENET-A is a much harder dataset.

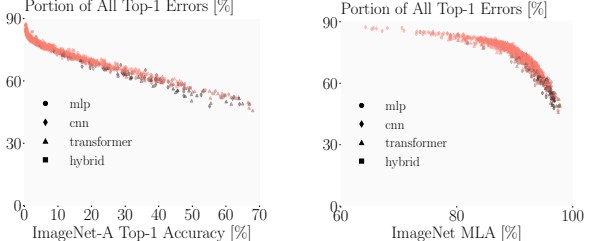

Figure 31: Portion for remaining failures over IMAGENET-A top-1 (left) and IMAGENET MLA (right).

However, analyzing Fig. 31, we observe that both the IMAGENET-A top-1 and IMAGENET MLA accuracy are good predictors for the portion of model failures which decreases with increasing model accuracy. This highlights again and perhaps surprisingly that IMAGENET MLA underreports progress in model performance, even on particularly challenging subdistributions like IMAGENET-A.

# D Extended Examples on Fine-Grained OOV Error Classification

In this section we provide further step by step examples of the procedure that determines if a given model mistake is categorized as a fine-grained OOV error. The sample in Fig. 32 is confirmed to be fine-grained OOV, the one in Fig. 33 is not fine-grained and the one in Fig. 34 is rejected as OOV.

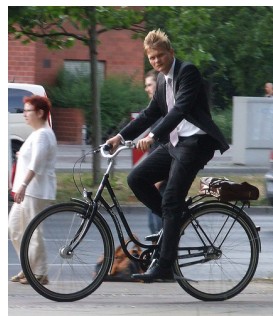

GT: `suit`
Pred: `bicycle-built-for-two`
Pred superclass: `motor_cycle`

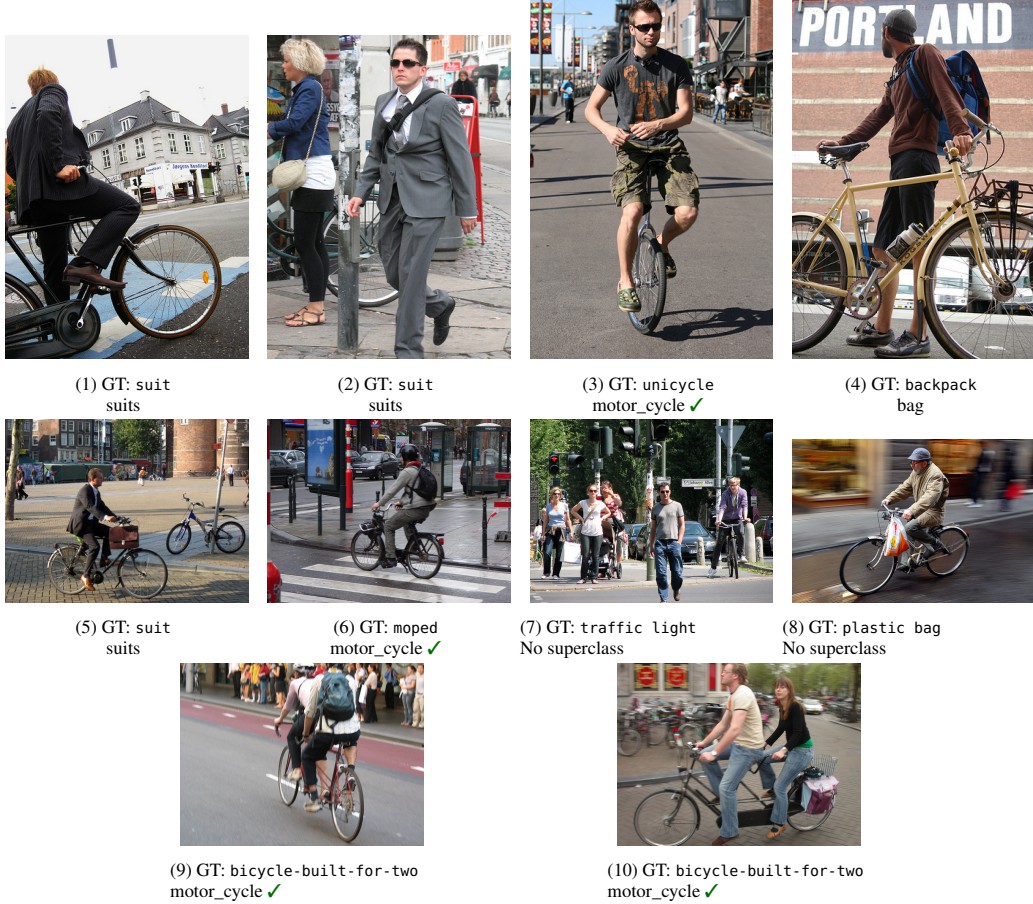

(1) GT: `suit`
suits

(2) GT: `suit`
suits

(3) GT: `unicycle`
motor_cycle ✓

(4) GT: `backpack`
bag

(5) GT: `suit`
suits

(6) GT: `moped`
motor_cycle ✓

(7) GT: `traffic light`
No superclass

(8) GT: `plastic bag`
No superclass

(9) GT: `bicycle-built-for-two`
motor_cycle ✓

(10) GT: `bicycle-built-for-two`
motor_cycle ✓

Figure 32: **Fine-grained OOV confirmed**. The evaluated validation sample (top) has a ground-truth IMAGENET label (GT) `suit`, but the model predicts `bicycle-built-for-two` which falls in the "motor_cycle" superclass. Below, we show the top-10 training images that are the most visually similar to the validation sample (according to CLIP similarity) together with their IMAGENET labels and the superclasses of these labels. The labels of images 3, 6, 9 and 10 are in the same superclass as the predicted label (motor_cycle), hence we proceed with the procedure. After obtaining the in-vocabulary and OOV label proposals, as described in Sec. 3.4, the proposals with the highest probability according to CLIP are: `safety bicycle`, `ordinary bicycle`, `velocipede` and `bicycle`, all of which are OOV, confirming that the error is indeed fine-grained OOV.

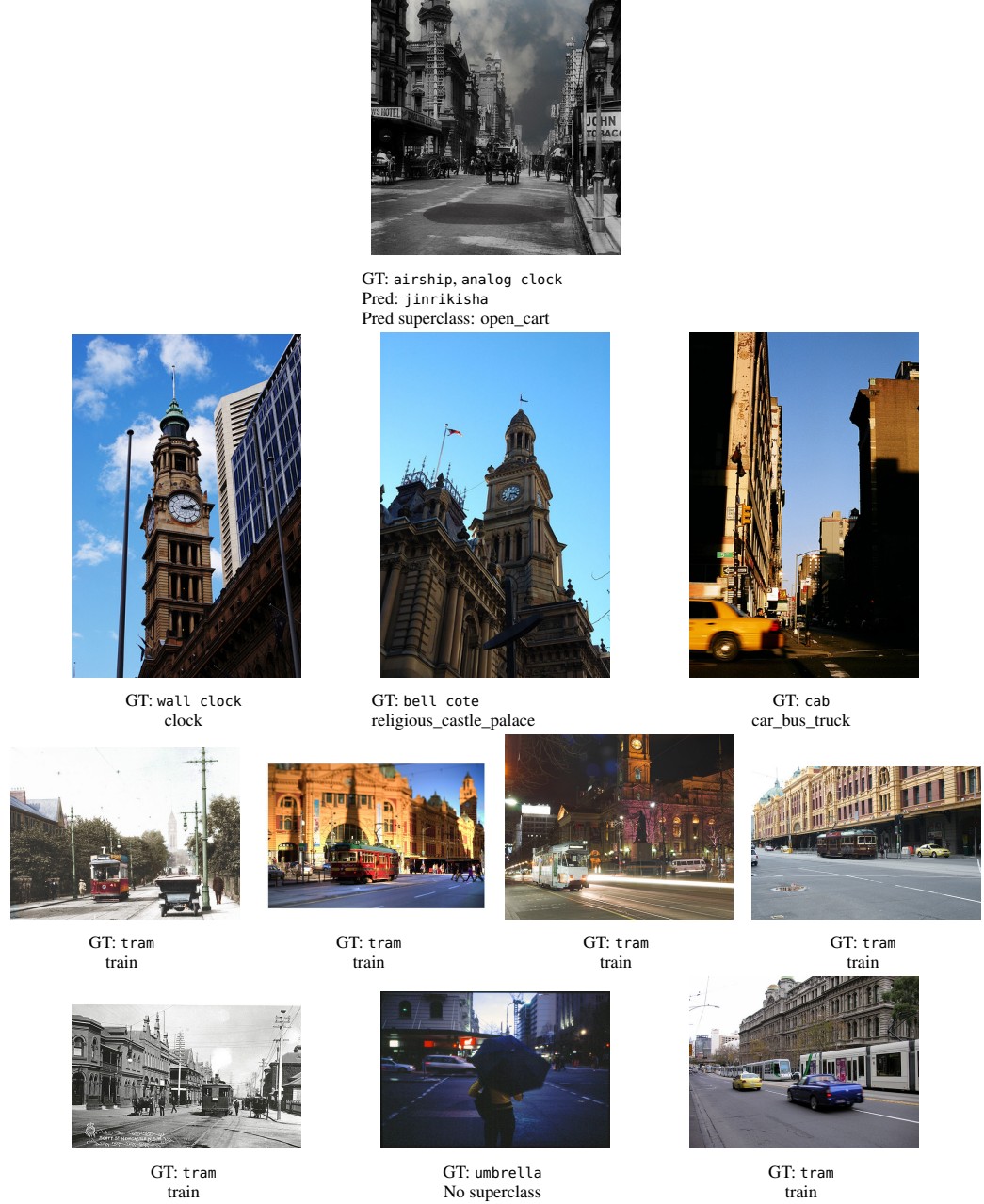

GT: airship, analog clock
Pred: jinrikisha
Pred superclass: open_cart

GT: wall clock
clock

GT: bell cote
religious_castle_palace

GT: cab
car_bus_truck

GT: tram
train

GT: tram
train

GT: tram
train

GT: tram
train

GT: tram
train

GT: umbrella
No superclass

GT: tram
train

Figure 33: **Fine-grained OOV rejected**. The evaluated validation sample (top) has ground-truth IMAGENET multi-labels (GT) airship and analog clock, but the model predicts jinrikisha, arguably attempting to classify the carriages on the street. The predicted label is in the "open_cart" superclass. Next, we show the top-10 training images that are the most visually similar to the validation sample (according to CLIP similarity) together with their IMAGENET labels and the superclasses of these labels. None of the labels is in the "open_cart" superclass, thus we terminate the procedure and reject the mistake as a possible fine-grained OOV error.

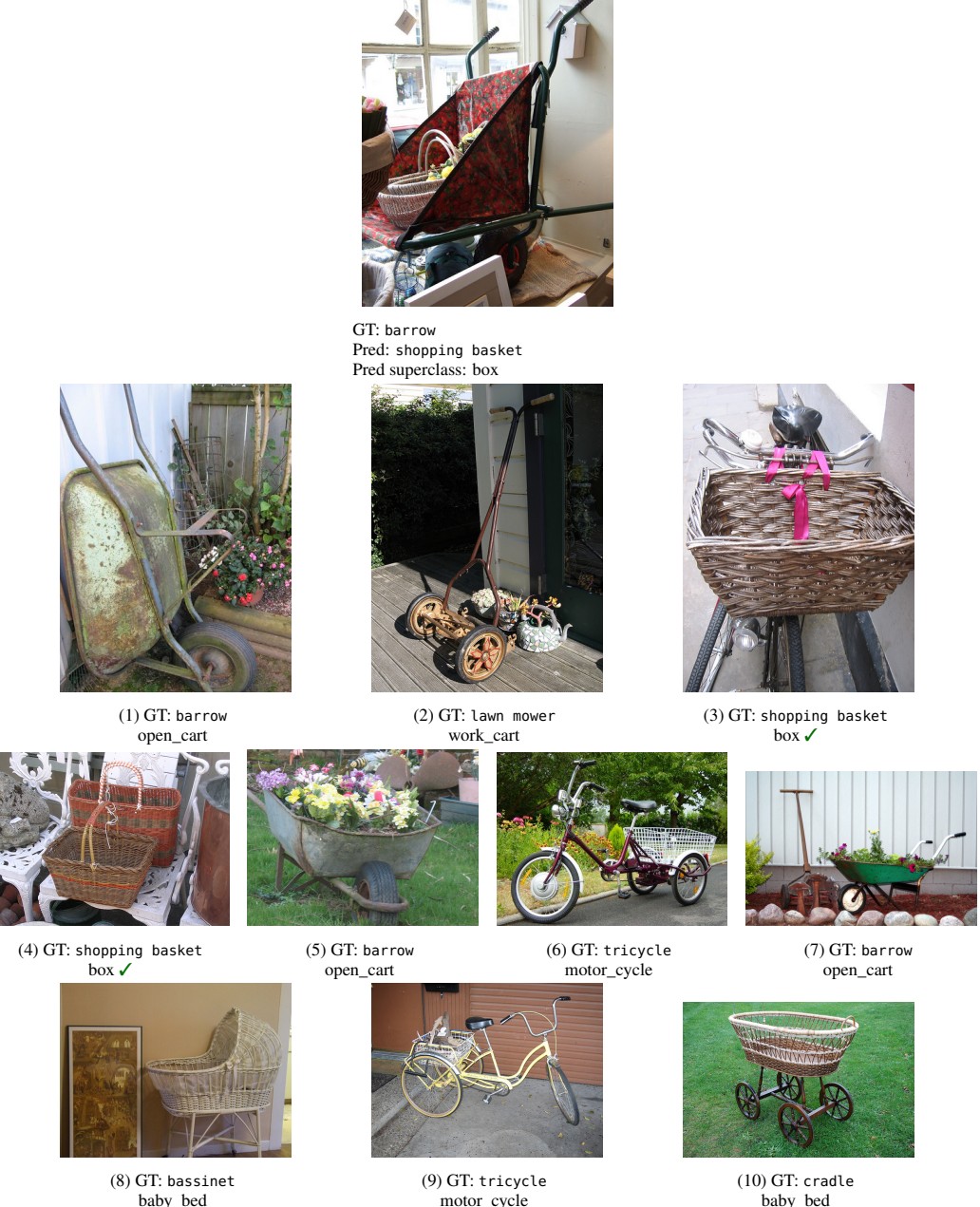

GT: barrow
Pred: shopping basket
Pred superclass: box

(1) GT: barrow
open_cart

(2) GT: lawn mower
work_cart

(3) GT: shopping basket
box ✓

(4) GT: shopping basket
box ✓

(5) GT: barrow
open_cart

(6) GT: tricycle
motor_cycle

(7) GT: barrow
open_cart

(8) GT: bassinet
baby_bed

(9) GT: tricycle
motor_cycle

(10) GT: cradle
baby_bed

Figure 34: **Fine-grained OOV rejected**. The evaluated validation sample (top) has a ground-truth IMAGENET label (GT) barrow, but the model predicts shopping basket, arguably attempting to classify the basket in the cart. The predicted label is in the "box" superclass. Next, we show the top-10 training images that are the most visually similar to the validation sample (according to CLIP similarity) together with their IMAGENET labels and the superclasses of these labels. The labels of images 3 and 4 are in the same superclass as the predicted label (box), hence we proceed with the procedure. After obtaining the in-vocabulary (IV) and OOV label proposals, as described in Sec. 3.4, the proposal with the highest probability according to CLIP is shopping cart, which is in-vocabulary, therefore, the mistake is not categorized as a fine-grained OOV error. The other most likely proposals are: shopping basket (IV), bushel basket (OOV), basket (OOV) and handbasket (OOV).

# E    Mistake Examples by Error Type

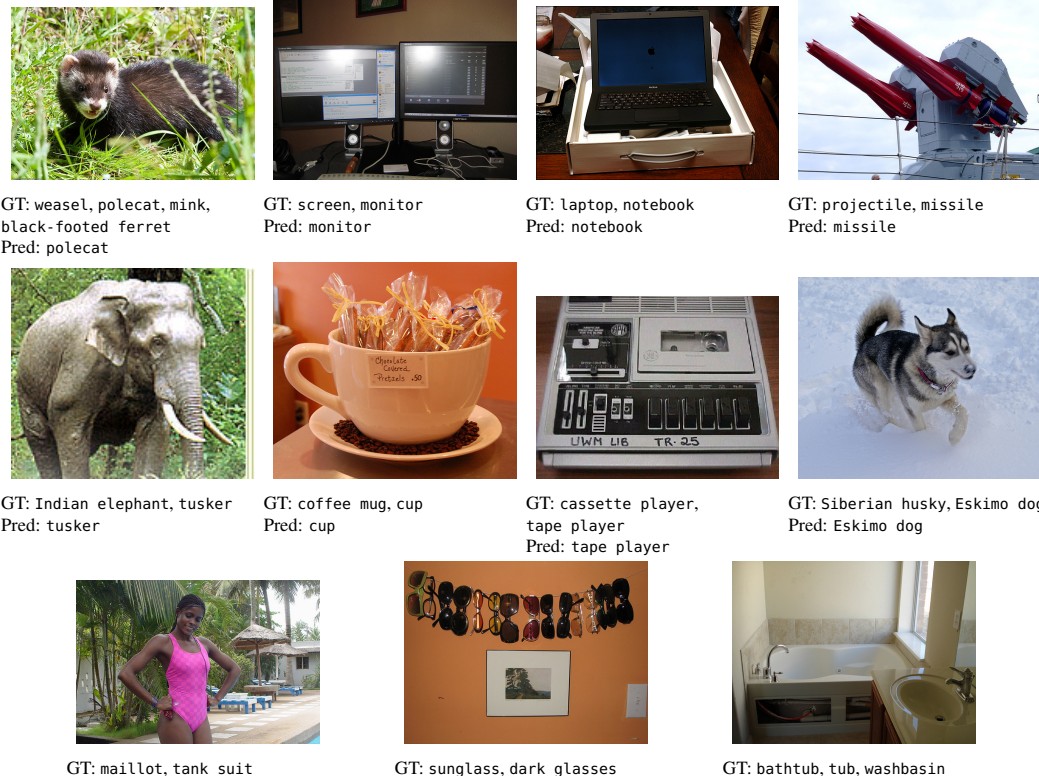

GT: weasel, polecat, mink, black-footed ferret
Pred: polecat

GT: screen, monitor
Pred: monitor

GT: laptop, notebook
Pred: notebook

GT: projectile, missile
Pred: missile

GT: Indian elephant, tusker
Pred: tusker

GT: coffee mug, cup
Pred: cup

GT: cassette player, tape player
Pred: tape player

GT: Siberian husky, Eskimo dog
Pred: Eskimo dog

GT: maillot, tank suit
Pred: tank suit

GT: sunglass, dark glasses
Pred: dark glasses

GT: bathtub, tub, washbasin
Pred: tub

Figure 35: **Class overlap**: examples of common prediction errors (Pred) due to overlapping classes. The correct (individual) multi-labels (GT) are separated by commas; the original IMAGENET label is listed first.

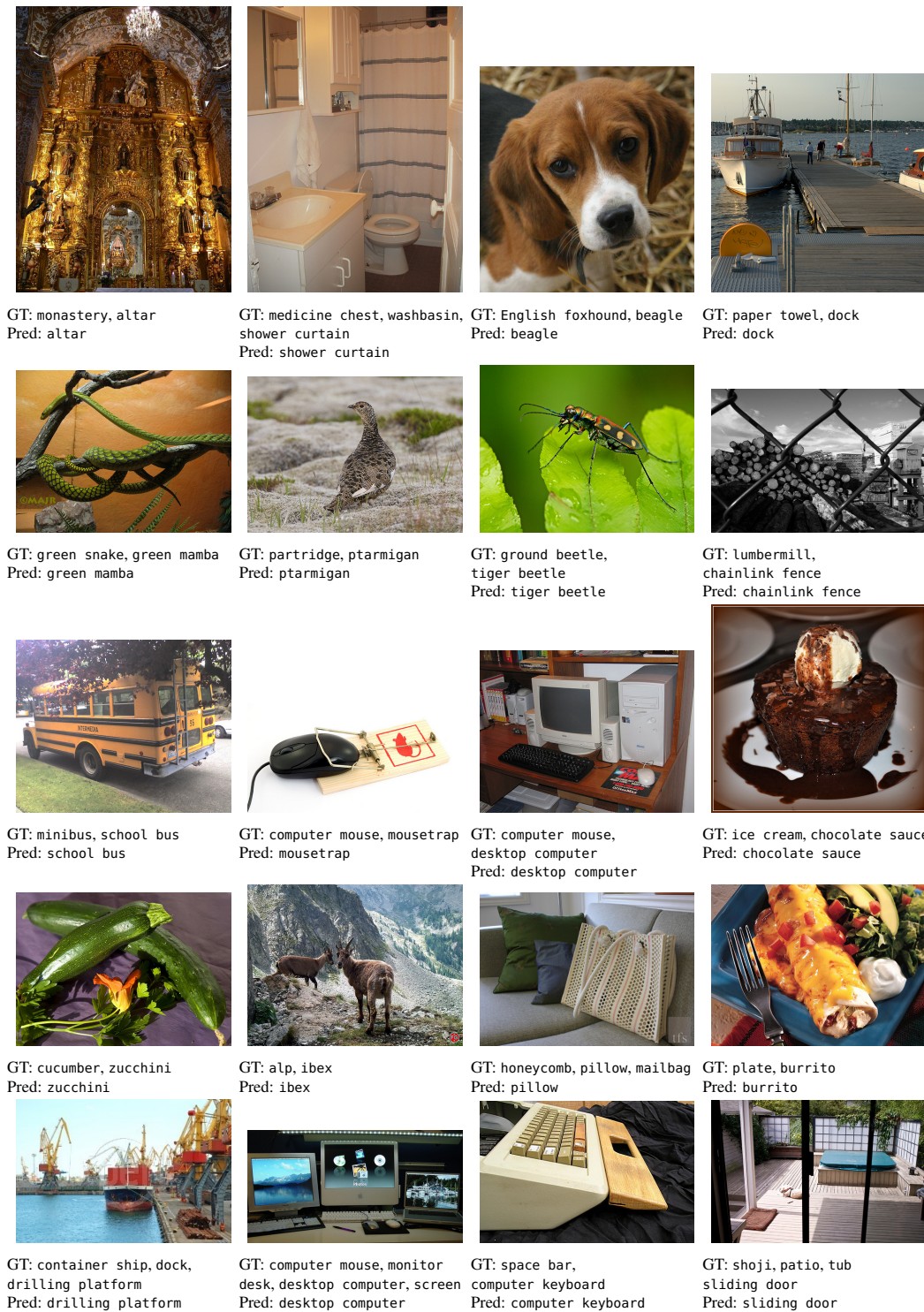

Figure 36: **Multi-label**: examples of common prediction errors (Pred) due to multi-label errors. The correct (individual) multi-labels (GT) are separated by commas; the original IMAGENET label is listed first.

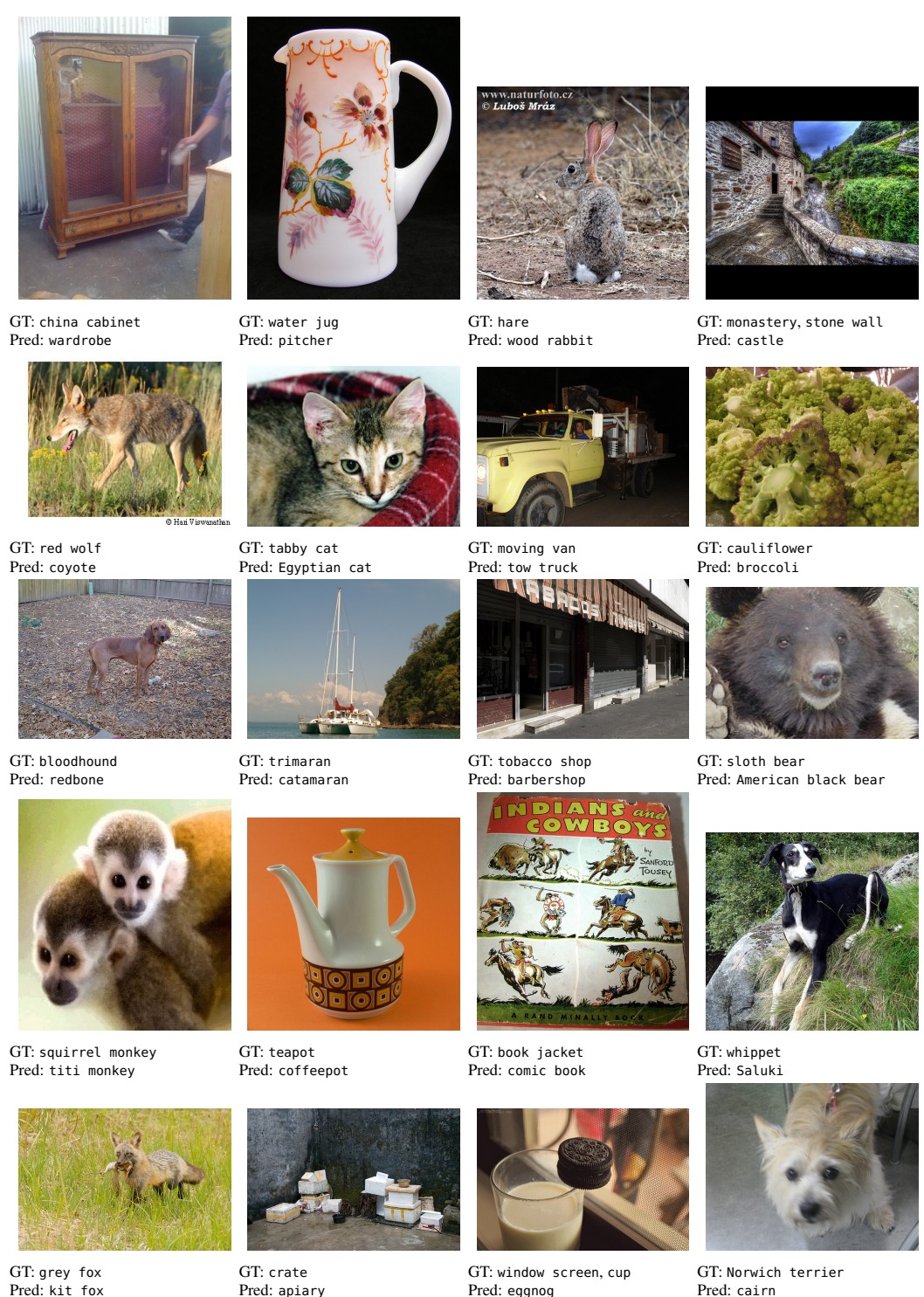

GT: china cabinet
Pred: wardrobe

GT: water jug
Pred: pitcher

GT: hare
Pred: wood rabbit

GT: monastery, stone wall
Pred: castle

GT: red wolf
Pred: coyote

GT: tabby cat
Pred: Egyptian cat

GT: moving van
Pred: tow truck

GT: cauliflower
Pred: broccoli

GT: bloodhound
Pred: redbone

GT: trimaran
Pred: catamaran

GT: tobacco shop
Pred: barbershop

GT: sloth bear
Pred: American black bear

GT: squirrel monkey
Pred: titi monkey

GT: teapot
Pred: coffeepot

GT: book jacket
Pred: comic book

GT: whippet
Pred: Saluki

GT: grey fox
Pred: kit fox

GT: crate
Pred: apiary

GT: window screen, cup
Pred: eggnog

GT: Norwich terrier
Pred: cairn

Figure 37: **Fine-grained**: examples of common prediction errors (Pred) due to fine-grained errors. The correct (individual) multi-labels (GT) are separated by commas; the original IMAGENET label is listed first.

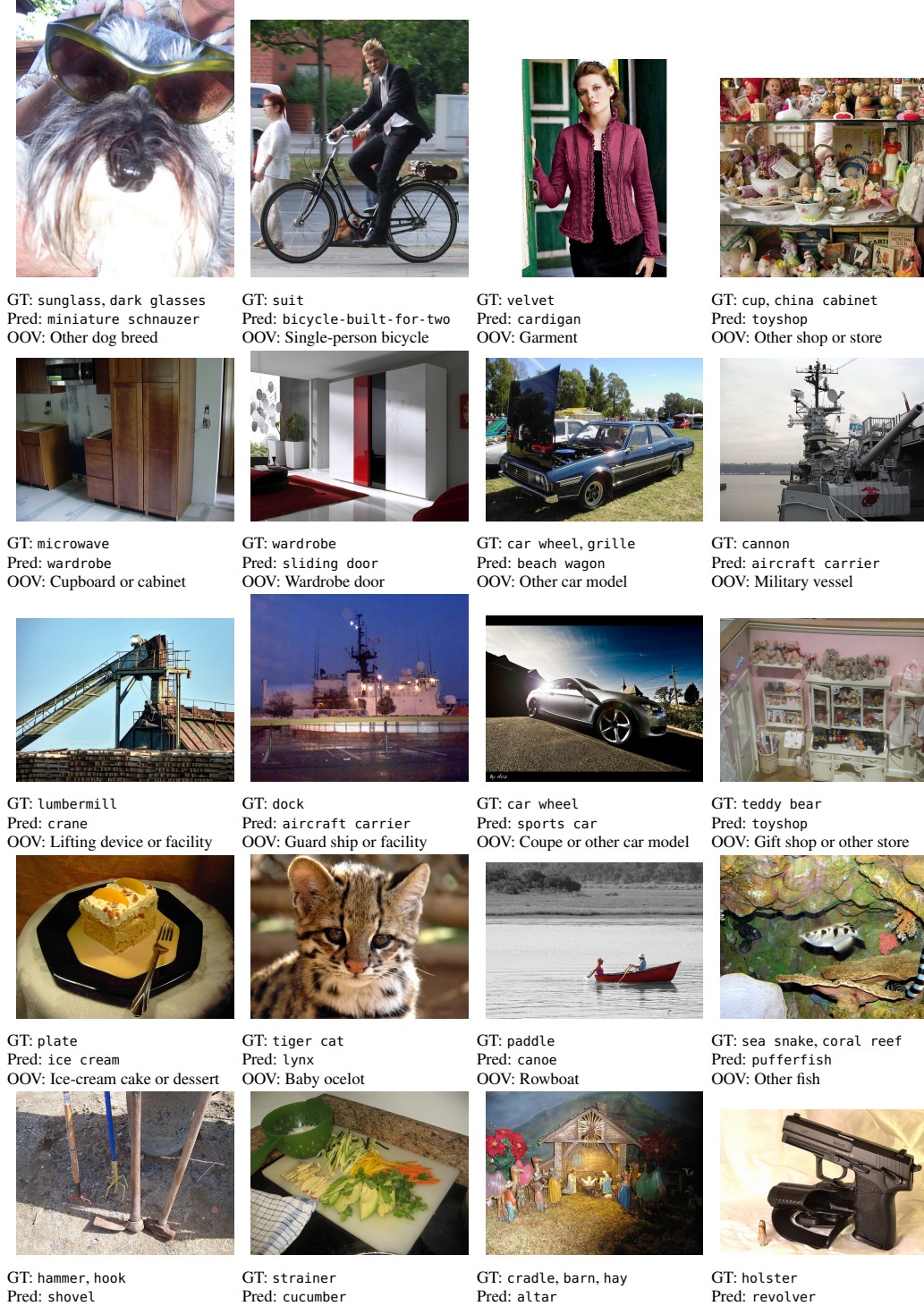

GT: sunglass, dark glasses
Pred: miniature schnauzer
OOV: Other dog breed

GT: suit
Pred: bicycle-built-for-two
OOV: Single-person bicycle

GT: velvet
Pred: cardigan
OOV: Garment

GT: cup, china cabinet
Pred: toyshop
OOV: Other shop or store

GT: microwave
Pred: wardrobe
OOV: Cupboard or cabinet

GT: wardrobe
Pred: sliding door
OOV: Wardrobe door

GT: car wheel, grille
Pred: beach wagon
OOV: Other car model

GT: cannon
Pred: aircraft carrier
OOV: Military vessel

GT: lumbermill
Pred: crane
OOV: Lifting device or facility

GT: dock
Pred: aircraft carrier
OOV: Guard ship or facility

GT: car wheel
Pred: sports car
OOV: Coupe or other car model

GT: teddy bear
Pred: toyshop
OOV: Gift shop or other store

GT: plate
Pred: ice cream
OOV: Ice-cream cake or dessert

GT: tiger cat
Pred: lynx
OOV: Baby ocelot

GT: paddle
Pred: canoe
OOV: Rowboat

GT: sea snake, coral reef
Pred: pufferfish
OOV: Other fish

GT: hammer, hook
Pred: shovel
OOV: Other tool / shovel type

GT: strainer
Pred: cucumber
OOV: Other vegetable

GT: cradle, barn, hay
Pred: altar
OOV: Altar crib

GT: holster
Pred: revolver
OOV: Semiautomatic pistol

Figure 38: **Fine-grained OOV**: examples of common prediction errors (Pred) due to fine-grained OOV errors. The correct (individual) multi-labels (GT) are separated by commas; the original IMAGENET label is listed first.

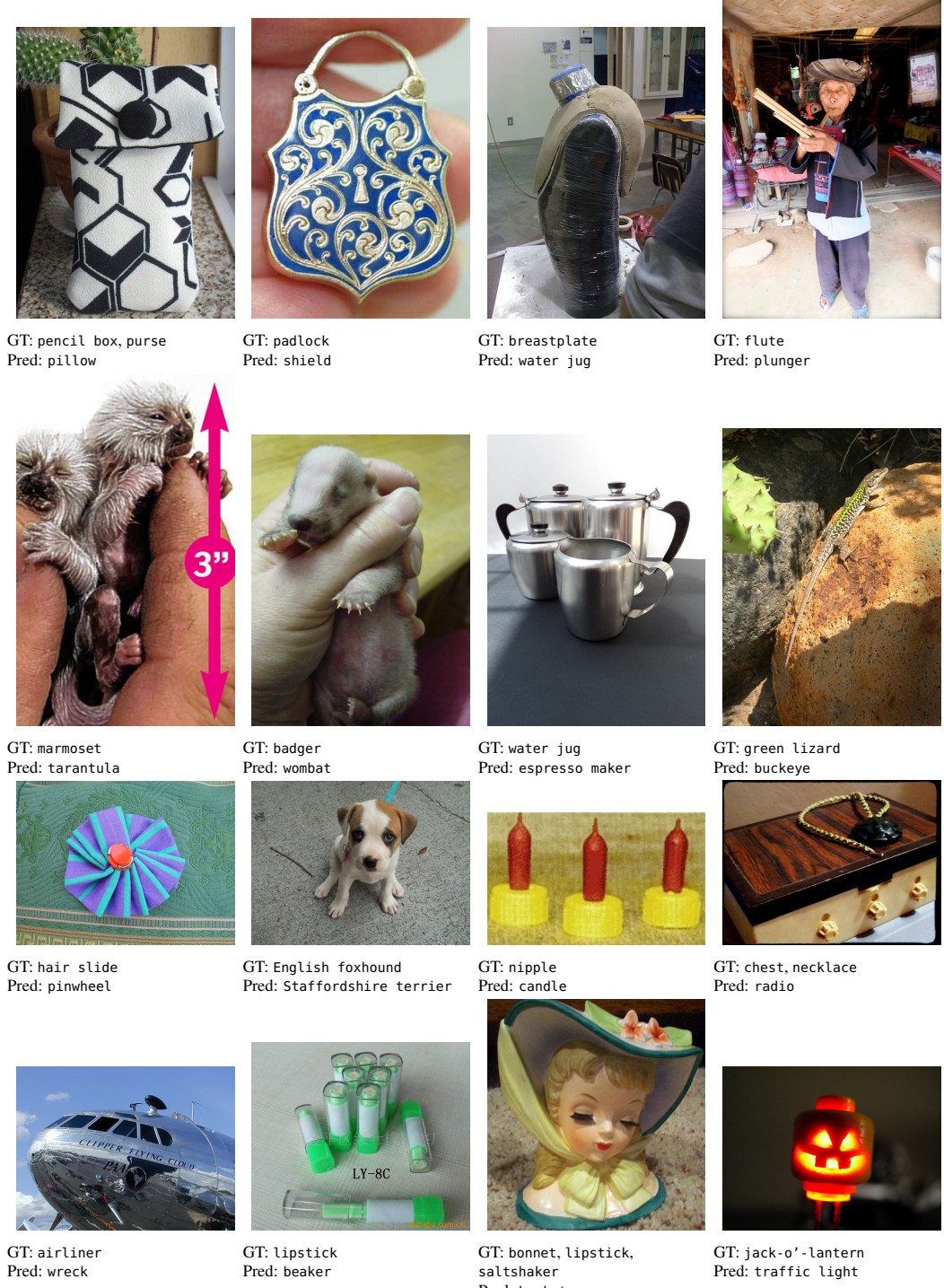

Figure 39: **Non-prototypical**: examples of common prediction errors (Pred) due to non-prototypical instances. The correct (individual) multi-labels (GT) are separated by commas; the original IMAGENET label is listed first.

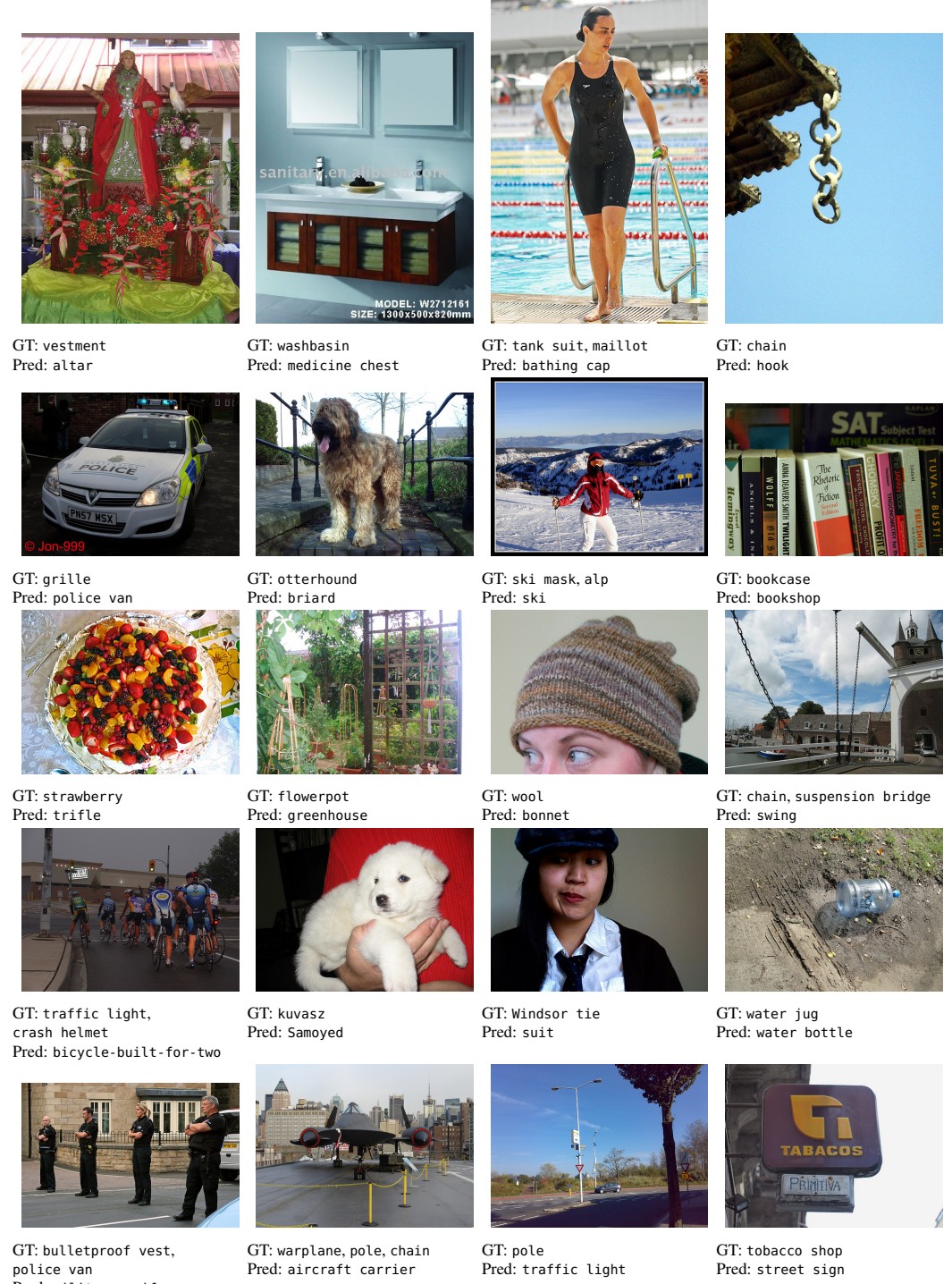

GT: vestment
Pred: altar

GT: washbasin
Pred: medicine chest

GT: tank suit, maillot
Pred: bathing cap

GT: chain
Pred: hook

GT: grille
Pred: police van

GT: otterhound
Pred: briard

GT: ski mask, alp
Pred: ski

GT: bookcase
Pred: bookshop

GT: strawberry
Pred: trifle

GT: flowerpot
Pred: greenhouse

GT: wool
Pred: bonnet

GT: chain, suspension bridge
Pred: swing

GT: traffic light,
crash helmet
Pred: bicycle-built-for-two

GT: kuvasz
Pred: Samoyed

GT: Windsor tie
Pred: suit

GT: water jug
Pred: water bottle

GT: bulletproof vest,
police van
Pred: military uniform

GT: warplane, pole, chain
Pred: aircraft carrier

GT: pole
Pred: traffic light

GT: tobacco shop
Pred: street sign

Figure 40: **Spurious correlations**: examples of common prediction errors (Pred) due to spurious correlations. The correct (individual) multi-labels (GT) are separated by commas; the original IMAGENET label is listed first.

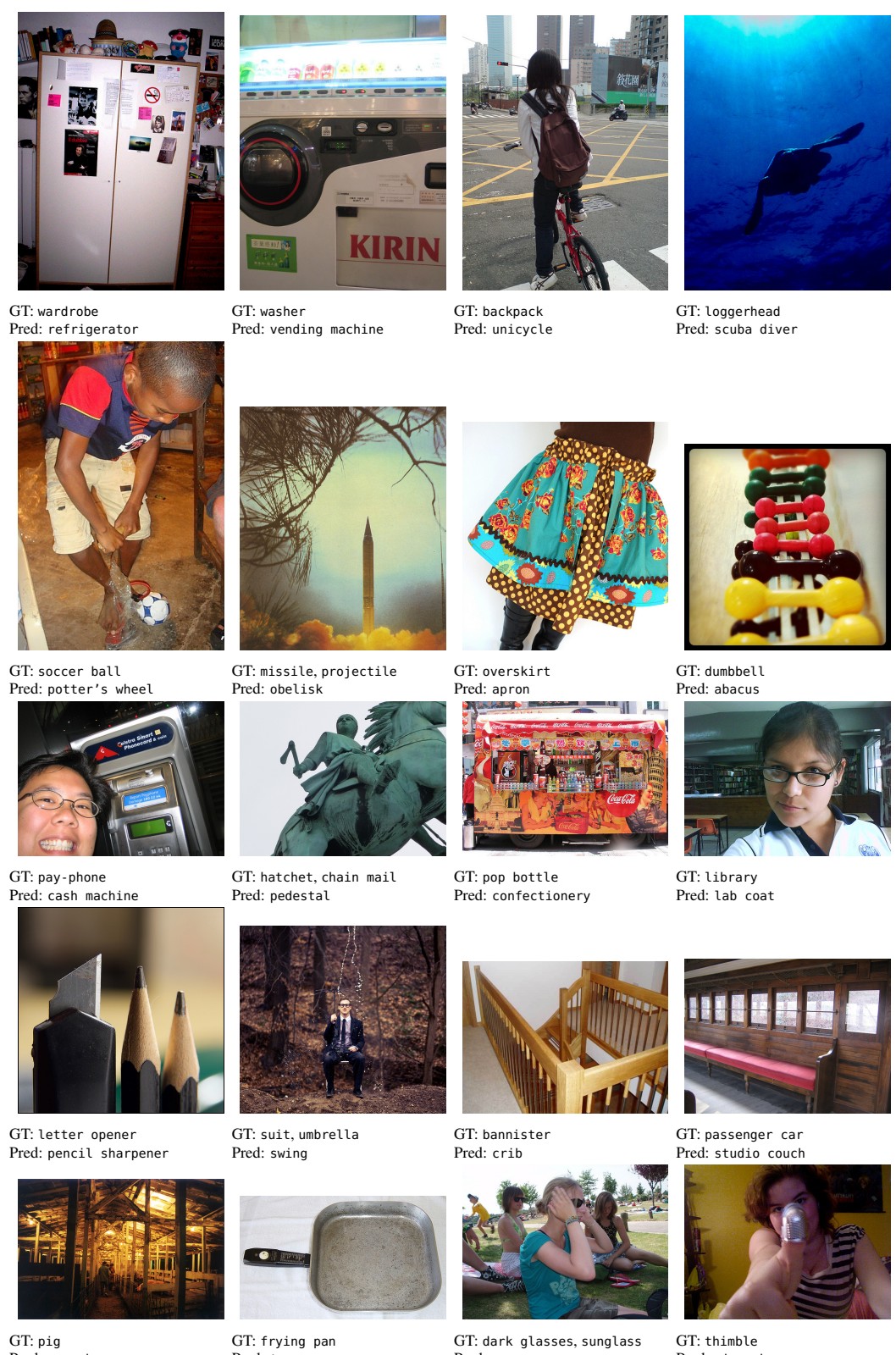

GT: wardrobe
Pred: refrigerator

GT: washer
Pred: vending machine

GT: backpack
Pred: unicycle

GT: loggerhead
Pred: scuba diver

GT: soccer ball
Pred: potter's wheel

GT: missile, projectile
Pred: obelisk

GT: overskirt
Pred: apron

GT: dumbbell
Pred: abacus

GT: pay-phone
Pred: cash machine

GT: hatchet, chain mail
Pred: pedestal

GT: pop bottle
Pred: confectionery

GT: library
Pred: lab coat

GT: letter opener
Pred: pencil sharpener

GT: suit, umbrella
Pred: swing

GT: bannister
Pred: crib

GT: passenger car
Pred: studio couch

GT: pig
Pred: greenhouse

GT: frying pan
Pred: tray

GT: dark glasses, sunglass
Pred: sunscreen

GT: thimble
Pred: microphone

Figure 41: **Model failures**: examples of common prediction errors (Pred) classified as model failures. The correct (individual) multi-labels (GT) are separated by commas; the original IMAGENET label is listed first.

# F   Experimental Setup

In this section, we provide further details on our evaluation.

We publish our code and detailed instructions on how to reproduce our results at `https://github.com/eth-sri/automated-error-analysis`.

## F.1   Computing the Multi-Label Accuracy

When computing the models' multi-label accuracies, we closely follow the recommendations by Shankar et al. (2020)[3]. In particular, we discard the set of problematic images, then compute per-class accuracies and finally average them to obtain the multi-label accuracy. We consider a model prediction to be correct if it is marked as correct or unclear in the multi-label dataset.

## F.2   Datasets and Labels

We consider the validation set of the ILSVRC-2012 subset of IMAGENET (Deng et al., 2009; Russakovsky et al., 2015), available under a non-commercial research license[4]. More concretely, we use the subset of this validation set labeled by Shankar et al. (2020) and then Vasudevan et al. (2022), with the labels[3] being available under Apache License 2.0. We further evaluate our pipeline on the ImageNet-A dataset (Hendrycks et al., 2021) available under MIT License[5].

## F.3   Summary of Evaluated Models

Table 3 contains a list of all models we considered in this study and a subset of their metadata. The models were obtained from multiple sources: Torchvision[6], `torch.hub`[7], HuggingFace[8], and `timm`[9]. They can all be automatically downloaded and evaluated using our open-sourced code. After collecting all model outputs (6 days for ImageNet and 1 day for ImageNet-A on a single GeForce RTX 2080 Ti GPU), running our error analysis pipeline on all models takes 12 to 24 hours using a single GeForce RTX 2080 Ti GPU for ImageNet and ImageNet-A respectively.

Table 3: List of evaluated models and a subset of their metadata.

| No | Model ID | Source | Arch | Dataset |
|---|---|---|---|---|
| 1. | alexnet | torch | cnn | ImageNet-1k |
| 2. | bat_resnext26ts.ch_in1k | timm | cnn | ImageNet-1k |
| 3. | convformer_b36.sail_in1k | timm | cnn | ImageNet-1k |
| 4. | convformer_b36.sail_in1k_384 | timm | cnn | ImageNet-1k |
| 5. | convformer_b36.sail_in22k_ft_in1k | timm | cnn | ImageNet-21k |
| 6. | convformer_b36.sail_in22k_ft_in1k_384 | timm | cnn | ImageNet-21k |
| 7. | convformer_m36.sail_in1k | timm | cnn | ImageNet-1k |
| 8. | convformer_m36.sail_in1k_384 | timm | cnn | ImageNet-1k |
| 9. | convformer_m36.sail_in22k_ft_in1k | timm | cnn | ImageNet-21k |
| 10. | convformer_m36.sail_in22k_ft_in1k_384 | timm | cnn | ImageNet-21k |
| 11. | convformer_s18.sail_in1k | timm | cnn | ImageNet-1k |
| 12. | convformer_s18.sail_in1k_384 | timm | cnn | ImageNet-1k |
| 13. | convformer_s18.sail_in22k_ft_in1k | timm | cnn | ImageNet-21k |
| 14. | convformer_s18.sail_in22k_ft_in1k_384 | timm | cnn | ImageNet-21k |
| 15. | convformer_s36.sail_in1k | timm | cnn | ImageNet-1k |
| 16. | convformer_s36.sail_in1k_384 | timm | cnn | ImageNet-1k |
| 17. | convformer_s36.sail_in22k_ft_in1k | timm | cnn | ImageNet-21k |

Table continues onto next page

---

[3] `https://www.tensorflow.org/datasets/catalog/imagenet2012_multilabel`

[4] Fore more details see `https://image-net.org/download`

[5] `https://github.com/hendrycks/natural-adv-examples`

[6] `https://pytorch.org/vision/stable/models.html`

[7] `https://pytorch.org/hub/`

[8] `https://huggingface.co/models`

[9] `https://github.com/huggingface/pytorch-image-models`

| No | Model ID | Source | Arch | Dataset |
|----|----------|--------|------|---------|
| | Model summaries (continued from previous page) | | | |
| 18. | convformer_s36.sail_in22k_ft_in1k_384 | timm | cnn | ImageNet-21k |
| 19. | convmixer_1024_20_ks9_p14.in1k | timm | cnn | ImageNet-1k |
| 20. | convmixer_1536_20.in1k | timm | cnn | ImageNet-1k |
| 21. | convmixer_768_32.in1k | timm | cnn | ImageNet-1k |
| 22. | convnext-base-224 | huggingface | cnn | ImageNet-1k |
| 23. | convnext-base-224-22k-1k | huggingface | cnn | ImageNet-21k |
| 24. | convnext-base-384 | huggingface | cnn | ImageNet-1k |
| 25. | convnext-base-384-22k-1k | huggingface | cnn | ImageNet-21k |
| 26. | convnext-large-224 | huggingface | cnn | ImageNet-1k |
| 27. | convnext-large-384 | huggingface | cnn | ImageNet-1k |
| 28. | convnext-large-384-22k-1k | huggingface | cnn | ImageNet-21k |
| 29. | convnext-small-224 | huggingface | cnn | ImageNet-1k |
| 30. | convnext-tiny-224 | huggingface | cnn | ImageNet-1k |
| 31. | convnext-xlarge-224-22k-1k | huggingface | cnn | ImageNet-21k |
| 32. | convnext-xlarge-384-22k-1k | huggingface | cnn | ImageNet-21k |
| 33. | convnext_atto.d2_in1k | timm | cnn | ImageNet-1k |
| 34. | convnext_atto_ols.a2_in1k | timm | cnn | ImageNet-1k |
| 35. | convnext_base | torch | cnn | ImageNet-1k |
| 36. | convnext_base.clip_laion2b_augreg_ft_in12k_in1k | timm | cnn | LAION-2B |
| 37. | convnext_base.clip_laion2b_augreg_ft_in12k_in1k_384 | timm | cnn | LAION-2B |
| 38. | convnext_base.clip_laion2b_augreg_ft_in1k | timm | cnn | LAION-2B |
| 39. | convnext_femto.d1_in1k | timm | cnn | ImageNet-1k |
| 40. | convnext_femto_ols.d1_in1k | timm | cnn | ImageNet-1k |
| 41. | convnext_large | torch | cnn | ImageNet-1k |
| 42. | convnext_large.fb_in22k_ft_in1k | timm | cnn | ImageNet-21k |
| 43. | convnext_large_mlp.clip_laion2b_augreg_ft_in1k | timm | cnn | LAION-2B |
| 44. | convnext_large_mlp.clip_laion2b_augreg_ft_in1k_384 | timm | cnn | LAION-2B |
| 45. | convnext_large_mlp.clip_laion2b_soup_ft_in12k_in1k_320 | timm | cnn | LAION-2B |
| 46. | convnext_large_mlp.clip_laion2b_soup_ft_in12k_in1k_384 | timm | cnn | LAION-2B |
| 47. | convnext_nano.d1h_in1k | timm | cnn | ImageNet-1k |
| 48. | convnext_nano_ols.d1h_in1k | timm | cnn | ImageNet-1k |
| 49. | convnext_pico.d1_in1k | timm | cnn | ImageNet-1k |
| 50. | convnext_pico_ols.d1_in1k | timm | cnn | ImageNet-1k |
| 51. | convnext_small | torch | cnn | ImageNet-1k |
| 52. | convnext_small.fb_in22k_ft_in1k | timm | cnn | ImageNet-21k |
| 53. | convnext_small.fb_in22k_ft_in1k_384 | timm | cnn | ImageNet-21k |
| 54. | convnext_tiny | torch | cnn | ImageNet-1k |
| 55. | convnext_tiny.fb_in22k_ft_in1k | timm | cnn | ImageNet-21k |
| 56. | convnext_tiny.fb_in22k_ft_in1k_384 | timm | cnn | ImageNet-21k |
| 57. | convnext_tiny_hnf.a2h_in1k | timm | cnn | ImageNet-1k |
| 58. | convnext_xxlarge.clip_laion2b_soup_ft_in1k | timm | cnn | LAION-2B |
| 59. | convnextv2_atto.fcmae_ft_in1k | timm | cnn | ImageNet-1k |
| 60. | convnextv2_base.fcmae_ft_in1k | timm | cnn | ImageNet-1k |
| 61. | convnextv2_base.fcmae_ft_in22k_in1k | timm | cnn | ImageNet-21k |
| 62. | convnextv2_base.fcmae_ft_in22k_in1k_384 | timm | cnn | ImageNet-21k |
| 63. | convnextv2_femto.fcmae_ft_in1k | timm | cnn | ImageNet-1k |
| 64. | convnextv2_huge.fcmae_ft_in1k | timm | cnn | ImageNet-1k |
| 65. | convnextv2_huge.fcmae_ft_in22k_in1k_384 | timm | cnn | ImageNet-21k |
| 66. | convnextv2_huge.fcmae_ft_in22k_in1k_512 | timm | cnn | ImageNet-21k |
| 67. | convnextv2_large.fcmae_ft_in1k | timm | cnn | ImageNet-1k |
| 68. | convnextv2_large.fcmae_ft_in22k_in1k | timm | cnn | ImageNet-21k |
| 69. | convnextv2_large.fcmae_ft_in22k_in1k_384 | timm | cnn | ImageNet-21k |
| 70. | convnextv2_nano.fcmae_ft_in1k | timm | cnn | ImageNet-1k |
| 71. | convnextv2_nano.fcmae_ft_in22k_in1k | timm | cnn | ImageNet-21k |
| 72. | convnextv2_nano.fcmae_ft_in22k_in1k_384 | timm | cnn | ImageNet-21k |
| 73. | convnextv2_pico.fcmae_ft_in1k | timm | cnn | ImageNet-1k |
| 74. | convnextv2_tiny.fcmae_ft_in1k | timm | cnn | ImageNet-1k |
| 75. | convnextv2_tiny.fcmae_ft_in22k_in1k | timm | cnn | ImageNet-21k |
| 76. | convnextv2_tiny.fcmae_ft_in22k_in1k_384 | timm | cnn | ImageNet-21k |
| | Table continues onto next page | | | |

| No | Model ID | Source | Arch | Dataset |
|----|----------|--------|------|---------|
| | Model summaries (continued from previous page) | | | |
| 77. | cs3darknet_focus_l.c2ns_in1k | timm | cnn | ImageNet-1k |
| 78. | cs3darknet_focus_m.c2ns_in1k | timm | cnn | ImageNet-1k |
| 79. | cs3darknet_l.c2ns_in1k | timm | cnn | ImageNet-1k |
| 80. | cs3darknet_m.c2ns_in1k | timm | cnn | ImageNet-1k |
| 81. | cs3darknet_x.c2ns_in1k | timm | cnn | ImageNet-1k |
| 82. | cs3sedarknet_l.c2ns_in1k | timm | cnn | ImageNet-1k |
| 83. | cs3sedarknet_x.c2ns_in1k | timm | cnn | ImageNet-1k |
| 84. | cspdarknet53.ra_in1k | timm | cnn | ImageNet-1k |
| 85. | cspresnet50.ra_in1k | timm | cnn | ImageNet-1k |
| 86. | cspresnext50.ra_in1k | timm | cnn | ImageNet-1k |
| 87. | darknet53.c2ns_in1k | timm | cnn | ImageNet-1k |
| 88. | darknetaa53.c2ns_in1k | timm | cnn | ImageNet-1k |
| 89. | densenet121 | torch | cnn | ImageNet-1k |
| 90. | densenet121.ra_in1k | timm | cnn | ImageNet-1k |
| 91. | densenet161 | torch | cnn | ImageNet-1k |
| 92. | densenet169 | torch | cnn | ImageNet-1k |
| 93. | densenet201 | torch | cnn | ImageNet-1k |
| 94. | densenetblur121d.ra_in1k | timm | cnn | ImageNet-1k |
| 95. | dla102.in1k | timm | cnn | ImageNet-1k |
| 96. | dla102x.in1k | timm | cnn | ImageNet-1k |
| 97. | dla102x2.in1k | timm | cnn | ImageNet-1k |
| 98. | dla169.in1k | timm | cnn | ImageNet-1k |
| 99. | dla34.in1k | timm | cnn | ImageNet-1k |
| 100. | dla46_c.in1k | timm | cnn | ImageNet-1k |
| 101. | dla46x_c.in1k | timm | cnn | ImageNet-1k |
| 102. | dla60.in1k | timm | cnn | ImageNet-1k |
| 103. | dla60_res2net.in1k | timm | cnn | ImageNet-1k |
| 104. | dla60_res2next.in1k | timm | cnn | ImageNet-1k |
| 105. | dla60x.in1k | timm | cnn | ImageNet-1k |
| 106. | dla60x_c.in1k | timm | cnn | ImageNet-1k |
| 107. | dm_nfnet_f0.dm_in1k | timm | cnn | ImageNet-1k |
| 108. | dm_nfnet_f1.dm_in1k | timm | cnn | ImageNet-1k |
| 109. | dm_nfnet_f2.dm_in1k | timm | cnn | ImageNet-1k |
| 110. | dm_nfnet_f3.dm_in1k | timm | cnn | ImageNet-1k |
| 111. | dm_nfnet_f4.dm_in1k | timm | cnn | ImageNet-1k |
| 112. | dm_nfnet_f5.dm_in1k | timm | cnn | ImageNet-1k |
| 113. | dm_nfnet_f6.dm_in1k | timm | cnn | ImageNet-1k |
| 114. | dpn107.mx_in1k | timm | cnn | ImageNet-1k |
| 115. | dpn131.mx_in1k | timm | cnn | ImageNet-1k |
| 116. | dpn68.mx_in1k | timm | cnn | ImageNet-1k |
| 117. | dpn68b.mx_in1k | timm | cnn | ImageNet-1k |
| 118. | dpn68b.ra_in1k | timm | cnn | ImageNet-1k |
| 119. | dpn92.mx_in1k | timm | cnn | ImageNet-1k |
| 120. | dpn98.mx_in1k | timm | cnn | ImageNet-1k |
| 121. | eca_halonext26ts.c1_in1k | timm | cnn | ImageNet-1k |
| 122. | eca_nfnet_l0.ra2_in1k | timm | cnn | ImageNet-1k |
| 123. | eca_nfnet_l1.ra2_in1k | timm | cnn | ImageNet-1k |
| 124. | eca_nfnet_l2.ra3_in1k | timm | cnn | ImageNet-1k |
| 125. | eca_resnet33ts.ra2_in1k | timm | cnn | ImageNet-1k |
| 126. | eca_resnext26ts.ch_in1k | timm | cnn | ImageNet-1k |
| 127. | ecaresnet101d.miil_in1k | timm | cnn | ImageNet-1k |
| 128. | ecaresnet101d_pruned.miil_in1k | timm | cnn | ImageNet-1k |
| 129. | ecaresnet269d.ra2_in1k | timm | cnn | ImageNet-1k |
| 130. | ecaresnet26t.ra2_in1k | timm | cnn | ImageNet-1k |
| 131. | ecaresnet50d.miil_in1k | timm | cnn | ImageNet-1k |
| 132. | ecaresnet50d_pruned.miil_in1k | timm | cnn | ImageNet-1k |
| 133. | ecaresnet50t.a1_in1k | timm | cnn | ImageNet-1k |
| 134. | ecaresnet50t.a2_in1k | timm | cnn | ImageNet-1k |
| 135. | ecaresnet50t.a3_in1k | timm | cnn | ImageNet-1k |
| | Table continues onto next page | | | |

| No | Model ID | Source | Arch | Dataset |
|---|---|---|---|---|
| 136. | ecaresnet50t.ra2_in1k | timm | cnn | ImageNet-1k |
| 137. | ecaresnetlight.miil_in1k | timm | cnn | ImageNet-1k |
| 138. | efficientnet_b0.ra_in1k | timm | cnn | ImageNet-1k |
| 139. | efficientnet_b1.ft_in1k | timm | cnn | ImageNet-1k |
| 140. | efficientnet_b1_pruned.in1k | timm | cnn | ImageNet-1k |
| 141. | efficientnet_b2.ra_in1k | timm | cnn | ImageNet-1k |
| 142. | efficientnet_b2_pruned.in1k | timm | cnn | ImageNet-1k |
| 143. | efficientnet_b3.ra2_in1k | timm | cnn | ImageNet-1k |
| 144. | efficientnet_b3_pruned.in1k | timm | cnn | ImageNet-1k |
| 145. | efficientnet_b4.ra2_in1k | timm | cnn | ImageNet-1k |
| 146. | efficientnet_el.ra_in1k | timm | cnn | ImageNet-1k |
| 147. | efficientnet_el_pruned.in1k | timm | cnn | ImageNet-1k |
| 148. | efficientnet_em.ra2_in1k | timm | cnn | ImageNet-1k |
| 149. | efficientnet_es.ra_in1k | timm | cnn | ImageNet-1k |
| 150. | efficientnet_es_pruned.in1k | timm | cnn | ImageNet-1k |
| 151. | efficientnet_lite0.ra_in1k | timm | cnn | ImageNet-1k |
| 152. | efficientnetv2_rw_m.agc_in1k | timm | cnn | ImageNet-1k |
| 153. | efficientnetv2_rw_s.ra2_in1k | timm | cnn | ImageNet-1k |
| 154. | efficientnetv2_rw_t.ra2_in1k | timm | cnn | ImageNet-1k |
| 155. | fbnetc_100.rmsp_in1k | timm | cnn | ImageNet-1k |
| 156. | fbnetv3_b.ra2_in1k | timm | cnn | ImageNet-1k |
| 157. | fbnetv3_d.ra2_in1k | timm | cnn | ImageNet-1k |
| 158. | fbnetv3_g.ra2_in1k | timm | cnn | ImageNet-1k |
| 159. | focalnet_base_lrf.ms_in1k | timm | cnn | ImageNet-1k |
| 160. | focalnet_base_srf.ms_in1k | timm | cnn | ImageNet-1k |
| 161. | focalnet_small_lrf.ms_in1k | timm | cnn | ImageNet-1k |
| 162. | focalnet_small_srf.ms_in1k | timm | cnn | ImageNet-1k |
| 163. | focalnet_tiny_lrf.ms_in1k | timm | cnn | ImageNet-1k |
| 164. | focalnet_tiny_srf.ms_in1k | timm | cnn | ImageNet-1k |
| 165. | gc_efficientnetv2_rw_t.agc_in1k | timm | cnn | ImageNet-1k |
| 166. | gcresnet33ts.ra2_in1k | timm | cnn | ImageNet-1k |
| 167. | gcresnet50t.ra2_in1k | timm | cnn | ImageNet-1k |
| 168. | gcresnext26ts.ch_in1k | timm | cnn | ImageNet-1k |
| 169. | gcresnext50ts.ch_in1k | timm | cnn | ImageNet-1k |
| 170. | googlenet | torch | cnn | ImageNet-1k |
| 171. | halonet26t.a1h_in1k | timm | cnn | ImageNet-1k |
| 172. | halonet50ts.a1h_in1k | timm | cnn | ImageNet-1k |
| 173. | hardcorenas_a.miil_green_in1k | timm | cnn | ImageNet-1k |
| 174. | hardcorenas_b.miil_green_in1k | timm | cnn | ImageNet-1k |
| 175. | hardcorenas_c.miil_green_in1k | timm | cnn | ImageNet-1k |
| 176. | hardcorenas_d.miil_green_in1k | timm | cnn | ImageNet-1k |
| 177. | hardcorenas_e.miil_green_in1k | timm | cnn | ImageNet-1k |
| 178. | hardcorenas_f.miil_green_in1k | timm | cnn | ImageNet-1k |
| 179. | hrnet_w18.ms_aug_in1k | timm | cnn | ImageNet-1k |
| 180. | hrnet_w18.ms_in1k | timm | cnn | ImageNet-1k |
| 181. | hrnet_w18_small.ms_in1k | timm | cnn | ImageNet-1k |
| 182. | hrnet_w18_small_v2.ms_in1k | timm | cnn | ImageNet-1k |
| 183. | hrnet_w18_ssld.paddle_in1k | timm | cnn | ImageNet-1k |
| 184. | hrnet_w30.ms_in1k | timm | cnn | ImageNet-1k |
| 185. | hrnet_w32.ms_in1k | timm | cnn | ImageNet-1k |
| 186. | hrnet_w40.ms_in1k | timm | cnn | ImageNet-1k |
| 187. | hrnet_w44.ms_in1k | timm | cnn | ImageNet-1k |
| 188. | hrnet_w48.ms_in1k | timm | cnn | ImageNet-1k |
| 189. | hrnet_w48_ssld.paddle_in1k | timm | cnn | ImageNet-1k |
| 190. | hrnet_w64.ms_in1k | timm | cnn | ImageNet-1k |
| 191. | inception_resnet_v2.tf_ens_adv_in1k | timm | cnn | ImageNet-1k |
| 192. | inception_resnet_v2.tf_in1k | timm | cnn | ImageNet-1k |
| 193. | inception_v3 | torch | cnn | ImageNet-1k |
| 194. | inception_v3.gluon_in1k | timm | cnn | ImageNet-1k |

| No | Model ID | Source | Arch | Dataset |
|----|----------|--------|------|---------|
| | Model summaries (continued from previous page) | | | |
| 195. | inception_v3.tf_adv_in1k | timm | cnn | ImageNet-1k |
| 196. | inception_v3.tf_in1k | timm | cnn | ImageNet-1k |
| 197. | inception_v4.tf_in1k | timm | cnn | ImageNet-1k |
| 198. | lambda_resnet26rpt_256.c1_in1k | timm | cnn | ImageNet-1k |
| 199. | lambda_resnet26t.c1_in1k | timm | cnn | ImageNet-1k |
| 200. | lambda_resnet50ts.a1h_in1k | timm | cnn | ImageNet-1k |
| 201. | lcnet_050.ra2_in1k | timm | cnn | ImageNet-1k |
| 202. | lcnet_075.ra2_in1k | timm | cnn | ImageNet-1k |
| 203. | lcnet_100.ra2_in1k | timm | cnn | ImageNet-1k |
| 204. | mixnet_l.ft_in1k | timm | cnn | ImageNet-1k |
| 205. | mixnet_m.ft_in1k | timm | cnn | ImageNet-1k |
| 206. | mixnet_s.ft_in1k | timm | cnn | ImageNet-1k |
| 207. | mixnet_xl.ra_in1k | timm | cnn | ImageNet-1k |
| 208. | mnasnet0_5 | torch | cnn | ImageNet-1k |
| 209. | mnasnet0_75 | torch | cnn | ImageNet-1k |
| 210. | mnasnet1_0 | torch | cnn | ImageNet-1k |
| 211. | mnasnet1_3 | torch | cnn | ImageNet-1k |
| 212. | mnasnet_100.rmsp_in1k | timm | cnn | ImageNet-1k |
| 213. | mnasnet_small.lamb_in1k | timm | cnn | ImageNet-1k |
| 214. | mobilenet_v2 | torch | cnn | ImageNet-1k |
| 215. | mobilenet_v3_large | torch | cnn | ImageNet-1k |
| 216. | mobilenet_v3_small | torch | cnn | ImageNet-1k |
| 217. | mobilenetv2_050.lamb_in1k | timm | cnn | ImageNet-1k |
| 218. | mobilenetv2_100.ra_in1k | timm | cnn | ImageNet-1k |
| 219. | mobilenetv2_110d.ra_in1k | timm | cnn | ImageNet-1k |
| 220. | mobilenetv2_120d.ra_in1k | timm | cnn | ImageNet-1k |
| 221. | mobilenetv2_140.ra_in1k | timm | cnn | ImageNet-1k |
| 222. | mobilenetv3_large_100.miil_in21k_ft_in1k | timm | cnn | ImageNet-21k |
| 223. | mobilenetv3_large_100.ra_in1k | timm | cnn | ImageNet-1k |
| 224. | mobilenetv3_rw.rmsp_in1k | timm | cnn | ImageNet-1k |
| 225. | mobilenetv3_small_050.lamb_in1k | timm | cnn | ImageNet-1k |
| 226. | mobilenetv3_small_075.lamb_in1k | timm | cnn | ImageNet-1k |
| 227. | mobilenetv3_small_100.lamb_in1k | timm | cnn | ImageNet-1k |
| 228. | nf_regnet_b1.ra2_in1k | timm | cnn | ImageNet-1k |
| 229. | nf_resnet50.ra2_in1k | timm | cnn | ImageNet-1k |
| 230. | nfnet_l0.ra2_in1k | timm | cnn | ImageNet-1k |
| 231. | regnet_x_16gf | torch | cnn | ImageNet-1k |
| 232. | regnet_x_1_6gf | torch | cnn | ImageNet-1k |
| 233. | regnet_x_32gf | torch | cnn | ImageNet-1k |
| 234. | regnet_x_3_2gf | torch | cnn | ImageNet-1k |
| 235. | regnet_x_400mf | torch | cnn | ImageNet-1k |
| 236. | regnet_x_800mf | torch | cnn | ImageNet-1k |
| 237. | regnet_x_8gf | torch | cnn | ImageNet-1k |
| 238. | regnet_y_128gf_swag | torch | cnn | Instagram |
| 239. | regnet_y_16gf | torch | cnn | ImageNet-1k |
| 240. | regnet_y_16gf_swag | torch | cnn | Instagram |
| 241. | regnet_y_1_6gf | torch | cnn | ImageNet-1k |
| 242. | regnet_y_32gf | torch | cnn | ImageNet-1k |
| 243. | regnet_y_32gf_swag | torch | cnn | Instagram |
| 244. | regnet_y_3_2gf | torch | cnn | ImageNet-1k |
| 245. | regnet_y_400mf | torch | cnn | ImageNet-1k |
| 246. | regnet_y_800mf | torch | cnn | ImageNet-1k |
| 247. | regnet_y_8gf | torch | cnn | ImageNet-1k |
| 248. | regnetv_040.ra3_in1k | timm | cnn | ImageNet-1k |
| 249. | regnetv_064.ra3_in1k | timm | cnn | ImageNet-1k |
| 250. | regnetx_002.pycls_in1k | timm | cnn | ImageNet-1k |
| 251. | regnetx_004.pycls_in1k | timm | cnn | ImageNet-1k |
| 252. | regnetx_006.pycls_in1k | timm | cnn | ImageNet-1k |
| 253. | regnetx_008.pycls_in1k | timm | cnn | ImageNet-1k |
| | Table continues onto next page | | | |

| No | Model ID | Source | Arch | Dataset |
|---|---|---|---|---|
| | Model summaries (continued from previous page) | | | |
| 254. | regnetx_016.pycls_in1k | timm | cnn | ImageNet-1k |
| 255. | regnetx_032.pycls_in1k | timm | cnn | ImageNet-1k |
| 256. | regnetx_040.pycls_in1k | timm | cnn | ImageNet-1k |
| 257. | regnetx_064.pycls_in1k | timm | cnn | ImageNet-1k |
| 258. | regnetx_080.pycls_in1k | timm | cnn | ImageNet-1k |
| 259. | regnetx_120.pycls_in1k | timm | cnn | ImageNet-1k |
| 260. | regnetx_160.pycls_in1k | timm | cnn | ImageNet-1k |
| 261. | regnetx_320.pycls_in1k | timm | cnn | ImageNet-1k |
| 262. | regnety_002.pycls_in1k | timm | cnn | ImageNet-1k |
| 263. | regnety_004.pycls_in1k | timm | cnn | ImageNet-1k |
| 264. | regnety_006.pycls_in1k | timm | cnn | ImageNet-1k |
| 265. | regnety_008.pycls_in1k | timm | cnn | ImageNet-1k |
| 266. | regnety_016.pycls_in1k | timm | cnn | ImageNet-1k |
| 267. | regnety_032.pycls_in1k | timm | cnn | ImageNet-1k |
| 268. | regnety_032.ra_in1k | timm | cnn | ImageNet-1k |
| 269. | regnety_040.pycls_in1k | timm | cnn | ImageNet-1k |
| 270. | regnety_040.ra3_in1k | timm | cnn | ImageNet-1k |
| 271. | regnety_064.pycls_in1k | timm | cnn | ImageNet-1k |
| 272. | regnety_064.ra3_in1k | timm | cnn | ImageNet-1k |
| 273. | regnety_080.pycls_in1k | timm | cnn | ImageNet-1k |
| 274. | regnety_080.ra3_in1k | timm | cnn | ImageNet-1k |
| 275. | regnety_120.pycls_in1k | timm | cnn | ImageNet-1k |
| 276. | regnety_1280.seer_ft_in1k | timm | cnn | RII[10] |
| 277. | regnety_160.deit_in1k | timm | cnn | ImageNet-1k |
| 278. | regnety_160.pycls_in1k | timm | cnn | ImageNet-1k |
| 279. | regnety_2560.seer_ft_in1k | timm | cnn | RII[10] |
| 280. | regnety_320.pycls_in1k | timm | cnn | ImageNet-1k |
| 281. | regnety_320.seer_ft_in1k | timm | cnn | RII[10] |
| 282. | regnety_640.seer_ft_in1k | timm | cnn | RII[10] |
| 283. | regnetz_040.ra3_in1k | timm | cnn | ImageNet-1k |
| 284. | regnetz_040_h.ra3_in1k | timm | cnn | ImageNet-1k |
| 285. | regnetz_b16.ra3_in1k | timm | cnn | ImageNet-1k |
| 286. | regnetz_c16.ra3_in1k | timm | cnn | ImageNet-1k |
| 287. | regnetz_c16_evos.ch_in1k | timm | cnn | ImageNet-1k |
| 288. | regnetz_d32.ra3_in1k | timm | cnn | ImageNet-1k |
| 289. | regnetz_d8.ra3_in1k | timm | cnn | ImageNet-1k |
| 290. | regnetz_d8_evos.ch_in1k | timm | cnn | ImageNet-1k |
| 291. | regnetz_e8.ra3_in1k | timm | cnn | ImageNet-1k |
| 292. | repvgg_a2.rvgg_in1k | timm | cnn | ImageNet-1k |
| 293. | repvgg_b0.rvgg_in1k | timm | cnn | ImageNet-1k |
| 294. | repvgg_b1.rvgg_in1k | timm | cnn | ImageNet-1k |
| 295. | repvgg_b1g4.rvgg_in1k | timm | cnn | ImageNet-1k |
| 296. | repvgg_b2.rvgg_in1k | timm | cnn | ImageNet-1k |
| 297. | repvgg_b2g4.rvgg_in1k | timm | cnn | ImageNet-1k |
| 298. | repvgg_b3.rvgg_in1k | timm | cnn | ImageNet-1k |
| 299. | repvgg_b3g4.rvgg_in1k | timm | cnn | ImageNet-1k |
| 300. | res2net101_26w_4s.in1k | timm | cnn | ImageNet-1k |
| 301. | res2net101d.in1k | timm | cnn | ImageNet-1k |
| 302. | res2net50_14w_8s.in1k | timm | cnn | ImageNet-1k |
| 303. | res2net50_26w_4s.in1k | timm | cnn | ImageNet-1k |
| 304. | res2net50_26w_6s.in1k | timm | cnn | ImageNet-1k |
| 305. | res2net50_26w_8s.in1k | timm | cnn | ImageNet-1k |
| 306. | res2net50_48w_2s.in1k | timm | cnn | ImageNet-1k |
| 307. | res2net50d.in1k | timm | cnn | ImageNet-1k |
| 308. | res2next50.in1k | timm | cnn | ImageNet-1k |
| 309. | resnest101e.in1k | timm | cnn | ImageNet-1k |
| 310. | resnest14d.gluon_in1k | timm | cnn | ImageNet-1k |
| | Table continues onto next page | | | |

---

[10]RandomInternetImages

| No | Model ID | Source | Arch | Dataset |
|---|---|---|---|---|
| 311. | resnest200e.in1k | timm | cnn | ImageNet-1k |
| 312. | resnest269e.in1k | timm | cnn | ImageNet-1k |
| 313. | resnest26d.gluon_in1k | timm | cnn | ImageNet-1k |
| 314. | resnest50d.in1k | timm | cnn | ImageNet-1k |
| 315. | resnest50d_1s4x24d.in1k | timm | cnn | ImageNet-1k |
| 316. | resnest50d_4s2x40d.in1k | timm | cnn | ImageNet-1k |
| 317. | resnet101 | torch | cnn | ImageNet-1k |
| 318. | resnet101.a1_in1k | timm | cnn | ImageNet-1k |
| 319. | resnet101.a1h_in1k | timm | cnn | ImageNet-1k |
| 320. | resnet101.a2_in1k | timm | cnn | ImageNet-1k |
| 321. | resnet101.a3_in1k | timm | cnn | ImageNet-1k |
| 322. | resnet101.gluon_in1k | timm | cnn | ImageNet-1k |
| 323. | resnet101c.gluon_in1k | timm | cnn | ImageNet-1k |
| 324. | resnet101d.gluon_in1k | timm | cnn | ImageNet-1k |
| 325. | resnet101d.ra2_in1k | timm | cnn | ImageNet-1k |
| 326. | resnet101s.gluon_in1k | timm | cnn | ImageNet-1k |
| 327. | resnet10t.c3_in1k | timm | cnn | ImageNet-1k |
| 328. | resnet14t.c3_in1k | timm | cnn | ImageNet-1k |
| 329. | resnet152 | torch | cnn | ImageNet-1k |
| 330. | resnet152.a1_in1k | timm | cnn | ImageNet-1k |
| 331. | resnet152.a1h_in1k | timm | cnn | ImageNet-1k |
| 332. | resnet152.a2_in1k | timm | cnn | ImageNet-1k |
| 333. | resnet152.a3_in1k | timm | cnn | ImageNet-1k |
| 334. | resnet152.gluon_in1k | timm | cnn | ImageNet-1k |
| 335. | resnet152c.gluon_in1k | timm | cnn | ImageNet-1k |
| 336. | resnet152d.gluon_in1k | timm | cnn | ImageNet-1k |
| 337. | resnet152d.ra2_in1k | timm | cnn | ImageNet-1k |
| 338. | resnet152s.gluon_in1k | timm | cnn | ImageNet-1k |
| 339. | resnet18 | torch | cnn | ImageNet-1k |
| 340. | resnet18.a1_in1k | timm | cnn | ImageNet-1k |
| 341. | resnet18.a2_in1k | timm | cnn | ImageNet-1k |
| 342. | resnet18.a3_in1k | timm | cnn | ImageNet-1k |
| 343. | resnet18.gluon_in1k | timm | cnn | ImageNet-1k |
| 344. | resnet18_ssl | torch | cnn | Flickr YFCC |
| 345. | resnet18_swsl | torch | cnn | Instagram |
| 346. | resnet18d.ra2_in1k | timm | cnn | ImageNet-1k |
| 347. | resnet200d.ra2_in1k | timm | cnn | ImageNet-1k |
| 348. | resnet26.bt_in1k | timm | cnn | ImageNet-1k |
| 349. | resnet26d.bt_in1k | timm | cnn | ImageNet-1k |
| 350. | resnet26t.ra2_in1k | timm | cnn | ImageNet-1k |
| 351. | resnet32ts.ra2_in1k | timm | cnn | ImageNet-1k |
| 352. | resnet33ts.ra2_in1k | timm | cnn | ImageNet-1k |
| 353. | resnet34 | torch | cnn | ImageNet-1k |
| 354. | resnet34.a1_in1k | timm | cnn | ImageNet-1k |
| 355. | resnet34.a2_in1k | timm | cnn | ImageNet-1k |
| 356. | resnet34.a3_in1k | timm | cnn | ImageNet-1k |
| 357. | resnet34.bt_in1k | timm | cnn | ImageNet-1k |
| 358. | resnet34.gluon_in1k | timm | cnn | ImageNet-1k |
| 359. | resnet34d.ra2_in1k | timm | cnn | ImageNet-1k |
| 360. | resnet50 | torch | cnn | ImageNet-1k |
| 361. | resnet50.a1_in1k | timm | cnn | ImageNet-1k |
| 362. | resnet50.a1h_in1k | timm | cnn | ImageNet-1k |
| 363. | resnet50.a2_in1k | timm | cnn | ImageNet-1k |
| 364. | resnet50.a3_in1k | timm | cnn | ImageNet-1k |
| 365. | resnet50.am_in1k | timm | cnn | ImageNet-1k |
| 366. | resnet50.b1k_in1k | timm | cnn | ImageNet-1k |
| 367. | resnet50.b2k_in1k | timm | cnn | ImageNet-1k |
| 368. | resnet50.bt_in1k | timm | cnn | ImageNet-1k |
| 369. | resnet50.c1_in1k | timm | cnn | ImageNet-1k |

| No | Model ID | Source | Arch | Dataset |
|----|----------|--------|------|---------|
| 370. | resnet50.c2_in1k | timm | cnn | ImageNet-1k |
| 371. | resnet50.d_in1k | timm | cnn | ImageNet-1k |
| 372. | resnet50.gluon_in1k | timm | cnn | ImageNet-1k |
| 373. | resnet50.ra_in1k | timm | cnn | ImageNet-1k |
| 374. | resnet50.ram_in1k | timm | cnn | ImageNet-1k |
| 375. | resnet50_gn.a1h_in1k | timm | cnn | ImageNet-1k |
| 376. | resnet50_ssl | torch | cnn | Flickr YFCC |
| 377. | resnet50_swsl | torch | cnn | Instagram |
| 378. | resnet50c.gluon_in1k | timm | cnn | ImageNet-1k |
| 379. | resnet50d.a1_in1k | timm | cnn | ImageNet-1k |
| 380. | resnet50d.a2_in1k | timm | cnn | ImageNet-1k |
| 381. | resnet50d.a3_in1k | timm | cnn | ImageNet-1k |
| 382. | resnet50d.gluon_in1k | timm | cnn | ImageNet-1k |
| 383. | resnet50d.ra2_in1k | timm | cnn | ImageNet-1k |
| 384. | resnet50s.gluon_in1k | timm | cnn | ImageNet-1k |
| 385. | resnet51q.ra2_in1k | timm | cnn | ImageNet-1k |
| 386. | resnet61q.ra2_in1k | timm | cnn | ImageNet-1k |
| 387. | resnetaa50.a1h_in1k | timm | cnn | ImageNet-1k |
| 388. | resnetblur50.bt_in1k | timm | cnn | ImageNet-1k |
| 389. | resnetrs101.tf_in1k | timm | cnn | ImageNet-1k |
| 390. | resnetrs152.tf_in1k | timm | cnn | ImageNet-1k |
| 391. | resnetrs200.tf_in1k | timm | cnn | ImageNet-1k |
| 392. | resnetrs270.tf_in1k | timm | cnn | ImageNet-1k |
| 393. | resnetrs350.tf_in1k | timm | cnn | ImageNet-1k |
| 394. | resnetrs420.tf_in1k | timm | cnn | ImageNet-1k |
| 395. | resnetrs50.tf_in1k | timm | cnn | ImageNet-1k |
| 396. | resnetv2_101.a1h_in1k | timm | cnn | ImageNet-1k |
| 397. | resnetv2_101x1_bit.goog_in21k_ft_in1k | timm | cnn | ImageNet-21k |
| 398. | resnetv2_101x3_bit.goog_in21k_ft_in1k | timm | cnn | ImageNet-21k |
| 399. | resnetv2_152x2_bit.goog_in21k_ft_in1k | timm | cnn | ImageNet-21k |
| 400. | resnetv2_152x2_bit.goog_teacher_in21k_ft_in1k | timm | cnn | ImageNet-21k |
| 401. | resnetv2_152x2_bit.goog_teacher_in21k_ft_in1k_384 | timm | cnn | ImageNet-21k |
| 402. | resnetv2_152x4_bit.goog_in21k_ft_in1k | timm | cnn | ImageNet-21k |
| 403. | resnetv2_50.a1h_in1k | timm | cnn | ImageNet-1k |
| 404. | resnetv2_50d_evos.ah_in1k | timm | cnn | ImageNet-1k |
| 405. | resnetv2_50d_gn.ah_in1k | timm | cnn | ImageNet-1k |
| 406. | resnetv2_50x1_bit.goog_distilled_in1k | timm | cnn | ImageNet-21k |
| 407. | resnetv2_50x1_bit.goog_in21k_ft_in1k | timm | cnn | ImageNet-21k |
| 408. | resnetv2_50x3_bit.goog_in21k_ft_in1k | timm | cnn | ImageNet-21k |
| 409. | resnext101_32x16d_ssl | torch | cnn | Flickr YFCC |
| 410. | resnext101_32x16d_swsl | torch | cnn | Instagram |
| 411. | resnext101_32x16d_wsl | torch | cnn | Instagram |
| 412. | resnext101_32x32d_wsl | torch | cnn | Instagram |
| 413. | resnext101_32x48d_wsl | torch | cnn | Instagram |
| 414. | resnext101_32x4d.gluon_in1k | timm | cnn | ImageNet-1k |
| 415. | resnext101_32x4d_ssl | torch | cnn | Flickr YFCC |
| 416. | resnext101_32x4d_swsl | torch | cnn | Instagram |
| 417. | resnext101_32x8d | torch | cnn | ImageNet-1k |
| 418. | resnext101_32x8d_ssl | torch | cnn | Flickr YFCC |
| 419. | resnext101_32x8d_swsl | torch | cnn | Instagram |
| 420. | resnext101_32x8d_wsl | torch | cnn | Instagram |
| 421. | resnext101_64x4d | torch | cnn | ImageNet-1k |
| 422. | resnext101_64x4d.c1_in1k | timm | cnn | ImageNet-1k |
| 423. | resnext101_64x4d.gluon_in1k | timm | cnn | ImageNet-1k |
| 424. | resnext26ts.ra2_in1k | timm | cnn | ImageNet-1k |
| 425. | resnext50_32x4d | torch | cnn | ImageNet-1k |
| 426. | resnext50_32x4d.a1_in1k | timm | cnn | ImageNet-1k |
| 427. | resnext50_32x4d.a1h_in1k | timm | cnn | ImageNet-1k |
| 428. | resnext50_32x4d.a2_in1k | timm | cnn | ImageNet-1k |

| No | Model ID | Source | Arch | Dataset |
|---|---|---|---|---|
| | Model summaries (continued from previous page) | | | |
| 429. | resnext50_32x4d.a3_in1k | timm | cnn | ImageNet-1k |
| 430. | resnext50_32x4d.gluon_in1k | timm | cnn | ImageNet-1k |
| 431. | resnext50_32x4d.ra_in1k | timm | cnn | ImageNet-1k |
| 432. | resnext50_32x4d_ssl | torch | cnn | Flickr YFCC |
| 433. | resnext50_32x4d_swsl | torch | cnn | Instagram |
| 434. | resnext50d_32x4d.bt_in1k | timm | cnn | ImageNet-1k |
| 435. | rexnet_100.nav_in1k | timm | cnn | ImageNet-1k |
| 436. | rexnet_130.nav_in1k | timm | cnn | ImageNet-1k |
| 437. | rexnet_150.nav_in1k | timm | cnn | ImageNet-1k |
| 438. | rexnet_200.nav_in1k | timm | cnn | ImageNet-1k |
| 439. | rexnet_300.nav_in1k | timm | cnn | ImageNet-1k |
| 440. | sehalonet33ts.ra2_in1k | timm | cnn | ImageNet-1k |
| 441. | semnasnet_075.rmsp_in1k | timm | cnn | ImageNet-1k |
| 442. | semnasnet_100.rmsp_in1k | timm | cnn | ImageNet-1k |
| 443. | seresnet152d.ra2_in1k | timm | cnn | ImageNet-1k |
| 444. | seresnet33ts.ra2_in1k | timm | cnn | ImageNet-1k |
| 445. | seresnet50.a1_in1k | timm | cnn | ImageNet-1k |
| 446. | seresnet50.a2_in1k | timm | cnn | ImageNet-1k |
| 447. | seresnet50.a3_in1k | timm | cnn | ImageNet-1k |
| 448. | seresnet50.ra2_in1k | timm | cnn | ImageNet-1k |
| 449. | seresnext101_32x4d.gluon_in1k | timm | cnn | ImageNet-1k |
| 450. | seresnext101_32x8d.ah_in1k | timm | cnn | ImageNet-1k |
| 451. | seresnext101_64x4d.gluon_in1k | timm | cnn | ImageNet-1k |
| 452. | seresnext101d_32x8d.ah_in1k | timm | cnn | ImageNet-1k |
| 453. | seresnext26d_32x4d.bt_in1k | timm | cnn | ImageNet-1k |
| 454. | seresnext26t_32x4d.bt_in1k | timm | cnn | ImageNet-1k |
| 455. | seresnext26ts.ch_in1k | timm | cnn | ImageNet-1k |
| 456. | seresnext50_32x4d.gluon_in1k | timm | cnn | ImageNet-1k |
| 457. | seresnext50_32x4d.racm_in1k | timm | cnn | ImageNet-1k |
| 458. | seresnextaa101d_32x8d.ah_in1k | timm | cnn | ImageNet-1k |
| 459. | shufflenet_v2_x0_5 | torch | cnn | ImageNet-1k |
| 460. | shufflenet_v2_x1_0 | torch | cnn | ImageNet-1k |
| 461. | shufflenet_v2_x1_5 | torch | cnn | ImageNet-1k |
| 462. | shufflenet_v2_x2_0 | torch | cnn | ImageNet-1k |
| 463. | skresnet18.ra_in1k | timm | cnn | ImageNet-1k |
| 464. | skresnet34.ra_in1k | timm | cnn | ImageNet-1k |
| 465. | skresnext50_32x4d.ra_in1k | timm | cnn | ImageNet-1k |
| 466. | squeezenet1_0 | torch | cnn | ImageNet-1k |
| 467. | squeezenet1_1 | torch | cnn | ImageNet-1k |
| 468. | tf_efficientnet_b0.aa_in1k | timm | cnn | ImageNet-1k |
| 469. | tf_efficientnet_b0.ap_in1k | timm | cnn | ImageNet-1k |
| 470. | tf_efficientnet_b0.in1k | timm | cnn | ImageNet-1k |
| 471. | tf_efficientnet_b0.ns_jft_in1k | timm | cnn | JFT |
| 472. | tf_efficientnet_b1.aa_in1k | timm | cnn | ImageNet-1k |
| 473. | tf_efficientnet_b1.ap_in1k | timm | cnn | ImageNet-1k |
| 474. | tf_efficientnet_b1.in1k | timm | cnn | ImageNet-1k |
| 475. | tf_efficientnet_b1.ns_jft_in1k | timm | cnn | JFT |
| 476. | tf_efficientnet_b2.aa_in1k | timm | cnn | ImageNet-1k |
| 477. | tf_efficientnet_b2.ap_in1k | timm | cnn | ImageNet-1k |
| 478. | tf_efficientnet_b2.in1k | timm | cnn | ImageNet-1k |
| 479. | tf_efficientnet_b2.ns_jft_in1k | timm | cnn | JFT |
| 480. | tf_efficientnet_b3.aa_in1k | timm | cnn | ImageNet-1k |
| 481. | tf_efficientnet_b3.ap_in1k | timm | cnn | ImageNet-1k |
| 482. | tf_efficientnet_b3.in1k | timm | cnn | ImageNet-1k |
| 483. | tf_efficientnet_b3.ns_jft_in1k | timm | cnn | JFT |
| 484. | tf_efficientnet_b4.aa_in1k | timm | cnn | ImageNet-1k |
| 485. | tf_efficientnet_b4.ap_in1k | timm | cnn | ImageNet-1k |
| 486. | tf_efficientnet_b4.in1k | timm | cnn | ImageNet-1k |
| 487. | tf_efficientnet_b4.ns_jft_in1k | timm | cnn | JFT |

| No | Model ID | Source | Arch | Dataset |
|---|---|---|---|---|
| | Model summaries (continued from previous page) | | | |
| 488. | tf_efficientnet_b5.aa_in1k | timm | cnn | ImageNet-1k |
| 489. | tf_efficientnet_b5.ap_in1k | timm | cnn | ImageNet-1k |
| 490. | tf_efficientnet_b5.in1k | timm | cnn | ImageNet-1k |
| 491. | tf_efficientnet_b5.ns_jft_in1k | timm | cnn | JFT |
| 492. | tf_efficientnet_b5.ra_in1k | timm | cnn | ImageNet-1k |
| 493. | tf_efficientnet_b6.aa_in1k | timm | cnn | ImageNet-1k |
| 494. | tf_efficientnet_b6.ap_in1k | timm | cnn | ImageNet-1k |
| 495. | tf_efficientnet_b6.ns_jft_in1k | timm | cnn | JFT |
| 496. | tf_efficientnet_b7.aa_in1k | timm | cnn | ImageNet-1k |
| 497. | tf_efficientnet_b7.ap_in1k | timm | cnn | ImageNet-1k |
| 498. | tf_efficientnet_b7.ns_jft_in1k | timm | cnn | JFT |
| 499. | tf_efficientnet_b7.ra_in1k | timm | cnn | ImageNet-1k |
| 500. | tf_efficientnet_b8.ap_in1k | timm | cnn | ImageNet-1k |
| 501. | tf_efficientnet_b8.ra_in1k | timm | cnn | ImageNet-1k |
| 502. | tf_efficientnet_cc_b0_4e.in1k | timm | cnn | ImageNet-1k |
| 503. | tf_efficientnet_cc_b0_8e.in1k | timm | cnn | ImageNet-1k |
| 504. | tf_efficientnet_cc_b1_8e.in1k | timm | cnn | ImageNet-1k |
| 505. | tf_efficientnet_el.in1k | timm | cnn | ImageNet-1k |
| 506. | tf_efficientnet_em.in1k | timm | cnn | ImageNet-1k |
| 507. | tf_efficientnet_es.in1k | timm | cnn | ImageNet-1k |
| 508. | tf_efficientnet_l2.ns_jft_in1k | timm | cnn | JFT |
| 509. | tf_efficientnet_l2.ns_jft_in1k_475 | timm | cnn | JFT |
| 510. | tf_efficientnet_lite0.in1k | timm | cnn | ImageNet-1k |
| 511. | tf_efficientnet_lite1.in1k | timm | cnn | ImageNet-1k |
| 512. | tf_efficientnet_lite2.in1k | timm | cnn | ImageNet-1k |
| 513. | tf_efficientnet_lite3.in1k | timm | cnn | ImageNet-1k |
| 514. | tf_efficientnet_lite4.in1k | timm | cnn | ImageNet-1k |
| 515. | tf_efficientnetv2_b0.in1k | timm | cnn | ImageNet-1k |
| 516. | tf_efficientnetv2_b1.in1k | timm | cnn | ImageNet-1k |
| 517. | tf_efficientnetv2_b2.in1k | timm | cnn | ImageNet-1k |
| 518. | tf_efficientnetv2_b3.in1k | timm | cnn | ImageNet-1k |
| 519. | tf_efficientnetv2_b3.in21k_ft_in1k | timm | cnn | ImageNet-21k |
| 520. | tf_efficientnetv2_l.in1k | timm | cnn | ImageNet-1k |
| 521. | tf_efficientnetv2_l.in21k_ft_in1k | timm | cnn | ImageNet-21k |
| 522. | tf_efficientnetv2_m.in1k | timm | cnn | ImageNet-1k |
| 523. | tf_efficientnetv2_m.in21k_ft_in1k | timm | cnn | ImageNet-21k |
| 524. | tf_efficientnetv2_s.in1k | timm | cnn | ImageNet-1k |
| 525. | tf_efficientnetv2_s.in21k_ft_in1k | timm | cnn | ImageNet-21k |
| 526. | tf_efficientnetv2_xl.in21k_ft_in1k | timm | cnn | ImageNet-21k |
| 527. | tf_mixnet_l.in1k | timm | cnn | ImageNet-1k |
| 528. | tf_mixnet_m.in1k | timm | cnn | ImageNet-1k |
| 529. | tf_mixnet_s.in1k | timm | cnn | ImageNet-1k |
| 530. | tf_mobilenetv3_large_075.in1k | timm | cnn | ImageNet-1k |
| 531. | tf_mobilenetv3_large_100.in1k | timm | cnn | ImageNet-1k |
| 532. | tf_mobilenetv3_large_minimal_100.in1k | timm | cnn | ImageNet-1k |
| 533. | tf_mobilenetv3_small_075.in1k | timm | cnn | ImageNet-1k |
| 534. | tf_mobilenetv3_small_100.in1k | timm | cnn | ImageNet-1k |
| 535. | tf_mobilenetv3_small_minimal_100.in1k | timm | cnn | ImageNet-1k |
| 536. | tinynet_a.in1k | timm | cnn | ImageNet-1k |
| 537. | tinynet_b.in1k | timm | cnn | ImageNet-1k |
| 538. | tinynet_c.in1k | timm | cnn | ImageNet-1k |
| 539. | tinynet_d.in1k | timm | cnn | ImageNet-1k |
| 540. | tinynet_e.in1k | timm | cnn | ImageNet-1k |
| 541. | tresnet_l.miil_in1k | timm | cnn | ImageNet-1k |
| 542. | tresnet_l.miil_in1k_448 | timm | cnn | ImageNet-1k |
| 543. | tresnet_m.miil_in1k | timm | cnn | ImageNet-1k |
| 544. | tresnet_m.miil_in1k_448 | timm | cnn | ImageNet-1k |
| 545. | tresnet_m.miil_in21k_ft_in1k | timm | cnn | ImageNet-21k |
| 546. | tresnet_v2_l.miil_in21k_ft_in1k | timm | cnn | ImageNet-21k |
| | Table continues onto next page | | | |

| No | Model ID | Source | Arch | Dataset |
|---|---|---|---|---|
| | Model summaries (continued from previous page) | | | |
| 547. | tresnet_xl.miil_in1k | timm | cnn | ImageNet-1k |
| 548. | tresnet_xl.miil_in1k_448 | timm | cnn | ImageNet-1k |
| 549. | vgg11 | torch | cnn | ImageNet-1k |
| 550. | vgg11_bn | torch | cnn | ImageNet-1k |
| 551. | vgg13 | torch | cnn | ImageNet-1k |
| 552. | vgg13_bn | torch | cnn | ImageNet-1k |
| 553. | vgg16 | torch | cnn | ImageNet-1k |
| 554. | vgg16_bn | torch | cnn | ImageNet-1k |
| 555. | vgg19 | torch | cnn | ImageNet-1k |
| 556. | vgg19_bn | torch | cnn | ImageNet-1k |
| 557. | wide_resnet101_2 | torch | cnn | ImageNet-1k |
| 558. | wide_resnet50_2 | torch | cnn | ImageNet-1k |
| 559. | wide_resnet50_2.racm_in1k | timm | cnn | ImageNet-1k |
| 560. | xception41.tf_in1k | timm | cnn | ImageNet-1k |
| 561. | xception41p.ra3_in1k | timm | cnn | ImageNet-1k |
| 562. | xception65.ra3_in1k | timm | cnn | ImageNet-1k |
| 563. | xception65.tf_in1k | timm | cnn | ImageNet-1k |
| 564. | xception65p.ra3_in1k | timm | cnn | ImageNet-1k |
| 565. | xception71.tf_in1k | timm | cnn | ImageNet-1k |
| 566. | caformer_b36.sail_in1k | timm | hybrid | ImageNet-1k |
| 567. | caformer_b36.sail_in1k_384 | timm | hybrid | ImageNet-1k |
| 568. | caformer_b36.sail_in22k_ft_in1k | timm | hybrid | ImageNet-21k |
| 569. | caformer_b36.sail_in22k_ft_in1k_384 | timm | hybrid | ImageNet-21k |
| 570. | caformer_m36.sail_in1k | timm | hybrid | ImageNet-1k |
| 571. | caformer_m36.sail_in1k_384 | timm | hybrid | ImageNet-1k |
| 572. | caformer_m36.sail_in22k_ft_in1k | timm | hybrid | ImageNet-21k |
| 573. | caformer_m36.sail_in22k_ft_in1k_384 | timm | hybrid | ImageNet-21k |
| 574. | caformer_s18.sail_in1k | timm | hybrid | ImageNet-1k |
| 575. | caformer_s18.sail_in1k_384 | timm | hybrid | ImageNet-1k |
| 576. | caformer_s18.sail_in22k_ft_in1k | timm | hybrid | ImageNet-21k |
| 577. | caformer_s18.sail_in22k_ft_in1k_384 | timm | hybrid | ImageNet-21k |
| 578. | caformer_s36.sail_in1k | timm | hybrid | ImageNet-1k |
| 579. | caformer_s36.sail_in1k_384 | timm | hybrid | ImageNet-1k |
| 580. | caformer_s36.sail_in22k_ft_in1k | timm | hybrid | ImageNet-21k |
| 581. | caformer_s36.sail_in22k_ft_in1k_384 | timm | hybrid | ImageNet-21k |
| 582. | coatnet_0_rw_224.sw_in1k | timm | hybrid | ImageNet-1k |
| 583. | coatnet_1_rw_224.sw_in1k | timm | hybrid | ImageNet-1k |
| 584. | coatnet_bn_0_rw_224.sw_in1k | timm | hybrid | ImageNet-1k |
| 585. | coatnet_nano_rw_224.sw_in1k | timm | hybrid | ImageNet-1k |
| 586. | coatnet_rmlp_1_rw_224.sw_in1k | timm | hybrid | ImageNet-1k |
| 587. | coatnet_rmlp_2_rw_224.sw_in1k | timm | hybrid | ImageNet-1k |
| 588. | coatnet_rmlp_nano_rw_224.sw_in1k | timm | hybrid | ImageNet-1k |
| 589. | coatnext_nano_rw_224.sw_in1k | timm | hybrid | ImageNet-1k |
| 590. | edgenext_base.in21k_ft_in1k | timm | hybrid | ImageNet-21k |
| 591. | edgenext_base.usi_in1k | timm | hybrid | ImageNet-1k |
| 592. | edgenext_small.usi_in1k | timm | hybrid | ImageNet-1k |
| 593. | edgenext_small_rw.sw_in1k | timm | hybrid | ImageNet-1k |
| 594. | edgenext_x_small.in1k | timm | hybrid | ImageNet-1k |
| 595. | edgenext_xx_small.in1k | timm | hybrid | ImageNet-1k |
| 596. | efficientformer_l1.snap_dist_in1k | timm | hybrid | ImageNet-1k |
| 597. | efficientformer_l3.snap_dist_in1k | timm | hybrid | ImageNet-1k |
| 598. | efficientformer_l7.snap_dist_in1k | timm | hybrid | ImageNet-1k |
| 599. | efficientformerv2_l.snap_dist_in1k | timm | hybrid | ImageNet-1k |
| 600. | efficientformerv2_s0.snap_dist_in1k | timm | hybrid | ImageNet-1k |
| 601. | efficientformerv2_s1.snap_dist_in1k | timm | hybrid | ImageNet-1k |
| 602. | efficientformerv2_s2.snap_dist_in1k | timm | hybrid | ImageNet-1k |
| 603. | levit_128.fb_dist_in1k | timm | hybrid | ImageNet-1k |
| 604. | levit_128s.fb_dist_in1k | timm | hybrid | ImageNet-1k |
| 605. | levit_192.fb_dist_in1k | timm | hybrid | ImageNet-1k |
| | Table continues onto next page | | | |

| No | Model ID | Source | Arch | Dataset |
|---|---|---|---|---|
| | Model summaries (continued from previous page) | | | |
| 606. | levit_256.fb_dist_in1k | timm | hybrid | ImageNet-1k |
| 607. | levit_384.fb_dist_in1k | timm | hybrid | ImageNet-1k |
| 608. | levit_conv_128.fb_dist_in1k | timm | hybrid | ImageNet-1k |
| 609. | levit_conv_128s.fb_dist_in1k | timm | hybrid | ImageNet-1k |
| 610. | levit_conv_192.fb_dist_in1k | timm | hybrid | ImageNet-1k |
| 611. | levit_conv_256.fb_dist_in1k | timm | hybrid | ImageNet-1k |
| 612. | levit_conv_384.fb_dist_in1k | timm | hybrid | ImageNet-1k |
| 613. | maxvit_base_tf_224.in1k | timm | hybrid | ImageNet-1k |
| 614. | maxvit_base_tf_384.in1k | timm | hybrid | ImageNet-1k |
| 615. | maxvit_base_tf_384.in21k_ft_in1k | timm | hybrid | ImageNet-21k |
| 616. | maxvit_base_tf_512.in1k | timm | hybrid | ImageNet-1k |
| 617. | maxvit_base_tf_512.in21k_ft_in1k | timm | hybrid | ImageNet-21k |
| 618. | maxvit_large_tf_224.in1k | timm | hybrid | ImageNet-1k |
| 619. | maxvit_large_tf_384.in1k | timm | hybrid | ImageNet-1k |
| 620. | maxvit_large_tf_384.in21k_ft_in1k | timm | hybrid | ImageNet-21k |
| 621. | maxvit_large_tf_512.in1k | timm | hybrid | ImageNet-1k |
| 622. | maxvit_large_tf_512.in21k_ft_in1k | timm | hybrid | ImageNet-21k |
| 623. | maxvit_nano_rw_256.sw_in1k | timm | hybrid | ImageNet-1k |
| 624. | maxvit_rmlp_nano_rw_256.sw_in1k | timm | hybrid | ImageNet-1k |
| 625. | maxvit_rmlp_pico_rw_256.sw_in1k | timm | hybrid | ImageNet-1k |
| 626. | maxvit_rmlp_small_rw_224.sw_in1k | timm | hybrid | ImageNet-1k |
| 627. | maxvit_rmlp_tiny_rw_256.sw_in1k | timm | hybrid | ImageNet-1k |
| 628. | maxvit_small_tf_224.in1k | timm | hybrid | ImageNet-1k |
| 629. | maxvit_small_tf_384.in1k | timm | hybrid | ImageNet-1k |
| 630. | maxvit_small_tf_512.in1k | timm | hybrid | ImageNet-1k |
| 631. | maxvit_tiny_rw_224.sw_in1k | timm | hybrid | ImageNet-1k |
| 632. | maxvit_tiny_tf_224.in1k | timm | hybrid | ImageNet-1k |
| 633. | maxvit_tiny_tf_384.in1k | timm | hybrid | ImageNet-1k |
| 634. | maxvit_tiny_tf_512.in1k | timm | hybrid | ImageNet-1k |
| 635. | maxvit_xlarge_tf_384.in21k_ft_in1k | timm | hybrid | ImageNet-21k |
| 636. | maxvit_xlarge_tf_512.in21k_ft_in1k | timm | hybrid | ImageNet-21k |
| 637. | maxxvit_rmlp_nano_rw_256.sw_in1k | timm | hybrid | ImageNet-1k |
| 638. | maxxvit_rmlp_small_rw_256.sw_in1k | timm | hybrid | ImageNet-1k |
| 639. | maxxvitv2_nano_rw_256.sw_in1k | timm | hybrid | ImageNet-1k |
| 640. | mobilevit_s.cvnets_in1k | timm | hybrid | ImageNet-1k |
| 641. | mobilevit_xs.cvnets_in1k | timm | hybrid | ImageNet-1k |
| 642. | mobilevit_xxs.cvnets_in1k | timm | hybrid | ImageNet-1k |
| 643. | mobilevitv2_050.cvnets_in1k | timm | hybrid | ImageNet-1k |
| 644. | mobilevitv2_075.cvnets_in1k | timm | hybrid | ImageNet-1k |
| 645. | mobilevitv2_100.cvnets_in1k | timm | hybrid | ImageNet-1k |
| 646. | mobilevitv2_125.cvnets_in1k | timm | hybrid | ImageNet-1k |
| 647. | mobilevitv2_150.cvnets_in1k | timm | hybrid | ImageNet-1k |
| 648. | mobilevitv2_150.cvnets_in22k_ft_in1k | timm | hybrid | ImageNet-21k |
| 649. | mobilevitv2_150.cvnets_in22k_ft_in1k_384 | timm | hybrid | ImageNet-21k |
| 650. | mobilevitv2_175.cvnets_in1k | timm | hybrid | ImageNet-1k |
| 651. | mobilevitv2_175.cvnets_in22k_ft_in1k | timm | hybrid | ImageNet-21k |
| 652. | mobilevitv2_175.cvnets_in22k_ft_in1k_384 | timm | hybrid | ImageNet-21k |
| 653. | mobilevitv2_200.cvnets_in1k | timm | hybrid | ImageNet-1k |
| 654. | mobilevitv2_200.cvnets_in22k_ft_in1k | timm | hybrid | ImageNet-21k |
| 655. | mobilevitv2_200.cvnets_in22k_ft_in1k_384 | timm | hybrid | ImageNet-21k |
| 656. | pvt_v2_b0.in1k | timm | hybrid | ImageNet-1k |
| 657. | pvt_v2_b1.in1k | timm | hybrid | ImageNet-1k |
| 658. | pvt_v2_b2.in1k | timm | hybrid | ImageNet-1k |
| 659. | pvt_v2_b2_li.in1k | timm | hybrid | ImageNet-1k |
| 660. | pvt_v2_b3.in1k | timm | hybrid | ImageNet-1k |
| 661. | pvt_v2_b4.in1k | timm | hybrid | ImageNet-1k |
| 662. | pvt_v2_b5.in1k | timm | hybrid | ImageNet-1k |
| 663. | gmixer_24_224.ra3_in1k | timm | mlp | ImageNet-1k |
| 664. | gmlp_s16_224.ra3_in1k | timm | mlp | ImageNet-1k |
| | Table continues onto next page | | | |

| No | Model ID | Source | Arch | Dataset |
|----|----------|--------|------|---------|
| 665. | mixer_b16_224.goog_in21k_ft_in1k | timm | mlp | ImageNet-21k |
| 666. | mixer_b16_224.miil_in21k_ft_in1k | timm | mlp | ImageNet-21k |
| 667. | mixer_l16_224.goog_in21k_ft_in1k | timm | mlp | ImageNet-21k |
| 668. | poolformer_m36.sail_in1k | timm | mlp | ImageNet-1k |
| 669. | poolformer_m48.sail_in1k | timm | mlp | ImageNet-1k |
| 670. | poolformer_s12.sail_in1k | timm | mlp | ImageNet-1k |
| 671. | poolformer_s24.sail_in1k | timm | mlp | ImageNet-1k |
| 672. | poolformer_s36.sail_in1k | timm | mlp | ImageNet-1k |
| 673. | poolformerv2_m36.sail_in1k | timm | mlp | ImageNet-1k |
| 674. | poolformerv2_m48.sail_in1k | timm | mlp | ImageNet-1k |
| 675. | poolformerv2_s12.sail_in1k | timm | mlp | ImageNet-1k |
| 676. | poolformerv2_s24.sail_in1k | timm | mlp | ImageNet-1k |
| 677. | poolformerv2_s36.sail_in1k | timm | mlp | ImageNet-1k |
| 678. | resmlp_12_224.fb_distilled_in1k | timm | mlp | ImageNet-1k |
| 679. | resmlp_12_224.fb_in1k | timm | mlp | ImageNet-1k |
| 680. | resmlp_24_224.fb_distilled_in1k | timm | mlp | ImageNet-1k |
| 681. | resmlp_24_224.fb_in1k | timm | mlp | ImageNet-1k |
| 682. | resmlp_36_224.fb_distilled_in1k | timm | mlp | ImageNet-1k |
| 683. | resmlp_36_224.fb_in1k | timm | mlp | ImageNet-1k |
| 684. | resmlp_big_24_224.fb_distilled_in1k | timm | mlp | ImageNet-1k |
| 685. | resmlp_big_24_224.fb_in1k | timm | mlp | ImageNet-1k |
| 686. | resmlp_big_24_224.fb_in22k_ft_in1k | timm | mlp | ImageNet-21k |
| 687. | beit_base_patch16_224.in22k_ft_in22k_in1k | timm | vit | ImageNet-21k |
| 688. | beit_base_patch16_384.in22k_ft_in22k_in1k | timm | vit | ImageNet-21k |
| 689. | beit_large_patch16_224.in22k_ft_in22k_in1k | timm | vit | ImageNet-21k |
| 690. | beit_large_patch16_384.in22k_ft_in22k_in1k | timm | vit | ImageNet-21k |
| 691. | beit_large_patch16_512.in22k_ft_in22k_in1k | timm | vit | ImageNet-21k |
| 692. | beitv2_base_patch16_224.in1k_ft_in1k | timm | vit | ImageNet-1k |
| 693. | beitv2_base_patch16_224.in1k_ft_in22k_in1k | timm | vit | ImageNet-21k |
| 694. | beitv2_large_patch16_224.in1k_ft_in1k | timm | vit | ImageNet-1k |
| 695. | beitv2_large_patch16_224.in1k_ft_in22k_in1k | timm | vit | ImageNet-21k |
| 696. | cait_m36_384.fb_dist_in1k | timm | vit | ImageNet-1k |
| 697. | cait_m48_448.fb_dist_in1k | timm | vit | ImageNet-1k |
| 698. | cait_s24_224.fb_dist_in1k | timm | vit | ImageNet-1k |
| 699. | cait_s24_384.fb_dist_in1k | timm | vit | ImageNet-1k |
| 700. | cait_s36_384.fb_dist_in1k | timm | vit | ImageNet-1k |
| 701. | cait_xs24_384.fb_dist_in1k | timm | vit | ImageNet-1k |
| 702. | cait_xxs24_224.fb_dist_in1k | timm | vit | ImageNet-1k |
| 703. | cait_xxs24_384.fb_dist_in1k | timm | vit | ImageNet-1k |
| 704. | cait_xxs36_224.fb_dist_in1k | timm | vit | ImageNet-1k |
| 705. | cait_xxs36_384.fb_dist_in1k | timm | vit | ImageNet-1k |
| 706. | coat_lite_medium.in1k | timm | vit | ImageNet-1k |
| 707. | coat_lite_medium_384.in1k | timm | vit | ImageNet-1k |
| 708. | coat_lite_mini.in1k | timm | vit | ImageNet-1k |
| 709. | coat_lite_small.in1k | timm | vit | ImageNet-1k |
| 710. | coat_lite_tiny.in1k | timm | vit | ImageNet-1k |
| 711. | coat_mini.in1k | timm | vit | ImageNet-1k |
| 712. | coat_small.in1k | timm | vit | ImageNet-1k |
| 713. | coat_tiny.in1k | timm | vit | ImageNet-1k |
| 714. | convit_base.fb_in1k | timm | vit | ImageNet-1k |
| 715. | convit_small.fb_in1k | timm | vit | ImageNet-1k |
| 716. | convit_tiny.fb_in1k | timm | vit | ImageNet-1k |
| 717. | crossvit_15_240.in1k | timm | vit | ImageNet-1k |
| 718. | crossvit_15_dagger_240.in1k | timm | vit | ImageNet-1k |
| 719. | crossvit_15_dagger_408.in1k | timm | vit | ImageNet-1k |
| 720. | crossvit_18_240.in1k | timm | vit | ImageNet-1k |
| 721. | crossvit_18_dagger_240.in1k | timm | vit | ImageNet-1k |
| 722. | crossvit_18_dagger_408.in1k | timm | vit | ImageNet-1k |
| 723. | crossvit_9_240.in1k | timm | vit | ImageNet-1k |

| No | Model ID | Source | Arch | Dataset |
|---|---|---|---|---|
| | Model summaries (continued from previous page) | | | |
| 724. | crossvit_9_dagger_240.in1k | timm | vit | ImageNet-1k |
| 725. | crossvit_base_240.in1k | timm | vit | ImageNet-1k |
| 726. | crossvit_small_240.in1k | timm | vit | ImageNet-1k |
| 727. | crossvit_tiny_240.in1k | timm | vit | ImageNet-1k |
| 728. | davit_base.msft_in1k | timm | vit | ImageNet-1k |
| 729. | davit_small.msft_in1k | timm | vit | ImageNet-1k |
| 730. | davit_tiny.msft_in1k | timm | vit | ImageNet-1k |
| 731. | deit3_base_patch16_224.fb_in1k | timm | vit | ImageNet-1k |
| 732. | deit3_base_patch16_224.fb_in22k_ft_in1k | timm | vit | ImageNet-21k |
| 733. | deit3_base_patch16_384.fb_in1k | timm | vit | ImageNet-1k |
| 734. | deit3_base_patch16_384.fb_in22k_ft_in1k | timm | vit | ImageNet-21k |
| 735. | deit3_huge_patch14_224.fb_in1k | timm | vit | ImageNet-1k |
| 736. | deit3_huge_patch14_224.fb_in22k_ft_in1k | timm | vit | ImageNet-21k |
| 737. | deit3_large_patch16_224.fb_in1k | timm | vit | ImageNet-1k |
| 738. | deit3_large_patch16_224.fb_in22k_ft_in1k | timm | vit | ImageNet-21k |
| 739. | deit3_large_patch16_384.fb_in1k | timm | vit | ImageNet-1k |
| 740. | deit3_large_patch16_384.fb_in22k_ft_in1k | timm | vit | ImageNet-21k |
| 741. | deit3_medium_patch16_224.fb_in1k | timm | vit | ImageNet-1k |
| 742. | deit3_medium_patch16_224.fb_in22k_ft_in1k | timm | vit | ImageNet-21k |
| 743. | deit3_small_patch16_224.fb_in1k | timm | vit | ImageNet-1k |
| 744. | deit3_small_patch16_224.fb_in22k_ft_in1k | timm | vit | ImageNet-21k |
| 745. | deit3_small_patch16_384.fb_in1k | timm | vit | ImageNet-1k |
| 746. | deit3_small_patch16_384.fb_in22k_ft_in1k | timm | vit | ImageNet-21k |
| 747. | deit_base_distilled_patch16_224.fb_in1k | timm | vit | ImageNet-1k |
| 748. | deit_base_distilled_patch16_384.fb_in1k | timm | vit | ImageNet-1k |
| 749. | deit_base_patch16_224.fb_in1k | timm | vit | ImageNet-1k |
| 750. | deit_base_patch16_384.fb_in1k | timm | vit | ImageNet-1k |
| 751. | deit_small_distilled_patch16_224.fb_in1k | timm | vit | ImageNet-1k |
| 752. | deit_small_patch16_224.fb_in1k | timm | vit | ImageNet-1k |
| 753. | deit_tiny_distilled_patch16_224.fb_in1k | timm | vit | ImageNet-1k |
| 754. | deit_tiny_patch16_224.fb_in1k | timm | vit | ImageNet-1k |
| 755. | eva02_base_patch14_448.mim_in22k_ft_in1k | timm | vit | ImageNet-21k |
| 756. | eva02_base_patch14_448.mim_in22k_ft_in22k_in1k | timm | vit | ImageNet-21k |
| 757. | eva02_large_patch14_448.mim_in22k_ft_in1k | timm | vit | ImageNet-21k |
| 758. | eva02_large_patch14_448.mim_in22k_ft_in22k_in1k | timm | vit | ImageNet-21k |
| 759. | eva02_large_patch14_448.mim_m38m_ft_in1k | timm | vit | Merged |
| 760. | eva02_large_patch14_448.mim_m38m_ft_in22k_in1k | timm | vit | Merged |
| 761. | eva02_small_patch14_336.mim_in22k_ft_in1k | timm | vit | ImageNet-21k |
| 762. | eva02_tiny_patch14_336.mim_in22k_ft_in1k | timm | vit | ImageNet-21k |
| 763. | eva_giant_patch14_224.clip_ft_in1k | timm | vit | LAION-400M |
| 764. | eva_giant_patch14_336.clip_ft_in1k | timm | vit | LAION-400M |
| 765. | eva_giant_patch14_336.m30m_ft_in22k_in1k | timm | vit | Merged |
| 766. | eva_giant_patch14_560.m30m_ft_in22k_in1k | timm | vit | Merged |
| 767. | eva_large_patch14_196.in22k_ft_in1k | timm | vit | ImageNet-21k |
| 768. | eva_large_patch14_196.in22k_ft_in22k_in1k | timm | vit | ImageNet-21k |
| 769. | eva_large_patch14_336.in22k_ft_in1k | timm | vit | ImageNet-21k |
| 770. | eva_large_patch14_336.in22k_ft_in22k_in1k | timm | vit | ImageNet-21k |
| 771. | flexivit_base.1200ep_in1k | timm | vit | ImageNet-1k |
| 772. | flexivit_base.300ep_in1k | timm | vit | ImageNet-1k |
| 773. | flexivit_base.600ep_in1k | timm | vit | ImageNet-1k |
| 774. | flexivit_large.1200ep_in1k | timm | vit | ImageNet-1k |
| 775. | flexivit_large.300ep_in1k | timm | vit | ImageNet-1k |
| 776. | flexivit_large.600ep_in1k | timm | vit | ImageNet-1k |
| 777. | flexivit_small.1200ep_in1k | timm | vit | ImageNet-1k |
| 778. | flexivit_small.300ep_in1k | timm | vit | ImageNet-1k |
| 779. | flexivit_small.600ep_in1k | timm | vit | ImageNet-1k |
| 780. | gcvit_base.in1k | timm | vit | ImageNet-1k |
| 781. | gcvit_small.in1k | timm | vit | ImageNet-1k |
| 782. | gcvit_tiny.in1k | timm | vit | ImageNet-1k |

Table continues onto next page

| No | Model ID | Source | Arch | Dataset |
|---|---|---|---|---|
| | Model summaries (continued from previous page) | | | |
| 783. | gcvit_xtiny.in1k | timm | vit | ImageNet-1k |
| 784. | gcvit_xxtiny.in1k | timm | vit | ImageNet-1k |
| 785. | mvitv2_base.fb_in1k | timm | vit | ImageNet-1k |
| 786. | mvitv2_large.fb_in1k | timm | vit | ImageNet-1k |
| 787. | mvitv2_small.fb_in1k | timm | vit | ImageNet-1k |
| 788. | mvitv2_tiny.fb_in1k | timm | vit | ImageNet-1k |
| 789. | nest_base_jx.goog_in1k | timm | vit | ImageNet-1k |
| 790. | nest_small_jx.goog_in1k | timm | vit | ImageNet-1k |
| 791. | nest_tiny_jx.goog_in1k | timm | vit | ImageNet-1k |
| 792. | pit_b_224.in1k | timm | vit | ImageNet-1k |
| 793. | pit_b_distilled_224.in1k | timm | vit | ImageNet-1k |
| 794. | pit_s_224.in1k | timm | vit | ImageNet-1k |
| 795. | pit_s_distilled_224.in1k | timm | vit | ImageNet-1k |
| 796. | pit_ti_224.in1k | timm | vit | ImageNet-1k |
| 797. | pit_ti_distilled_224.in1k | timm | vit | ImageNet-1k |
| 798. | pit_xs_224.in1k | timm | vit | ImageNet-1k |
| 799. | pit_xs_distilled_224.in1k | timm | vit | ImageNet-1k |
| 800. | swin_b | torch | vit | ImageNet-1k |
| 801. | swin_base_patch4_window12_384.ms_in1k | timm | vit | ImageNet-1k |
| 802. | swin_base_patch4_window12_384.ms_in22k_ft_in1k | timm | vit | ImageNet-21k |
| 803. | swin_base_patch4_window7_224.ms_in1k | timm | vit | ImageNet-1k |
| 804. | swin_base_patch4_window7_224.ms_in22k_ft_in1k | timm | vit | ImageNet-21k |
| 805. | swin_large_patch4_window12_384.ms_in22k_ft_in1k | timm | vit | ImageNet-21k |
| 806. | swin_large_patch4_window7_224.ms_in22k_ft_in1k | timm | vit | ImageNet-21k |
| 807. | swin_s | torch | vit | ImageNet-1k |
| 808. | swin_s3_base_224.ms_in1k | timm | vit | ImageNet-1k |
| 809. | swin_s3_small_224.ms_in1k | timm | vit | ImageNet-1k |
| 810. | swin_s3_tiny_224.ms_in1k | timm | vit | ImageNet-1k |
| 811. | swin_small_patch4_window7_224.ms_in1k | timm | vit | ImageNet-1k |
| 812. | swin_small_patch4_window7_224.ms_in22k_ft_in1k | timm | vit | ImageNet-21k |
| 813. | swin_t | torch | vit | ImageNet-1k |
| 814. | swin_tiny_patch4_window7_224.ms_in1k | timm | vit | ImageNet-1k |
| 815. | swin_tiny_patch4_window7_224.ms_in22k_ft_in1k | timm | vit | ImageNet-21k |
| 816. | swin_v2_b | torch | vit | ImageNet-1k |
| 817. | swin_v2_s | torch | vit | ImageNet-1k |
| 818. | swin_v2_t | torch | vit | ImageNet-1k |
| 819. | swinv2_base_window12to16_192to256.ms_in22k_ft_in1k | timm | vit | ImageNet-21k |
| 820. | swinv2_base_window12to24_192to384.ms_in22k_ft_in1k | timm | vit | ImageNet-21k |
| 821. | swinv2_base_window16_256.ms_in1k | timm | vit | ImageNet-1k |
| 822. | swinv2_base_window8_256.ms_in1k | timm | vit | ImageNet-1k |
| 823. | swinv2_large_window12to16_192to256.ms_in22k_ft_in1k | timm | vit | ImageNet-21k |
| 824. | swinv2_large_window12to24_192to384.ms_in22k_ft_in1k | timm | vit | ImageNet-21k |
| 825. | swinv2_small_window16_256.ms_in1k | timm | vit | ImageNet-1k |
| 826. | swinv2_small_window8_256.ms_in1k | timm | vit | ImageNet-1k |
| 827. | swinv2_tiny_window16_256.ms_in1k | timm | vit | ImageNet-1k |
| 828. | swinv2_tiny_window8_256.ms_in1k | timm | vit | ImageNet-1k |
| 829. | twins_pcpvt_base.in1k | timm | vit | ImageNet-1k |
| 830. | twins_pcpvt_large.in1k | timm | vit | ImageNet-1k |
| 831. | twins_pcpvt_small.in1k | timm | vit | ImageNet-1k |
| 832. | twins_svt_base.in1k | timm | vit | ImageNet-1k |
| 833. | twins_svt_large.in1k | timm | vit | ImageNet-1k |
| 834. | twins_svt_small.in1k | timm | vit | ImageNet-1k |
| 835. | vit_b_16 | torch | vit | ImageNet-1k |
| 836. | vit_b_16_swag | torch | vit | Instagram |
| 837. | vit_b_32 | torch | vit | ImageNet-1k |
| 838. | vit_base_patch16_224.augreg2_in21k_ft_in1k | timm | vit | ImageNet-21k |
| 839. | vit_base_patch16_224.augreg_in1k | timm | vit | ImageNet-1k |
| 840. | vit_base_patch16_224.augreg_in21k_ft_in1k | timm | vit | ImageNet-21k |
| 841. | vit_base_patch16_224.orig_in21k_ft_in1k | timm | vit | ImageNet-21k |
| | Table continues onto next page | | | |

| No | Model ID | Source | Arch | Dataset |
|---|---|---|---|---|
| 842. | vit_base_patch16_224.sam_in1k | timm | vit | ImageNet-1k |
| 843. | vit_base_patch16_224_miil.in21k_ft_in1k | timm | vit | ImageNet-21k |
| 844. | vit_base_patch16_384.augreg_in1k | timm | vit | ImageNet-1k |
| 845. | vit_base_patch16_384.augreg_in21k_ft_in1k | timm | vit | ImageNet-21k |
| 846. | vit_base_patch16_384.orig_in21k_ft_in1k | timm | vit | ImageNet-21k |
| 847. | vit_base_patch16_clip_224.laion2b_ft_in12k_in1k | timm | vit | LAION-2B |
| 848. | vit_base_patch16_clip_224.laion2b_ft_in1k | timm | vit | LAION-2B |
| 849. | vit_base_patch16_clip_224.openai_ft_in12k_in1k | timm | vit | WIT |
| 850. | vit_base_patch16_clip_224.openai_ft_in1k | timm | vit | WIT |
| 851. | vit_base_patch16_clip_384.laion2b_ft_in12k_in1k | timm | vit | LAION-2B |
| 852. | vit_base_patch16_clip_384.laion2b_ft_in1k | timm | vit | LAION-2B |
| 853. | vit_base_patch16_clip_384.openai_ft_in12k_in1k | timm | vit | WIT |
| 854. | vit_base_patch16_clip_384.openai_ft_in1k | timm | vit | WIT |
| 855. | vit_base_patch16_rpn_224.sw_in1k | timm | vit | ImageNet-1k |
| 856. | vit_base_patch32_224.augreg_in1k | timm | vit | ImageNet-1k |
| 857. | vit_base_patch32_224.augreg_in21k_ft_in1k | timm | vit | ImageNet-21k |
| 858. | vit_base_patch32_224.sam_in1k | timm | vit | ImageNet-1k |
| 859. | vit_base_patch32_384.augreg_in1k | timm | vit | ImageNet-1k |
| 860. | vit_base_patch32_384.augreg_in21k_ft_in1k | timm | vit | ImageNet-21k |
| 861. | vit_base_patch32_clip_224.laion2b_ft_in12k_in1k | timm | vit | LAION-2B |
| 862. | vit_base_patch32_clip_224.laion2b_ft_in1k | timm | vit | LAION-2B |
| 863. | vit_base_patch32_clip_224.openai_ft_in1k | timm | vit | WIT |
| 864. | vit_base_patch32_clip_384.laion2b_ft_in12k_in1k | timm | vit | LAION-2B |
| 865. | vit_base_patch32_clip_384.openai_ft_in12k_in1k | timm | vit | WIT |
| 866. | vit_base_patch32_clip_448.laion2b_ft_in12k_in1k | timm | vit | LAION-2B |
| 867. | vit_base_patch8_224.augreg2_in21k_ft_in1k | timm | vit | ImageNet-21k |
| 868. | vit_base_patch8_224.augreg_in21k_ft_in1k | timm | vit | ImageNet-21k |
| 869. | vit_base_r50_s16_384.orig_in21k_ft_in1k | timm | vit | ImageNet-21k |
| 870. | vit_h_14_swag | torch | vit | Instagram |
| 871. | vit_huge_patch14_clip_224.laion2b_ft_in12k_in1k | timm | vit | LAION-2B |
| 872. | vit_huge_patch14_clip_224.laion2b_ft_in1k | timm | vit | LAION-2B |
| 873. | vit_huge_patch14_clip_336.laion2b_ft_in12k_in1k | timm | vit | LAION-2B |
| 874. | vit_l_16 | torch | vit | ImageNet-1k |
| 875. | vit_l_16_swag | torch | vit | Instagram |
| 876. | vit_l_32 | torch | vit | ImageNet-1k |
| 877. | vit_large_patch14_clip_224.laion2b_ft_in12k_in1k | timm | vit | LAION-2B |
| 878. | vit_large_patch14_clip_224.laion2b_ft_in1k | timm | vit | LAION-2B |
| 879. | vit_large_patch14_clip_224.openai_ft_in12k_in1k | timm | vit | WIT |
| 880. | vit_large_patch14_clip_224.openai_ft_in1k | timm | vit | WIT |
| 881. | vit_large_patch14_clip_336.laion2b_ft_in12k_in1k | timm | vit | LAION-2B |
| 882. | vit_large_patch14_clip_336.laion2b_ft_in1k | timm | vit | LAION-2B |
| 883. | vit_large_patch14_clip_336.openai_ft_in12k_in1k | timm | vit | WIT |
| 884. | vit_large_patch16_224.augreg_in21k_ft_in1k | timm | vit | ImageNet-21k |
| 885. | vit_large_patch16_384.augreg_in21k_ft_in1k | timm | vit | ImageNet-21k |
| 886. | vit_large_patch32_384.orig_in21k_ft_in1k | timm | vit | ImageNet-21k |
| 887. | vit_large_r50_s32_224.augreg_in21k_ft_in1k | timm | vit | ImageNet-21k |
| 888. | vit_large_r50_s32_384.augreg_in21k_ft_in1k | timm | vit | ImageNet-21k |
| 889. | vit_relpos_base_patch16_224.sw_in1k | timm | vit | ImageNet-1k |
| 890. | vit_relpos_base_patch16_clsgap_224.sw_in1k | timm | vit | ImageNet-1k |
| 891. | vit_relpos_base_patch32_plus_rpn_256.sw_in1k | timm | vit | ImageNet-1k |
| 892. | vit_relpos_medium_patch16_224.sw_in1k | timm | vit | ImageNet-1k |
| 893. | vit_relpos_medium_patch16_cls_224.sw_in1k | timm | vit | ImageNet-1k |
| 894. | vit_relpos_medium_patch16_rpn_224.sw_in1k | timm | vit | ImageNet-1k |
| 895. | vit_relpos_small_patch16_224.sw_in1k | timm | vit | ImageNet-1k |
| 896. | vit_small_patch16_224.augreg_in1k | timm | vit | ImageNet-1k |
| 897. | vit_small_patch16_224.augreg_in21k_ft_in1k | timm | vit | ImageNet-21k |
| 898. | vit_small_patch16_384.augreg_in1k | timm | vit | ImageNet-1k |
| 899. | vit_small_patch16_384.augreg_in21k_ft_in1k | timm | vit | ImageNet-21k |
| 900. | vit_small_patch32_224.augreg_in21k_ft_in1k | timm | vit | ImageNet-21k |

Table continues onto next page

| No | Model ID | Source | Arch | Dataset |
|---|---|---|---|---|
| 901. | vit_small_patch32_384.augreg_in21k_ft_in1k | timm | vit | ImageNet-21k |
| 902. | vit_small_r26_s32_224.augreg_in21k_ft_in1k | timm | vit | ImageNet-21k |
| 903. | vit_small_r26_s32_384.augreg_in21k_ft_in1k | timm | vit | ImageNet-21k |
| 904. | vit_srelpos_medium_patch16_224.sw_in1k | timm | vit | ImageNet-1k |
| 905. | vit_srelpos_small_patch16_224.sw_in1k | timm | vit | ImageNet-1k |
| 906. | vit_tiny_patch16_224.augreg_in21k_ft_in1k | timm | vit | ImageNet-21k |
| 907. | vit_tiny_patch16_384.augreg_in21k_ft_in1k | timm | vit | ImageNet-21k |
| 908. | vit_tiny_r_s16_p8_224.augreg_in21k_ft_in1k | timm | vit | ImageNet-21k |
| 909. | vit_tiny_r_s16_p8_384.augreg_in21k_ft_in1k | timm | vit | ImageNet-21k |
| 910. | volo_d1_224.sail_in1k | timm | vit | ImageNet-1k |
| 911. | volo_d1_384.sail_in1k | timm | vit | ImageNet-1k |
| 912. | volo_d2_224.sail_in1k | timm | vit | ImageNet-1k |
| 913. | volo_d2_384.sail_in1k | timm | vit | ImageNet-1k |
| 914. | volo_d3_224.sail_in1k | timm | vit | ImageNet-1k |
| 915. | volo_d3_448.sail_in1k | timm | vit | ImageNet-1k |
| 916. | volo_d4_224.sail_in1k | timm | vit | ImageNet-1k |
| 917. | volo_d4_448.sail_in1k | timm | vit | ImageNet-1k |
| 918. | volo_d5_224.sail_in1k | timm | vit | ImageNet-1k |
| 919. | volo_d5_448.sail_in1k | timm | vit | ImageNet-1k |
| 920. | volo_d5_512.sail_in1k | timm | vit | ImageNet-1k |
| 921. | xcit_large_24_p16_224.fb_dist_in1k | timm | vit | ImageNet-1k |
| 922. | xcit_large_24_p16_224.fb_in1k | timm | vit | ImageNet-1k |
| 923. | xcit_large_24_p16_384.fb_dist_in1k | timm | vit | ImageNet-1k |
| 924. | xcit_large_24_p8_224.fb_dist_in1k | timm | vit | ImageNet-1k |
| 925. | xcit_large_24_p8_224.fb_in1k | timm | vit | ImageNet-1k |
| 926. | xcit_large_24_p8_384.fb_dist_in1k | timm | vit | ImageNet-1k |
| 927. | xcit_medium_24_p16_224.fb_dist_in1k | timm | vit | ImageNet-1k |
| 928. | xcit_medium_24_p16_224.fb_in1k | timm | vit | ImageNet-1k |
| 929. | xcit_medium_24_p16_384.fb_dist_in1k | timm | vit | ImageNet-1k |
| 930. | xcit_medium_24_p8_224.fb_dist_in1k | timm | vit | ImageNet-1k |
| 931. | xcit_medium_24_p8_224.fb_in1k | timm | vit | ImageNet-1k |
| 932. | xcit_medium_24_p8_384.fb_dist_in1k | timm | vit | ImageNet-1k |
| 933. | xcit_nano_12_p16_224.fb_dist_in1k | timm | vit | ImageNet-1k |
| 934. | xcit_nano_12_p16_224.fb_in1k | timm | vit | ImageNet-1k |
| 935. | xcit_nano_12_p16_384.fb_dist_in1k | timm | vit | ImageNet-1k |
| 936. | xcit_nano_12_p8_224.fb_dist_in1k | timm | vit | ImageNet-1k |
| 937. | xcit_nano_12_p8_224.fb_in1k | timm | vit | ImageNet-1k |
| 938. | xcit_nano_12_p8_384.fb_dist_in1k | timm | vit | ImageNet-1k |
| 939. | xcit_small_12_p16_224.fb_dist_in1k | timm | vit | ImageNet-1k |
| 940. | xcit_small_12_p16_224.fb_in1k | timm | vit | ImageNet-1k |
| 941. | xcit_small_12_p16_384.fb_dist_in1k | timm | vit | ImageNet-1k |
| 942. | xcit_small_12_p8_224.fb_dist_in1k | timm | vit | ImageNet-1k |
| 943. | xcit_small_12_p8_224.fb_in1k | timm | vit | ImageNet-1k |
| 944. | xcit_small_12_p8_384.fb_dist_in1k | timm | vit | ImageNet-1k |
| 945. | xcit_small_24_p16_224.fb_dist_in1k | timm | vit | ImageNet-1k |
| 946. | xcit_small_24_p16_224.fb_in1k | timm | vit | ImageNet-1k |
| 947. | xcit_small_24_p16_384.fb_dist_in1k | timm | vit | ImageNet-1k |
| 948. | xcit_small_24_p8_224.fb_dist_in1k | timm | vit | ImageNet-1k |
| 949. | xcit_small_24_p8_224.fb_in1k | timm | vit | ImageNet-1k |
| 950. | xcit_small_24_p8_384.fb_dist_in1k | timm | vit | ImageNet-1k |
| 951. | xcit_tiny_12_p16_224.fb_dist_in1k | timm | vit | ImageNet-1k |
| 952. | xcit_tiny_12_p16_224.fb_in1k | timm | vit | ImageNet-1k |
| 953. | xcit_tiny_12_p16_384.fb_dist_in1k | timm | vit | ImageNet-1k |
| 954. | xcit_tiny_12_p8_224.fb_dist_in1k | timm | vit | ImageNet-1k |
| 955. | xcit_tiny_12_p8_224.fb_in1k | timm | vit | ImageNet-1k |
| 956. | xcit_tiny_12_p8_384.fb_dist_in1k | timm | vit | ImageNet-1k |
| 957. | xcit_tiny_24_p16_224.fb_dist_in1k | timm | vit | ImageNet-1k |
| 958. | xcit_tiny_24_p16_224.fb_in1k | timm | vit | ImageNet-1k |
| 959. | xcit_tiny_24_p16_384.fb_dist_in1k | timm | vit | ImageNet-1k |

| No | Model ID | Source | Arch | Dataset |
|---|---|---|---|---|
| 960. | xcit_tiny_24_p8_224.fb_dist_in1k | timm | vit | ImageNet-1k |
| 961. | xcit_tiny_24_p8_224.fb_in1k | timm | vit | ImageNet-1k |
| 962. | xcit_tiny_24_p8_384.fb_dist_in1k | timm | vit | ImageNet-1k |

## F.4   Fine-grained Superclasses

In this section, we detail the superclasses we use for detecting fine-grained and fine-grained OOV errors and their constituting IMAGENET classes. When defining these superclasses, we focused on visual and semantic similarity rather than positions in the WordNet hierarchy. We believe that the latter is fundamentally unsuitable to obtain high-quality groupings, as visually similar or closely related classes often end up having a large WordNet distance, while very different classes are close. Further, we ensured that superclass definitions are not overly generic, providing a much more fine-grained split than prior works. The superclasses are defined programmatically in our code and can be easily reused and adapted by others.

List of superclass definitions:

1. **butterfly** – 6 classes:

(n02276258, admiral)                    (n02277742, ringlet)
(n02279972, monarch)                    (n02280649, cabbage butterfly)
(n02281406, sulphur butterfly)          (n02281787, lycaenid)

2. **beetle_plus** – 13 classes:

(n02165105, tiger beetle)               (n02165456, ladybug)
(n02167151, ground beetle)              (n02168699, long-horned beetle)
(n02169497, leaf beetle)                (n02172182, dung beetle)
(n02174001, rhinoceros beetle)          (n02177972, weevil)
(n02226429, grasshopper)                (n02229544, cricket)
(n02233338, cockroach)                  (n02256656, cicada)
(n02259212, leafhopper)

3. **insect_rest** – 13 classes:

(n02190166, fly)                        (n02206856, bee)
(n02219486, ant)                        (n02226429, grasshopper)
(n02229544, cricket)                    (n02231487, walking stick)
(n02233338, cockroach)                  (n02236044, mantis)
(n02256656, cicada)                     (n02259212, leafhopper)
(n02264363, lacewing)                   (n02268443, dragonfly)
(n02268853, damselfly)

4. **isopod_trilobite_chiton** – 3 classes:

(n01768244, trilobite)                  (n01955084, chiton)
(n01990800, isopod)

5. **crab_scorpion** – 9 classes:

(n01770393, scorpion)                   (n01978287, Dungeness crab)
(n01978455, rock crab)                  (n01980166, fiddler crab)
(n01981276, king crab)                  (n01983481, American lobster)
(n01984695, spiny lobster)              (n01985128, crayfish)
(n01986214, hermit crab)

6. **spider** – 8 classes:

(n01770081, harvestman)                 (n01773157, black and gold garden spider)
(n01773549, barn spider)                (n01773797, garden spider)
(n01774384, black widow)                (n01774750, tarantula)
(n01775062, wolf spider)                (n01776313, tick)

7. **marine_invertebrate** – 7 classes:

```
(n01914609, sea anemone)              (n01917289, brain coral)
(n01950731, sea slug)                 (n02317335, starfish)
(n02319095, sea urchin)               (n09256479, coral reef)
(n12985857, coral fungus)
```

8. **conch_chambered_nautilus** – 2 classes:
```
(n01943899, conch)                    (n01968897, chambered nautilus)
```

9. **invertebrate_rest** – 7 classes:
```
(n01784675, centipede)                (n01924916, flatworm)
(n01930112, nematode)                 (n01944390, snail)
(n01945685, slug)                     (n01950731, sea slug)
(n02321529, sea cucumber)
```

10. **salamander_lizard** – 16 classes:
```
(n01629819, European fire salamander) (n01630670, common newt)
(n01631663, eft)                      (n01632458, spotted salamander)
(n01632777, axolotl)                  (n01675722, banded gecko)
(n01677366, common iguana)            (n01682714, American chameleon)
(n01685808, whiptail)                 (n01687978, agama)
(n01688243, frilled lizard)           (n01689811, alligator lizard)
(n01692333, Gila monster)             (n01693334, green lizard)
(n01694178, African chameleon)        (n01695060, Komodo dragon)
```

11. **frog** – 3 classes:
```
(n01641577, bullfrog)                 (n01644373, tree frog)
(n01644900, tailed frog)
```

12. **shark_and_aquatic_mammal** – 7 classes:
```
(n01484850, great white shark)        (n01491361, tiger shark)
(n01494475, hammerhead)               (n02066245, grey whale)
(n02071294, killer whale)             (n02074367, dugong)
(n02077923, sea lion)
```

13. **ray** – 2 classes:
```
(n01496331, electric ray)             (n01498041, stingray)
```

14. **fish_rest** – 11 classes:
```
(n01440764, tench)                    (n01443537, goldfish)
(n02514041, barracouta)               (n02526121, eel)
(n02536864, coho)                     (n02606052, rock beauty)
(n02607072, anemone fish)             (n02640242, sturgeon)
(n02641379, gar)                      (n02643566, lionfish)
(n02655020, puffer)
```

15. **turtle** – 5 classes:
```
(n01664065, loggerhead)               (n01665541, leatherback turtle)
(n01667114, mud turtle)               (n01667778, terrapin)
(n01669191, box turtle)
```

16. **crocodilian_reptile** – 2 classes:
```
(n01697457, African crocodile)        (n01698640, American alligator)
```

17. **snake** – 17 classes:
```
(n01728572, thunder snake)            (n01728920, ringneck snake)
(n01729322, hognose snake)            (n01729977, green snake)
(n01734418, king snake)               (n01735189, garter snake)
(n01737021, water snake)              (n01739381, vine snake)
(n01740131, night snake)              (n01742172, boa constrictor)
(n01744401, rock python)              (n01748264, Indian cobra)
(n01749939, green mamba)              (n01751748, sea snake)
(n01753488, horned viper)             (n01755581, diamondback)
(n01756291, sidewinder)
```

18. **green_snake_and_lizards** – 5 classes:

```
(n01682714, American chameleon)        (n01693334, green lizard)
(n01729977, green snake)               (n01739381, vine snake)
(n01749939, green mamba)
```

19. **non_green_snakes** – 15 classes:
```
(n01689811, alligator lizard)          (n01728572, thunder snake)
(n01728920, ringneck snake)            (n01729322, hognose snake)
(n01734418, king snake)                (n01735189, garter snake)
(n01737021, water snake)               (n01740131, night snake)
(n01742172, boa constrictor)           (n01744401, rock python)
(n01748264, Indian cobra)              (n01751748, sea snake)
(n01753488, horned viper)              (n01755581, diamondback)
(n01756291, sidewinder)
```

20. **cock_hen_gallinaceous** – 12 classes:
```
(n01514668, cock)                      (n01514859, hen)
(n01795545, black grouse)              (n01796340, ptarmigan)
(n01797886, ruffed grouse)             (n01798484, prairie chicken)
(n01806143, peacock)                   (n01806567, quail)
(n01807496, partridge)                 (n02017213, European gallinule)
(n02018207, American coot)             (n02018795, bustard)
```

21. **small_bird** – 15 classes:
```
(n01530575, brambling)                 (n01531178, goldfinch)
(n01532829, house finch)               (n01534433, junco)
(n01537544, indigo bunting)            (n01558993, robin)
(n01560419, bulbul)                    (n01580077, jay)
(n01582220, magpie)                    (n01592084, chickadee)
(n01601694, water ouzel)               (n01824575, coucal)
(n01828970, bee eater)                 (n01833805, hummingbird)
(n01843065, jacamar)
```

22. **bird_of_prey** – 3 classes:
```
(n01608432, kite)                      (n01614925, bald eagle)
(n01616318, vulture)
```

23. **parrot** – 4 classes:
```
(n01817953, African grey)              (n01818515, macaw)
(n01819313, sulphur-crested cockatoo)  (n01820546, lorikeet)
```

24. **big_beak_bird** – 2 classes:
```
(n01829413, hornbill)                  (n01843383, toucan)
```

25. **aquatic_bird_and_ostrich** – 25 classes:
```
(n01518878, ostrich)                   (n01847000, drake)
(n01855032, red-breasted merganser)    (n01855672, goose)
(n01860187, black swan)                (n02002556, white stork)
(n02002724, black stork)               (n02006656, spoonbill)
(n02007558, flamingo)                  (n02009229, little blue heron)
(n02009912, American egret)            (n02011460, bittern)
(n02012849, crane)                     (n02013706, limpkin)
(n02017213, European gallinule)        (n02018207, American coot)
(n02018795, bustard)                   (n02025239, ruddy turnstone)
(n02027492, red-backed sandpiper)      (n02028035, redshank)
(n02033041, dowitcher)                 (n02037110, oystercatcher)
(n02051845, pelican)                   (n02056570, king penguin)
(n02058221, albatross)
```

26. **tusker_elephant** – 3 classes:
```
(n01871265, tusker)                    (n02504013, Indian elephant)
(n02504458, African elephant)
```

27. **primate** – 20 classes:

```
(n02480495, orangutan)              (n02480855, gorilla)
(n02481823, chimpanzee)             (n02483362, gibbon)
(n02483708, siamang)                (n02484975, guenon)
(n02486261, patas)                  (n02486410, baboon)
(n02487347, macaque)                (n02488291, langur)
(n02488702, colobus)                (n02489166, proboscis monkey)
(n02490219, marmoset)               (n02492035, capuchin)
(n02492660, howler monkey)          (n02493509, titi)
(n02493793, spider monkey)          (n02494079, squirrel monkey)
(n02497673, Madagascar cat)         (n02500267, indri)
```

28. **bear_panda_marsupial** – 10 classes:
```
(n01877812, wallaby)                (n01882714, koala)
(n01883070, wombat)                 (n02132136, brown bear)
(n02133161, American black bear)    (n02134084, ice bear)
(n02134418, sloth bear)             (n02457408, three-toed sloth)
(n02509815, lesser panda)           (n02510455, giant panda)
```

29. **viverrine_musteline_rodent_prototherian** – 19 classes:
```
(n01872401, echidna)                (n01873310, platypus)
(n01877812, wallaby)                (n02137549, mongoose)
(n02138441, meerkat)                (n02342885, hamster)
(n02346627, porcupine)              (n02356798, fox squirrel)
(n02361337, marmot)                 (n02363005, beaver)
(n02364673, guinea pig)             (n02441942, weasel)
(n02442845, mink)                   (n02443114, polecat)
(n02443484, black-footed ferret)    (n02444819, otter)
(n02445715, skunk)                  (n02447366, badger)
(n02454379, armadillo)
```

30. **rabbit** – 3 classes:
```
(n02325366, wood rabbit)            (n02326432, hare)
(n02328150, Angora)
```

31. **ungulate** – 17 classes:
```
(n02389026, sorrel)                 (n02391049, zebra)
(n02395406, hog)                    (n02396427, wild boar)
(n02397096, warthog)                (n02398521, hippopotamus)
(n02403003, ox)                     (n02408429, water buffalo)
(n02410509, bison)                  (n02412080, ram)
(n02415577, bighorn)                (n02417914, ibex)
(n02422106, hartebeest)             (n02422699, impala)
(n02423022, gazelle)                (n02437312, Arabian camel)
(n02437616, llama)
```

32. **domestic_cat** – 5 classes:
```
(n02123045, tabby)                  (n02123159, tiger cat)
(n02123394, Persian cat)            (n02123597, Siamese cat)
(n02124075, Egyptian cat)
```

33. **wild_cat_big_cat** – 8 classes:
```
(n02125311, cougar)                 (n02127052, lynx)
(n02128385, leopard)                (n02128757, snow leopard)
(n02128925, jaguar)                 (n02129165, lion)
(n02129604, tiger)                  (n02130308, cheetah)
```

34. **wolf_wild_dog_hyena_fox** – 12 classes:
```
(n02114367, timber wolf)            (n02114548, white wolf)
(n02114712, red wolf)               (n02114855, coyote)
(n02115641, dingo)                  (n02115913, dhole)
(n02116738, African hunting dog)    (n02117135, hyena)
(n02119022, red fox)                (n02119789, kit fox)
(n02120079, Arctic fox)             (n02120505, grey fox)
```

35. **pug_boxer_bulldog** – 9 classes:

(n02093256, Staffordshire bullterrier)   (n02093428, American Staffordshire terrier)
(n02096585, Boston bull)                 (n02106550, Rottweiler)
(n02108089, boxer)                       (n02108422, bull mastiff)
(n02108915, French bulldog)              (n02110958, pug)
(n02112706, Brabancon griffon)

36. **toy_dog** – 31 classes:

(n02085620, Chihuahua)                   (n02085782, Japanese spaniel)
(n02085936, Maltese dog)                 (n02086079, Pekinese)
(n02086240, Shih-Tzu)                    (n02086646, Blenheim spaniel)
(n02086910, papillon)                    (n02087046, toy terrier)
(n02093754, Border terrier)              (n02094114, Norfolk terrier)
(n02094258, Norwich terrier)             (n02094433, Yorkshire terrier)
(n02096177, cairn)                       (n02096294, Australian terrier)
(n02096437, Dandie Dinmont)              (n02096585, Boston bull)
(n02097047, miniature schnauzer)         (n02097474, Tibetan terrier)
(n02097658, silky terrier)               (n02098105, soft-coated wheaten terrier)
(n02098286, West Highland white terrier) (n02098413, Lhasa)
(n02102318, cocker spaniel)              (n02107312, miniature pinscher)
(n02108915, French bulldog)              (n02110627, affenpinscher)
(n02112018, Pomeranian)                  (n02112706, Brabancon griffon)
(n02113624, toy poodle)                  (n02113712, miniature poodle)
(n02113799, standard poodle)

37. **corgi** – 2 classes:

(n02113023, Pembroke)                    (n02113186, Cardigan)

38. **terrier** – 26 classes:

(n02090721, Irish wolfhound)             (n02091635, otterhound)
(n02093647, Bedlington terrier)          (n02093754, Border terrier)
(n02093859, Kerry blue terrier)          (n02093991, Irish terrier)
(n02094114, Norfolk terrier)             (n02094258, Norwich terrier)
(n02094433, Yorkshire terrier)           (n02095314, wire-haired fox terrier)
(n02095570, Lakeland terrier)            (n02095889, Sealyham terrier)
(n02096051, Airedale)                    (n02096177, cairn)
(n02096294, Australian terrier)          (n02096437, Dandie Dinmont)
(n02097047, miniature schnauzer)         (n02097130, giant schnauzer)
(n02097209, standard schnauzer)          (n02097298, Scotch terrier)
(n02097474, Tibetan terrier)             (n02097658, silky terrier)
(n02098105, soft-coated wheaten terrier) (n02098286, West Highland white terrier)
(n02098413, Lhasa)                        (n02110627, affenpinscher)

39. **poodle_like** – 7 classes:

(n02093647, Bedlington terrier)          (n02093859, Kerry blue terrier)
(n02099429, curly-coated retriever)      (n02102973, Irish water spaniel)
(n02113624, toy poodle)                  (n02113712, miniature poodle)
(n02113799, standard poodle)

40. **standard_shepherd_dog** – 11 classes:

(n02091467, Norwegian elkhound)          (n02104365, schipperke)
(n02105056, groenendael)                 (n02105162, malinois)
(n02105412, kelpie)                      (n02105855, Shetland sheepdog)
(n02106030, collie)                      (n02106166, Border collie)
(n02106550, Rottweiler)                  (n02106662, German shepherd)
(n02112350, keeshond)

41. **hairy_shepherd_dog** – 6 classes:

(n02097474, Tibetan terrier)             (n02105251, briard)
(n02105505, komondor)                    (n02105641, Old English sheepdog)
(n02106382, Bouvier des Flandres)        (n02110627, affenpinscher)

42. **beagle_hound_pointer_spaniel_setter** – 27 classes:

(n02087394, Rhodesian ridgeback)       (n02088238, basset)
(n02088364, beagle)                    (n02088466, bloodhound)
(n02088632, bluetick)                  (n02089078, black-and-tan coonhound)
(n02089867, Walker hound)              (n02089973, English foxhound)
(n02090379, redbone)                   (n02092339, Weimaraner)
(n02100236, German short-haired pointer)  (n02100583, vizsla)
(n02100735, English setter)            (n02100877, Irish setter)
(n02101006, Gordon setter)             (n02101388, Brittany spaniel)
(n02101556, clumber)                   (n02102040, English springer)
(n02102177, Welsh springer spaniel)    (n02102318, cocker spaniel)
(n02102480, Sussex spaniel)            (n02102973, Irish water spaniel)
(n02107142, Doberman)                  (n02109047, Great Dane)
(n02110341, dalmatian)                 (n02110806, basenji)
(n02113978, Mexican hairless)

43. **spaniels** – 5 classes:

(n02086646, Blenheim spaniel)          (n02101388, Brittany spaniel)
(n02102040, English springer)          (n02102177, Welsh springer spaniel)
(n02102318, cocker spaniel)

44. **skinny_hound_rest** – 12 classes:

(n02088094, Afghan hound)              (n02090622, borzoi)
(n02090721, Irish wolfhound)           (n02091032, Italian greyhound)
(n02091134, whippet)                   (n02091244, Ibizan hound)
(n02091635, otterhound)                (n02091831, Saluki)
(n02092002, Scottish deerhound)        (n02107142, Doberman)
(n02110806, basenji)                   (n02113978, Mexican hairless)

45. **pinscher** – 3 classes:

(n02106550, Rottweiler)                (n02107142, Doberman)
(n02107312, miniature pinscher)

46. **sennenhunde_and_big_dogs** – 15 classes:

(n02088364, beagle)                    (n02106166, Border collie)
(n02106550, Rottweiler)                (n02107574, Greater Swiss Mountain dog)
(n02107683, Bernese mountain dog)      (n02107908, Appenzeller)
(n02108000, EntleBucher)               (n02108089, boxer)
(n02108422, bull mastiff)              (n02108551, Tibetan mastiff)
(n02109047, Great Dane)                (n02109525, Saint Bernard)
(n02111129, Leonberg)                  (n02111277, Newfoundland)
(n02112137, chow)

47. **retriever_like** – 10 classes:

(n02090379, redbone)                   (n02099267, flat-coated retriever)
(n02099429, curly-coated retriever)    (n02099601, golden retriever)
(n02099712, Labrador retriever)        (n02099849, Chesapeake Bay retriever)
(n02101006, Gordon setter)             (n02104029, kuvasz)
(n02111277, Newfoundland)              (n02111500, Great Pyrenees)

48. **husky** – 6 classes:

(n02091467, Norwegian elkhound)        (n02109961, Eskimo dog)
(n02110063, malamute)                  (n02110185, Siberian husky)
(n02111889, Samoyed)                   (n02112350, keeshond)

49. **spitz** – 4 classes:

(n02111889, Samoyed)                   (n02112018, Pomeranian)
(n02112137, chow)                      (n02112350, keeshond)

50. **fungi** – 7 classes:

(n07734744, mushroom)                  (n12998815, agaric)
(n13037406, gyromitra)                 (n13040303, stinkhorn)
(n13044778, earthstar)                 (n13052670, hen-of-the-woods)
(n13054560, bolete)

51. **geological_formation** – 9 classes:

(n09193705, alp)  (n09246464, cliff)
(n09288635, geyser)  (n09332890, lakeside)
(n09399592, promontory)  (n09421951, sandbar)
(n09428293, seashore)  (n09468604, valley)
(n09472597, volcano)

52. **bread** – 4 classes:

(n07684084, French loaf)  (n07693725, bagel)
(n07695742, pretzel)  (n07860988, dough)

53. **cruciferous_vegetable** – 3 classes:

(n07714571, head cabbage)  (n07714990, broccoli)
(n07715103, cauliflower)

54. **cauliflower_custard_apple** – 2 classes:

(n07715103, cauliflower)  (n07760859, custard apple)

55. **zucchini_cucumber** – 2 classes:

(n07716358, zucchini)  (n07718472, cucumber)

56. **squash** – 4 classes:

(n07716358, zucchini)  (n07716906, spaghetti squash)
(n07717410, acorn squash)  (n07717556, butternut squash)

57. **reproductive_structure** – 12 classes:

(n07720875, bell pepper)  (n07742313, Granny Smith)
(n07745940, strawberry)  (n07747607, orange)
(n07749582, lemon)  (n07753113, fig)
(n07754684, jackfruit)  (n07760859, custard apple)
(n07768694, pomegranate)  (n12267677, acorn)
(n12620546, hip)  (n12768682, buckeye)

58. **corn** – 2 classes:

(n12144580, corn)  (n13133613, ear)

59. **cardoon_artichoke** – 3 classes:

(n07718747, artichoke)  (n07730033, cardoon)
(n07753275, pineapple)

60. **dessert** – 3 classes:

(n07613480, trifle)  (n07614500, ice cream)
(n07836838, chocolate sauce)

61. **baby_bed** – 3 classes:

(n02804414, bassinet)  (n03125729, cradle)
(n03131574, crib)

62. **chair** – 4 classes:

(n02791124, barber chair)  (n03376595, folding chair)
(n04099969, rocking chair)  (n04429376, throne)

63. **entertainment_center_home_theatre** – 2 classes:

(n03290653, entertainment center)  (n03529860, home theater)

64. **cabinet** – 7 classes:

(n02870880, bookcase)  (n03016953, chiffonier)
(n03018349, china cabinet)  (n03290653, entertainment center)
(n03337140, file)  (n03742115, medicine chest)
(n04550184, wardrobe)

65. **table** – 2 classes:

(n03179701, desk)  (n03201208, dining table)

66. **robe_overgarment_cloak** – 10 classes:

```
(n02667093, abaya)                        (n02669723, academic gown)
(n02730930, apron)                        (n03045698, cloak)
(n03404251, fur coat)                     (n03617480, kimono)
(n03630383, lab coat)                     (n03980874, poncho)
(n04479046, trench coat)                  (n04532106, vestment)
```

67. **sweater** – 3 classes:
```
(n02963159, cardigan)                     (n03980874, poncho)
(n04370456, sweatshirt)
```

68. **suits** – 3 classes:
```
(n03763968, military uniform)             (n03877472, pajama)
(n04350905, suit)
```

69. **woolen_pieces** – 7 classes:
```
(n02963159, cardigan)                     (n03207743, dishrag)
(n03775071, mitten)                       (n03980874, poncho)
(n04229816, ski mask)                     (n04325704, stole)
(n04599235, wool)
```

70. **caps** – 3 classes:
```
(n02807133, bathing cap)                  (n02869837, bonnet)
(n04209133, shower cap)
```

71. **swimsuit** – 4 classes:
```
(n02837789, bikini)                       (n02892767, brassiere)
(n03710637, maillot)                      (n03710721, maillot)
```

72. **woman_dress** – 3 classes:
```
(n03450230, gown)                         (n03534580, hoopskirt)
(n03866082, overskirt)
```

73. **neckwear** – 4 classes:
```
(n02865351, bolo tie)                     (n02883205, bow tie)
(n03814906, necklace)                     (n04591157, Windsor tie)
```

74. **sock_and_glove** – 3 classes:
```
(n03026506, Christmas stocking)           (n03775071, mitten)
(n04254777, sock)
```

75. **breathing_device** – 3 classes:
```
(n03424325, gasmask)                      (n03868863, oxygen mask)
(n04251144, snorkel)
```

76. **helmet** – 3 classes:
```
(n03127747, crash helmet)                 (n03379051, football helmet)
(n03929855, pickelhaube)
```

77. **pickelhaube_bearskin** – 2 classes:
```
(n02817516, bearskin)                     (n03929855, pickelhaube)
```

78. **hat** – 3 classes:
```
(n02869837, bonnet)                       (n03124170, cowboy hat)
(n04259630, sombrero)
```

79. **armor** – 4 classes:
```
(n02895154, breastplate)                  (n03000247, chain mail)
(n03146219, cuirass)                      (n04192698, shield)
```

80. **cuirass_bulletproof_vest** – 2 classes:
```
(n02916936, bulletproof vest)             (n03146219, cuirass)
```

81. **footwear_rest** – 5 classes:
```
(n03047690, clog)                         (n03124043, cowboy boot)
(n03680355, Loafer)                       (n04120489, running shoe)
(n04133789, sandal)
```

82. **top_cover** – 4 classes:
```
(n02877765, bottlecap)                    (n03657121, lens cap)
(n03717622, manhole cover)                (n04019541, puck)
```

83. **lamp_light** – 5 classes:
  (n02948072, candle)                    (n03590841, jack-o'-lantern)
  (n03637318, lampshade)                 (n04286575, spotlight)
  (n04380533, table lamp)

84. **candle_torch** – 2 classes:
  (n02948072, candle)                    (n04456115, torch)

85. **ship_boat** – 15 classes:
  (n02687172, aircraft carrier)          (n02951358, canoe)
  (n02981792, catamaran)                 (n03095699, container ship)
  (n03344393, fireboat)                  (n03447447, gondola)
  (n03662601, lifeboat)                  (n03673027, liner)
  (n03947888, pirate)                    (n04147183, schooner)
  (n04273569, speedboat)                 (n04347754, submarine)
  (n04483307, trimaran)                  (n04606251, wreck)
  (n04612504, yawl)

86. **tank** – 3 classes:
  (n02704792, amphibian)                 (n03478589, half track)
  (n04389033, tank)

87. **snow_sled** – 3 classes:
  (n02860847, bobsled)                   (n03218198, dogsled)
  (n04252077, snowmobile)

88. **motor_cycle** – 6 classes:
  (n02835271, bicycle-built-for-two)     (n03785016, moped)
  (n03791053, motor scooter)             (n03792782, mountain bike)
  (n04482393, tricycle)                  (n04509417, unicycle)

89. **aircraft** – 3 classes:
  (n02690373, airliner)                  (n02692877, airship)
  (n04552348, warplane)

90. **balloon** – 3 classes:
  (n02692877, airship)                   (n02782093, balloon)
  (n03888257, parachute)

91. **train** – 6 classes:
  (n02917067, bullet train)              (n03272562, electric locomotive)
  (n03393912, freight car)               (n03895866, passenger car)
  (n04310018, steam locomotive)          (n04335435, streetcar)

92. **car_bus_truck** – 23 classes:
  (n02701002, ambulance)                 (n02704792, amphibian)
  (n02814533, beach wagon)               (n02930766, cab)
  (n03100240, convertible)               (n03345487, fire engine)
  (n03417042, garbage truck)             (n03594945, jeep)
  (n03670208, limousine)                 (n03769881, minibus)
  (n03770679, minivan)                   (n03777568, Model T)
  (n03796401, moving van)                (n03930630, pickup)
  (n03977966, police van)                (n04037443, racer)
  (n04065272, recreational vehicle)      (n04146614, school bus)
  (n04252225, snowplow)                  (n04285008, sports car)
  (n04461696, tow truck)                 (n04467665, trailer truck)
  (n04487081, trolleybus)

93. **racer_go_kart** – 2 classes:
  (n03444034, go-kart)                   (n04037443, racer)

94. **work_cart** – 4 classes:
  (n03384352, forklift)                  (n03444034, go-kart)
  (n03445924, golfcart)                  (n03649909, lawn mower)

95. **open_cart** – 4 classes:
  (n02797295, barrow)                    (n03538406, horse cart)
  (n03599486, jinrikisha)                (n03868242, oxcart)

96. **rocket_missile** – 3 classes:
  (n03773504, missile)                   (n04008634, projectile)
  (n04266014, space shuttle)

97. **keyboard_instrument** – 3 classes:
  (n03452741, grand piano)               (n03854065, organ)
  (n04515003, upright)

98. **stringed_instrument** – 5 classes:
  (n02676566, acoustic guitar)           (n02787622, banjo)
  (n02992211, cello)                     (n03272010, electric guitar)
  (n04536866, violin)

99. **wind_instrument_brass** – 4 classes:
  (n03110669, cornet)                    (n03394916, French horn)
  (n04141076, sax)                       (n04487394, trombone)

100. **wind_instrument_woodwind** – 4 classes:
  (n02804610, bassoon)                   (n03372029, flute)
  (n03838899, oboe)                      (n04141076, sax)

101. **wind_instrument_whistle** – 4 classes:
  (n02804610, bassoon)                   (n03372029, flute)
  (n03838899, oboe)                      (n04579432, whistle)

102. **percussion_instrument** – 5 classes:
  (n03017168, chime)                     (n03249569, drum)
  (n03447721, gong)                      (n03721384, marimba)
  (n04311174, steel drum)

103. **public_place** – 12 classes:
  (n02776631, bakery)                    (n02791270, barbershop)
  (n02871525, bookshop)                  (n02927161, butcher shop)
  (n03032252, cinema)                    (n03089624, confectionery)
  (n03461385, grocery store)             (n03661043, library)
  (n04081281, restaurant)                (n04200800, shoe shop)
  (n04443257, tobacco shop)              (n04462240, toyshop)

104. **bridge_pier_dam** – 5 classes:
  (n03160309, dam)                       (n03933933, pier)
  (n04311004, steel arch bridge)         (n04366367, suspension bridge)
  (n04532670, viaduct)

105. **game_ball** – 9 classes:
  (n02799071, baseball)                  (n02802426, basketball)
  (n03134739, croquet ball)              (n03445777, golf ball)
  (n03942813, ping-pong ball)            (n04118538, rugby ball)
  (n04254680, soccer ball)               (n04409515, tennis ball)
  (n04540053, volleyball)

106. **computer_monitor** – 8 classes:
  (n03180011, desktop computer)          (n03485407, hand-held computer)
  (n03642806, laptop)                    (n03782006, monitor)
  (n03832673, notebook)                  (n03857828, oscilloscope)
  (n04152593, screen)                    (n04404412, television)

107. **keyboard** – 6 classes:
  (n03085013, computer keyboard)         (n03485407, hand-held computer)
  (n03642806, laptop)                    (n03832673, notebook)
  (n04264628, space bar)                 (n04505470, typewriter keyboard)

108. **player** – 4 classes:

(n02979186, cassette player)          (n02988304, CD player)
(n04041544, radio)                    (n04392985, tape player)

109. **small_electronic_equipment** – 9 classes:

(n02978881, cassette)                 (n02992529, cellular telephone)
(n03271574, electric fan)             (n03485407, hand-held computer)
(n03492542, hard disc)                (n03584254, iPod)
(n03602883, joystick)                 (n04074963, remote control)
(n04372370, switch)

110. **phone** – 3 classes:

(n02992529, cellular telephone)       (n03187595, dial telephone)
(n03902125, pay-phone)

111. **radio_modem** – 3 classes:

(n03492542, hard disc)                (n03777754, modem)
(n04041544, radio)

112. **big_electronic_equipment** – 6 classes:

(n03691459, loudspeaker)              (n03777754, modem)
(n03857828, oscilloscope)             (n03924679, photocopier)
(n04004767, printer)                  (n04009552, projector)

113. **clock** – 9 classes:

(n02708093, analog clock)             (n02794156, barometer)
(n03706229, magnetic compass)         (n03841143, odometer)
(n03891332, parking meter)            (n04141975, scale)
(n04328186, stopwatch)                (n04355338, sundial)
(n04548280, wall clock)

114. **digital_clock** – 5 classes:

(n03196217, digital clock)            (n03197337, digital watch)
(n04141975, scale)                    (n04149813, scoreboard)
(n04328186, stopwatch)

115. **ruler** – 3 classes:

(n04118776, rule)                     (n04238763, slide rule)
(n04376876, syringe)

116. **firearm** – 3 classes:

(n02749479, assault rifle)            (n04086273, revolver)
(n04090263, rifle)

117. **gymnastic_apparatus** – 3 classes:

(n02777292, balance beam)             (n03535780, horizontal bar)
(n03888605, parallel bars)

118. **barbell_dumbbell** – 2 classes:

(n02790996, barbell)                  (n03255030, dumbbell)

119. **dish** – 13 classes:

(n04263257, soup bowl)                (n07579787, plate)
(n07583066, guacamole)                (n07584110, consomme)
(n07590611, hot pot)                  (n07697313, cheeseburger)
(n07697537, hotdog)                   (n07711569, mashed potato)
(n07831146, carbonara)                (n07871810, meat loaf)
(n07873807, pizza)                    (n07875152, potpie)
(n07880968, burrito)

120. **beverage_pot_jug** – 18 classes:

```
(n02815834, beaker)                        (n02823750, beer glass)
(n03062245, cocktail shaker)               (n03063599, coffee mug)
(n03063689, coffeepot)                     (n03443371, goblet)
(n03733805, measuring cup)                 (n03950228, pitcher)
(n04131690, saltshaker)                    (n04254120, soap dispenser)
(n04398044, teapot)                        (n04522168, vase)
(n04560804, water jug)                     (n04579145, whiskey jug)
(n07892512, red wine)                      (n07920052, espresso)
(n07930864, cup)                           (n07932039, eggnog)
```

121. **spatula_spoon_ladle** – 3 classes:
```
(n03633091, ladle)                         (n04270147, spatula)
(n04597913, wooden spoon)
```

122. **utensils** – 11 classes:
```
(n02909870, bucket)                        (n02939185, caldron)
(n03133878, Crock Pot)                     (n03259280, Dutch oven)
(n03400231, frying pan)                    (n03775546, mixing bowl)
(n03786901, mortar)                        (n03992509, potter's wheel)
(n04263257, soup bowl)                     (n04332243, strainer)
(n04596742, wok)
```

123. **bucket_pot** – 2 classes:
```
(n02909870, bucket)                        (n03991062, pot)
```

124. **bottle** – 4 classes:
```
(n02823428, beer bottle)                   (n03983396, pop bottle)
(n04557648, water bottle)                  (n04591713, wine bottle)
```

125. **tub_basin** – 3 classes:
```
(n02808440, bathtub)                       (n04493381, tub)
(n04553703, washbasin)
```

126. **barrel_can** – 4 classes:
```
(n02747177, ashcan)                        (n02795169, barrel)
(n03764736, milk can)                      (n04049303, rain barrel)
```

127. **big_appliance** – 7 classes:
```
(n03207941, dishwasher)                    (n03297495, espresso maker)
(n03761084, microwave)                     (n04070727, refrigerator)
(n04111531, rotisserie)                    (n04330267, stove)
(n04554684, washer)
```

128. **kitchen_appliance** – 4 classes:
```
(n03297495, espresso maker)                (n03761084, microwave)
(n04442312, toaster)                       (n04542943, waffle iron)
```

129. **small_appliance** – 4 classes:
```
(n03483316, hand blower)                   (n03584829, iron)
(n04179913, sewing machine)                (n04517823, vacuum)
```

130. **oven_stove** – 5 classes:
```
(n03761084, microwave)                     (n04111531, rotisserie)
(n04330267, stove)                         (n04442312, toaster)
(n04542943, waffle iron)
```

131. **space_heater_radiator** – 2 classes:
```
(n04040759, radiator)                      (n04265275, space heater)
```

132. **paper_tissue** – 2 classes:
```
(n03887697, paper towel)                   (n15075141, toilet tissue)
```

133. **cover_presentation** – 10 classes:
```
(n02786058, Band Aid)                      (n02840245, binder)
(n03291819, envelope)                      (n03598930, jigsaw puzzle)
(n03871628, packet)                        (n06359193, web site)
(n06596364, comic book)                    (n06785654, crossword puzzle)
(n07248320, book jacket)                   (n07565083, menu)
```

134. **toiletry** – 5 classes:
  (n03314780, face powder)                    (n03476991, hair spray)
  (n03690938, lotion)                         (n03916031, perfume)
  (n04357314, sunscreen)

135. **bed** – 2 classes:
  (n03388549, four-poster)                    (n04344873, studio couch)

136. **fabric** – 8 classes:
  (n02808304, bath towel)                     (n02834397, bib)
  (n03207743, dishrag)                        (n03223299, doormat)
  (n03485794, handkerchief)                   (n03998194, prayer rug)
  (n04209239, shower curtain)                 (n04525038, velvet)

137. **religious_castle_palace** – 12 classes:
  (n02699494, altar)                          (n02825657, bell cote)
  (n02980441, castle)                         (n03028079, church)
  (n03220513, dome)                           (n03781244, monastery)
  (n03788195, mosque)                         (n03877845, palace)
  (n03956157, planetarium)                    (n04346328, stupa)
  (n04486054, triumphal arch)                 (n04523525, vault)

138. **pole** – 3 classes:
  (n03355925, flagpole)                       (n03733131, maypole)
  (n03976657, pole)

139. **column** – 4 classes:
  (n03837869, obelisk)                        (n03903868, pedestal)
  (n03976657, pole)                           (n04458633, totem pole)

140. **stick** – 26 classes:
  (n02666196, abacus)                         (n02783161, ballpoint)
  (n02906734, broom)                          (n03141823, crutch)
  (n03250847, drumstick)                      (n03355925, flagpole)
  (n03388183, fountain pen)                   (n03658185, letter opener)
  (n03729826, matchstick)                     (n03759954, microphone)
  (n03804744, nail)                           (n03873416, paddle)
  (n03876231, paintbrush)                     (n03944341, pinwheel)
  (n03970156, plunger)                        (n04033901, quill)
  (n04039381, racket)                         (n04067472, reel)
  (n04141327, scabbard)                       (n04153751, screw)
  (n04208210, shovel)                         (n04228054, ski)
  (n04277352, spindle)                        (n04367480, swab)
  (n04485082, tripod)                         (n04501370, turnstile)

141. **big_standing_machine** – 4 classes:
  (n02977058, cash machine)                   (n03425413, gas pump)
  (n04243546, slot)                           (n04525305, vending machine)

142. **farm_machine** – 6 classes:
  (n03496892, harvester)                      (n03874293, paddlewheel)
  (n03967562, plow)                           (n04252225, snowplow)
  (n04428191, thresher)                       (n04465501, tractor)

143. **ax_meat_knife** – 2 classes:
  (n03041632, cleaver)                        (n03498962, hatchet)

144. **hand_tool** – 8 classes:
  (n02951585, can opener)                     (n02966687, carpenter's kit)
  (n03000684, chain saw)                      (n03109150, corkscrew)
  (n03481172, hammer)                         (n03954731, plane)
  (n03995372, power drill)                    (n04154565, screwdriver)

145. **sunglass** – 2 classes:
  (n04355933, sunglass)                       (n04356056, sunglasses)

146. **structure_near_water** – 4 classes:
(n02814860, beacon)                    (n02859443, boathouse)
(n02894605, breakwater)                (n03216828, dock)

147. **water_tower_beacon** – 2 classes:
(n02814860, beacon)                    (n04562935, water tower)

148. **camera_binoculars** – 3 classes:
(n02841315, binoculars)                (n03976467, Polaroid camera)
(n04069434, reflex camera)

149. **safe** – 2 classes:
(n03075370, combination lock)          (n04125021, safe)

150. **lock** – 2 classes:
(n03075370, combination lock)          (n03874599, padlock)

151. **box** – 9 classes:
(n02727426, apiary)                    (n02843684, birdhouse)
(n02971356, carton)                    (n03014705, chest)
(n03127925, crate)                     (n03482405, hamper)
(n03710193, mailbox)                   (n04204238, shopping basket)
(n04204347, shopping cart)

152. **ashcan_mailbox** – 2 classes:
(n02747177, ashcan)                    (n03710193, mailbox)

153. **bag** – 4 classes:
(n02769748, backpack)                  (n03709823, mailbag)
(n04026417, purse)                     (n04548362, wallet)

154. **purse** – 3 classes:
(n03908618, pencil box)                (n04026417, purse)
(n04548362, wallet)

155. **shoji_sliding_door** – 2 classes:
(n04201297, shoji)                     (n04239074, sliding door)

156. **construction_site** – 2 classes:
(n03126707, crane)                     (n03240683, drilling platform)

157. **patterned_structure** – 16 classes:
(n02788148, bannister)                 (n02999410, chain)
(n03000134, chainlink fence)           (n03065424, coil)
(n03347037, fire screen)               (n03459775, grille)
(n03530642, honeycomb)                 (n03930313, picket fence)
(n04005630, prison)                    (n04040759, radiator)
(n04265275, space heater)              (n04275548, spider web)
(n04326547, stone wall)                (n04589890, window screen)
(n04590129, window shade)              (n04604644, worm fence)

158. **roof** – 2 classes:
(n04417672, thatch)                    (n04435653, tile roof)

159. **small_object** – 20 classes:
(n02865351, bolo tie)                  (n02910353, buckle)
(n03314780, face powder)               (n03476684, hair slide)
(n03532672, hook)                      (n03666591, lighter)
(n03676483, lipstick)                  (n03692522, loupe)
(n03794056, mousetrap)                 (n03814906, necklace)
(n03840681, ocarina)                   (n03843555, oil filter)
(n03908714, pencil sharpener)          (n03929660, pick)
(n04116512, rubber eraser)             (n04127249, safety pin)
(n04131690, saltshaker)                (n04372370, switch)
(n04423845, thimble)                   (n04579432, whistle)

160. **housing_farming_structure** – 7 classes:

(n02793495, barn)                    (n02859443, boathouse)
(n03457902, greenhouse)              (n03697007, lumbermill)
(n03776460, mobile home)             (n03899768, patio)
(n04613696, yurt)

161. **solar_dish_telescope** – 2 classes:

(n04044716, radio telescope)         (n04258138, solar dish)

162. **Without a superclass** – 74 classes:

(n01622779, great grey owl)          (n01704323, triceratops)
(n01910747, jellyfish)               (n02672831, accordion)
(n02879718, bow)                     (n02892201, brass)
(n02950826, cannon)                  (n02965783, car mirror)
(n02966193, carousel)                (n02974003, car wheel)
(n03042490, cliff dwelling)          (n03188531, diaper)
(n03208938, disk brake)              (n03325584, feather boa)
(n03388043, fountain)                (n03467068, guillotine)
(n03494278, harmonica)               (n03495258, harp)
(n03527444, holster)                 (n03544143, hourglass)
(n03594734, jean)                    (n03595614, jersey)
(n03623198, knee pad)                (n03627232, knot)
(n03720891, maraca)                  (n03724870, mask)
(n03733281, maze)                    (n03743016, megalith)
(n03770439, miniskirt)               (n03787032, mortarboard)
(n03788365, mosquito net)            (n03792972, mountain tent)
(n03793489, mouse)                   (n03803284, muzzle)
(n03814639, neck brace)              (n03825788, nipple)
(n03884397, panpipe)                 (n03891251, park bench)
(n03920288, Petri dish)              (n03935335, piggy bank)
(n03937543, pill bottle)             (n03938244, pillow)
(n03958227, plastic bag)             (n03961711, plate rack)
(n03982430, pool table)              (n04023962, punching bag)
(n04033995, quilt)                   (n04136333, sarong)
(n04162706, seat belt)               (n04235860, sleeping bag)
(n04296562, stage)                   (n04317175, stethoscope)
(n04336792, stretcher)               (n04371430, swimming trunks)
(n04371774, swing)                   (n04399382, teddy)
(n04418357, theater curtain)         (n04447861, toilet seat)
(n04476259, tray)                    (n04507155, umbrella)
(n04584207, wig)                     (n04592741, wing)
(n06794110, street sign)             (n06874185, traffic light)
(n07615774, ice lolly)               (n07753592, banana)
(n07802026, hay)                     (n09229709, bubble)
(n09835506, ballplayer)              (n10148035, groom)
(n10565667, scuba diver)             (n11879895, rapeseed)
(n11939491, daisy)                   (n12057211, yellow lady's slipper)

