# OpenReview forum: "Automated Classification of Model Errors on ImageNet"
_NeurIPS.cc/2023/Conference — NeurIPS 2023 poster_

### Official Review · Reviewer_hvnS · 2023-06-30

**Soundness:** 3 good
**Presentation:** 2 fair
**Contribution:** 1 poor
**Rating:** 6
**Confidence:** 4

**Summary:**

This paper proposes a pipeline to automatically categorize ImageNet classification misclassifications. Previous work [1] defined and categorized those misclassifications into 4 types manually and this is the main motivation of automating the process. The results show that their pipeline has an accuracy of 60% based on human annotations of model misclassifications.


[1] analyzing the remaining mistakes on imagenet

I have read the author’s rebuttal and adjust the rating accordingly!

**Strengths:**

The paper is easy to follow as they mostly based on the previous work [1].

**Weaknesses:**

- The novelty of this paper is minimal and the motivation is not strongly supported. For example, the definitions and categorization of misclassifications have been done before in [1]. In writing, I do not see the reason why we need such categorization of misclassifications (i.e. the motivation). In Related work where I expect to see the motivation, the authors did not contrast their work with the literature then it is hard for me to see the contribution. I believe Related work can be greatly improved. Automating [1] is not considered novel at NeurIPS standard in my opinion.
- The authors only tested on 1 model and 1 dataset. It does not make the findings generalize. Another problem is the number of samples being examined. Based on Table 1, only 378 were tested!!!
- It looks like the authors are trying to do 4-way classification. In this case, it makes more sense to build a model to do that rather than having different methods for different misclassifications. The four types of misclassifications can be easily collected or synthesized.
- The authors only compared with [1]. The reviewer encourage them to extend the comparisons to others (e.g. [2]) and with other datasets and classifiers.

[2] ImageNet-Hard: The Hardest Images Remaining from a Study of the Power of Zoom and Spatial Biases in Image Classification

**Questions:**

N/A

---

> ### Author Rebuttal · Authors · 2023-08-10
>
> We thank Reviewer $\Rh$ for their review and answer their questions below.
>
> **Q: Can you discuss the motivation for your method? Where is the novelty in automating the categorization proposed by [1]?**
> Please see our general response for a more detailed motivation. Further, we believe the large number of works investigating ImageNet errors to be a testament to the exceptional interest in this area. While prior manual analyses such as Vasudevan et al.’s [1] can achieve very high precision, they have several disadvantages:  a) it is time-consuming even for precise models and intractable for imprecise ones, b) it requires a panel of expert reviewers, typically not available, and c) introduces inconsistencies due to human expert error. We alleviate all of these with our novel and effective automated categorization pipeline (see line 42 ff).
>
> **Q: Do you only evaluate a single model on 378 samples?**
> *No*, in fact, we evaluate *121* different models (trained classifiers), and always consider all images from the ImageNet validation set with multi-label annotations [1,2] (over 18.000). Perhaps the reviewer is referring to ViT-3B’s 378 multi-label errors, manually classified by Vasudevan et al. [1] and thus used by us to validate our automated approach in Table 1.
>
> **Q: Why don’t you simply train a classification model to classify the different error types?**
> Firstly, prior to our automated error categorization pipeline, only 548 labeled misclassifications from Vasudevan et al. [1] were available. Given that, the error classification depends not only on the image itself but also on the original label and model prediction (1.000.000 combinations for ImageNet), that would be an extremely small data set. Secondly, our pipeline, by its very nature, only considers images where the analyzed model failed, leaving a particularly hard subset of (training) images for any neural classifier. Finally, even if we could train such a classifier, it would suffer from the same lack of interpretability as the original model, making it ill-suited for an evaluation pipeline aiming to understand model errors.
>
> **Q: Could you compare with the ImageNet-Hard dataset and evaluate on other datasets and classifiers?**
> See our general response with regards to ImageNet-Hard and other datasets in general. As for other classifiers see the above response.
>
>
> These questions lead us to believe that reviewer $\Rh$ misunderstood the setting and high-level goal of our paper. While we will aim to improve the presentation to avoid such misunderstandings, we respectfully ask the reviewer to reconsider their review and acceptance score in light of these clarifications.
>
> **References**
> [1] Vasudevan et al. "When does dough become a bagel? Analyzing the remaining mistakes on ImageNet", NeurIPS’22
> [2] Shankar et al. "Evaluating machine accuracy on imagenet." ICML’20.

---

> > ### Comment · Reviewer_hvnS · 2023-08-13
> > **I'm eager to continue our discussion**
> >
> > I would like to thank the authors for the efforts in the rebuttal. After reading your rebuttal and reviews from other reviewers, my replies are:
> >
> > Regarding Q2, yes, I'm referring to the comparison with Vasudevan et al. [1].
> > Given that your objective is to classify models' errors, I find your evaluation valid only for the one specific model and the 378 samples (in which you drew comparisons with Vasudevan et al.).
> > In the case of the 121 models, your observation that MLF and MLA have an inverse correlation is to be expected, as high-accuracy models naturally exhibit a lower rate of errors.
> >
> >
> > Regarding Q3, I believe that human annotations aren't necessary for such failure samples. For instance, when using ResNet-50 on ImageNet, which achieves a 76.13% top1 accuracy, you'll have approximately 24% of the 50K validation samples available to train the 4-way classifiers. However, I understand this is more of an open-ended topic for discussion, and I don’t anticipate a solution within your rebuttal.
> >
> > For Q4, I referenced that particular work because they also categorize classifiers' misclassifications. I hope I clarified my point and insist that I did not misunderstand the paper’s settings.
> >
> > I am open to discussion and very willing to raise the score if the authors could clarify my concerns.

---

> > > ### Author Response · Authors · 2023-08-14
> > > **Authors response**
> > >
> > > Thank you for engaging in this discussion!
> > >
> > > Regarding Q2: Our goal in comparing with the results of Vasudevan et al. in Table 1 is not to "evaluate" our method as such given that categorising model errors is not a standardised or established task. As even expert human reviewers might disagree on the correct category for some errors, our goal is rather to validate that our automatic categorisation broadly agrees with the opinion of this set of human experts. We believe this to be sufficient given that (most of) our categorizations are based on carefully designed, human interpretable rules that could be proposed fully independently of Vasudevan et al.'s annotations. However, we are happy to extend our evaluation to the only other model (GreedySoups [1], also reviewed in Vasudevan et al.) for which such human annotations exist. Beyond this, a further evaluation would require additional human annotations to be collected on a large scale. Unfortunately this requires a panel of human experts with good understanding of the ImageNet label set, making it very hard to scale. We are thus open to the proposition of better evaluation approaches but believe our approach to be valid.
> > > Further, while higher accuracy models by definition have a lower error rate, we believe it to be surprising that the portion of multi-label errors that we classify as severe model failures (i.e., can not explain by any of the other error types) decreases with increasing model accuracy, especially since some prior work suggest recent progress on ImageNet to be mostly due to overfitting on labeling biases [2].
> > >
> > > Regarding Q3: We want to reiterate that we do not want to predict whether or not a given model will classify a sample correctly, rather given a misclassified sample with its multi-class labels and the model prediction, we wish to categorise this error. We are thus unsure how collecting 12k model errors of a ResNet-50 without a corresponding categorization would help to train such a 4-way classifier without manually labelling all these predictions. While unsupervised clustering might be a possibility, we believe it to be highly unlikely to recover clusters corresponding to our (or any other interpretable) error type definitions.
> > >
> > > Regarding Q4: We are happy to discuss ImageNetHard in more detail, however while they conduct an analysis of error types this merely consists of providing the top-1 prediction and the ground truth label to GPT3.5 and asking it to classify this as a "common" or "rare" error. From the authors description they roughly translate "common" errors to fine-grained classification errors. As this only depends on the predicted class and ground truth label, we could compare GPT3.5's classification of class tuples with that of our human superclass construction, but believe this to be of limited value as we would generally consider the human annotation to be the ground truth in these cases. Further, as this categorization does not seem to be publicly available, it would be fairly costly to collect GPT3.5's annotations for all occurring class tuples.
> > >
> > >
> > > [1] Wortsman et al., "Model soups: averaging weights of multiple fine-tuned models improves accuracy without increasing inference time" ICML'22\
> > > [2] Beyer et al., "Are we done with ImageNet?", arXiv'20

---

> > > > ### Comment · Reviewer_hvnS · 2023-08-17
> > > > **Concerns about the evaluation**
> > > >
> > > > Thanks for the replies!
> > > >
> > > > > Our goal in comparing with the results of Vasudevan et al. in Table 1 is not to "evaluate" our method as such given that categorising model errors is not a standardised or established task.
> > > >
> > > > I think this evaluation is to verify that your automated algorithm is working well (i.e., aligns with human annotations), right?
> > > >
> > > > You said:
> > > > > Our goal is rather to validate that our automatic categorisation broadly agrees with the opinion of this set of human experts.
> > > >
> > > > Then if this is your goal, why do you downweight the evaluation with Vasudevan as I think this comparison is, again, the most important and valid. In the paper, you also state that:
> > > > > To anchor our automated error analysis and verify that it is practical, we compare it to the work of 375 Vasudevan et al. (2022)
> > > >
> > > > Looking at Table 1, I also concern about the accuracy of your method (60\% matching with human experts) because most of them come from fine-grained mistakes that are much easier, in my opinion, to detect compared to the remaining three types of errors. Then, this result from the table does not generally prove its capability on ImageNet errors.

---

> > > > > ### Author Response · Authors · 2023-08-18
> > > > > **Addressing the concerns**
> > > > >
> > > > > Thank you for the reply!
> > > > >
> > > > > Yes, our goal with the results in Table 1 is to see how our pipeline aligns with manual error classification. In this context, we wish to further elaborate on three points.
> > > > >
> > > > > First, we would like to re-emphasize the ambiguous nature of the errors (as acknowledged by Vasudevan et al., “Our qualitative judgments [of what is a mistake] are therefore based on a biased worldview comprising the five panelists”).  Therefore, some drift is to be expected even between the annotations of two groups of human experts. Moreover, our pipeline encodes an implicit order for the error categorization in terms of severity, i.e., when possible, we select the least severe error type that would explain the model mistake. Human annotators have not done that. These points explain some of the differences we see compared to Vasudevan et al. and also highlight the strengths of our approach: namely, that it is consistent and repeatable.
> > > > >
> > > > > Second, we designed our error classification pipeline to be conservative: in Table 1 we abstain from classifying the error for 18% (69 out of 378) of the samples thus treating them as model failures. From the remaining 309, we are in exact agreement with Vasudevan et al. on 225 samples (73%). After further manual inspection (L382-383), we determined that of the 84 remaining samples where the error types differ, only for 32 images our error categorization does not explain the mistake, too, and the annotation from Vasudevan et al. is clearly preferable. To summarize, for 90% of the samples (277 out of 309) for which we provide an error classification, it plausibly explains the underlying mistake.
> > > > >
> > > > > Finally, we also performed our analysis for the other model evaluated by Vasudevan et al. (Greedy Soups) and show the results in the table below. Our automatic error categorizations exactly match for 164 out of 219 (75%) samples for which the pipeline provides an error type. On the remaining 55 images our error categorization is acceptable for 33, resulting again in 90% (197 out of 219) of samples for which the provided error classification explains the model mistake. Overall, these results indicate a very similar trend, thus further validating our modeling choices.
> > > > >
> > > > > Therefore, we believe that our automatic categorisation is indeed aligned with the opinion of human experts. We further note that an efficient and automatic classification pipeline is helpful even if it is not perfect, as it enables the study of trends at scale.
> > > > >
> > > > > | Error categories | FG | FG OOV | Non-prototypical | Spurious Correlation | Model failures | Total (row) |
> > > > > | :----------- | -----------: | -----------: | -----------: | -----------: | -----------: | -----------: |
> > > > > | Fine-grained | 140 | 12 | 1 | 7 | 12 | 172 |
> > > > > | Fine-grained OOV | 8 | 8 | 0 | 5 | 5 | 26 |
> > > > > | Non-prototypical | 7 | 1 | 9 | 2 | 0 | 19 |
> > > > > | Spurious Correlation | 7 | 5 | 0 | 7 | 13 | 32 |
> > > > > | Total (col) | 162 | 26 | 10 | 21 | 30 | 249 |

---

> > > > > > ### Comment · Reviewer_hvnS · 2023-08-21
> > > > > > **Final rating**
> > > > > >
> > > > > > Dear authors,
> > > > > >
> > > > > > I genuinely appreciate your efforts in conducting the second experiment with Greedy Soups.
> > > > > > I acknowledge the reply, "... highlight the strengths of our approach: namely, that it is consistent and repeatable" and it addresses the ambiguity in novelty.
> > > > > >
> > > > > > However, my primary concern remains the paper's practicability, particularly as it appears to excel mainly in addressing FG failures.
> > > > > >
> > > > > > In light of your recent efforts, I have raised my score to a 6. Best wishes for the AC's decision.

---

### Official Review · Reviewer_kbqQ · 2023-07-02

**Soundness:** 4 excellent
**Presentation:** 4 excellent
**Contribution:** 4 excellent
**Rating:** 7
**Confidence:** 4

**Summary:**

The paper introduces an automated analysis on the model errors in ImageNet.
The analysis sequentially categorizes into six types of errors with increasing
criticality, unveiling the true model failures. The results are contextualized
with previous work.

**Strengths:**

- This work is of high interest to the computer vision and machine learning
  community, given the extensive use of ImageNet as a benchmark.

- The quality and presentation of the analysis is excellent.

- The full code to conduct the pipeline presented in the paper is provided with
  the paper, which is extremely value considering the reproducability under the
  complexity of the analysis (wrt. the number of models, etc.). Although the
  analysis may seem very large-scale, it is surprisingly affordable and, with
  the code, is very easily reproducable.

- The analysis builds upon and refines the previous NeurIPS's work by Vasudevan
  et al..

- The sequential categorization, starting with the least critical category of
  error, is a very good idea.

**Weaknesses:**

- I could not find any major weakness of the paper that would
  impede its acceptance in the current state.

**Questions:**

- While not critical, I am wondering why the paper did mention to use the
  WordNet parent classes for the super classes. A single sentence on the
  fitness of the parent classes as super classes would we interesting.

**Limitations:**

The paper addresses its major limitations well.

---

> ### Author Rebuttal · Authors · 2023-08-10
>
> Thank you very much for the kind review. We are encouraged that you see our work as a valuable contribution to the community. Regarding your question about using the WordNet hierarchy to form superclasses, we refer you to the general response and are happy to answer any further questions.

---

> > ### Comment · Reviewer_kbqQ · 2023-08-21
> > **Thank you for the rebuttal**
> >
> > I would like to thank the authors for their detailed responses.
> >
> > While I still think this is an important and strong work, given the points raised by other reviewers, specifically **Reviewer GgcH** concerning the *manually obtained labels from previous work*, and the resulting issue of scalability to other models, I fear that my initial score of 8 may have been too high.
> >
> > I will update my score to 7.

---

> > > ### Author Response · Authors · 2023-08-21
> > > **Response by authors**
> > >
> > > Thank you for getting back to us!
> > >
> > > Assuming you are referring to the scalability to other datasets: we would like to point out that, as discussed in Q3 in the main response and acknowledged by reviewer GgcH in their most recent reply, a Re-Label based approach can be used to scale to new datasets. How well this aligns with the gold-standard of human expert labels, however, remains an exciting item for future work.
> > >
> > > Thus, we believe scalability is not severely limited.

---

### Official Review · Reviewer_HYyY · 2023-07-05

**Soundness:** 3 good
**Presentation:** 3 good
**Contribution:** 2 fair
**Rating:** 5
**Confidence:** 4

**Summary:**

The paper introduces an automated pipeline to classify ImageNet's errors from 4 categories: (i) fine-grained categories; (ii) fine-grained OOV; (iii) Non-prototypical instances; and (iv) Spurious correlations. The main contribution of the paper is this automated pipeline with which various errors can be categorized.  While there exists a plethora of recent works in classifying ImageNet errors, the paper does a good job of summarizing all these errors and also designing an automated pipeline to detect them for ImageNet.


**Strengths:**

- Good overview of ImageNet errors and a simple, but effective automated pipeline for error detection.
- A wide-range of models are tested which is a strong point for the paper.

**Weaknesses:**

- The analysis is restricted to only analysing ImageNet errors. While it is a good direction to inform vision research -- it will be beneficial to extend the framework to understand error types in other ImageNet variants such as ImageNet-A. What are the error distributions in such sets? The current analysis though good, is restrictive.
- The paper does not discuss recent works such as ImageNet-X (https://facebookresearch.github.io/imagenetx/site/home)  which also looks at understanding model failures. While the categories considered in this paper is a subset of ImageNet-X error types, can the authors compute an overlap metric between some of the error types detected in this paper and ImageNet-X?
- While a range of models are tested, can the authors distinguish between the pre-training strategies more in the plots? E.g., highlight SSL methods, Vision-Language pre-training methods and Supervised Pre-training methods using different colors / notations. This would make the paper stronger.
- Can the authors give some intuitions on the unexplained failures? What are the potential causes?
How can the error analysis learnings be useful in improving model training? Can the authors provide some intuition on this?

**Questions:**

See Weaknesses

**Limitations:**

- While the paper provides a good overview of ImageNet errors and provides an automated pipeline to explain a subset of those errors, the analysis is still restricted to ImageNet and is not extended to other ImageNet test sets where the drop in accuracy is more severe.
- Although the paper tests a large number of models, each component (e.g., pre-training strategy, augmentations) is not well dissected deeply.

---

> ### Author Rebuttal · Authors · 2023-08-10
>
> We thank Reviewer $\RH$ for their insightful feedback, helpful suggestions, and interesting questions which we address below.
>
> **Q: Can the proposed method be extended to ImageNet variants, such as ImageNet-A. What are the error distributions in such sets?**
> In principle, our pipeline can be extended to ImageNet-A. However, this would require multi-object annotations. To the best of our knowledge, no such annotations exist. While collecting such labels in general (see our response to $\RG$ for details) could be automated via the Re-Label approach, we believe on ImageNet-A this would be not fruitful as the images were specifically selected to be challenging (wrong or low-confidence).
>
> **Q: Are the error types considered in this paper a subset of ImageNet-X error types? If so, how do they map to each other?**
> The types of errors considered in ImageNet-X are mostly entirely orthogonal to the types of errors considered by us. Their works focus on confounders that make a particular image challenging such as an atypical pose, background, and lighting, while we focus more on systematic errors caused (partially) by labeling (set) choices and aggregate effects such as atypical pose, background, etc. into the non-prototypical instance category. Similarly, their broad subclass confounder, could, depending on context, be considered class overlap, OOV, or fine-grained misclassification errors in our categorization.
>
> **Q: Can you distinguish different pre-training strategies in your plots and analysis?**
> Great suggestion. Since the original submission, we have collected over 800 additional models and more detailed annotations including the exact pre-training method used. We are happy to update our plots correspondingly and include a more detailed analysis in an updated version of our paper.
> We include an updated plot corresponding to Figure 16 in the PDF attached to the main response. Generally, we observe that trends remain dominated by MLA and top-1 accuracy with the best-performing models all pretrained using self-supervised learning.
>
> **Q: Can you provide some intuitions on the unexplained failures?**
> Unfortunately, we can not give general intuitions on the unexplained failures, as their name implies. These are model failures that we believe no human (aware of the ImageNet label set) would make. Further, while better (more top-1 or MLA-accurate) models make fewer such errors, they do not follow a strict set inclusion, making an inherent hardness to these images that is not apparent to humans unlikely.
>
> **Q: How can this analysis be used to improve model training?**
> Please see our answer in the main response.

---

> > ### Comment · Reviewer_HYyY · 2023-08-16
> > **Reply to Authors**
> >
> > I thank the authors for their response; I will maintain my rating!

---

### Official Review · Reviewer_rfV8 · 2023-07-05

**Soundness:** 3 good
**Presentation:** 3 good
**Contribution:** 2 fair
**Rating:** 5
**Confidence:** 3

**Summary:**

This paper aims to create an automated pipeline for defining the types of errors made by ML models on ImageNet. Error categories are from previous work, and the paper compares the results between the previous human based pipeline and the newly automated version. The authors use the pipeline to evaluate over 100 models, and provide some analysis on the findings.

**Strengths:**

- Paper structure is clear
- Design decisions for automating each step of the pipeline make sense
- Results are validated and put in context with prior work

**Weaknesses:**

- Analysis of results seems limited. I would like to see additional analysis that builds upon the benefits of such pipelines. For example, if we can now easily get to severe model errors, can we do human analysis on these to create categories or better understand them?
- I would like to better understand the motivation behind this work, such as potential use cases for the pipeline to improve machine learning research

**Questions:**

See weaknesses

**Limitations:**

I believe that limitations are addressed adequately

---

> ### Author Rebuttal · Authors · 2023-08-10
>
> We thank reviewer $\Rr$ for their review and interesting questions, which we answer below.
>
> **Q: Can you provide an in-depth analysis of the results obtained using your automated pipeline, e.g., by manually categorizing model failures?**
> While we include some more high-level observations (e.g. the differences in error distribution trends for artifact vs. organism classes) and some more detailed observations (e.g., ViTs suffering from significantly fewer model failures than ConvNeXts on artifacts conditioned on the same multi-label accuracy), we believe a more detailed and especially manual analysis to be an interesting item for future work and out of scope here given the focus on the automation of the analysis.
>
> **Q: Can you provide a more in-depth motivation for this work, in particular how it can be used to improve machine learning?**
> The first important motivation for our work was to put recent methodological advances in computer vision into context. Indeed, we observe that the *portion* of errors that we categorize as severe model failures decreases with both top-1 and multi-label accuracy, suggesting that these metrics not only remain valuable measures for progress, in contrast to recent suggestions that they mostly improve by overfitting on labeling biases but even under-report this progress. Second, we believe that trends in error distributions such as the dominance of fine-grained misclassifications for organisms or the prevalence of fine-grained out-of-vocabulary errors for artifacts, can help inform methodological research.
> Finally, our automated fine-grained error evaluation could allow developers of novel training techniques to quickly assess the kind and distribution of errors their models still make, allowing interventions targeted at specific types to be evaluated more effectively.

---

> > ### Comment · Reviewer_rfV8 · 2023-08-19
> >
> > Thank you for your response. I choose to keep my score as the response does not fully address my concerns of limited analysis.

---

### Official Review · Reviewer_GgcH · 2023-07-12

**Soundness:** 2 fair
**Presentation:** 2 fair
**Contribution:** 2 fair
**Rating:** 4
**Confidence:** 3

**Summary:**

The paper proposes an automatic error classification pipeline for ImageNet models. In particular, they categorize mistakes into overlapping classes, multi-object images, fine-grained and fine-grained with out-of-vocabulary (OOV), non-prototypical, the examples influenced by spurious correlations and unexplained mistakes. To detect fine-grained OOV errors the authors used WordNet and an open world CLIP classifier. They evaluated 121 models and showed that better performing models (in terms top-1 accuracy or multi-label accuracy) have a lower _portion_ of unexplained errors, and higher portions of errors which are explained by class ambiguity and fine-grained cases. They analyzed error portion trends separately for objects and organisms ImageNet classes, and analyzed the effect of training dataset and model architecture.

**Strengths:**

While prior works studied questions similar to this paper such as [1] analyzed error types of one or a few high performing ImageNet classifiers such as ViT-3B, and [2] looked at how multi-label accuracy improves with model scale, this paper studies a more detailed question of how the error type distribution changes as we scale up the models in terms of the architecture size and pre-training dataset.
They also propose an automatic way to identify fine-rained out-of-vocabulary mistakes which relies on an external open world classifier model CLIP (assuming that we filtered out overlapping and fine-grained classes).

**References**

[1] Vasudevan, Vijay, et al. "When does dough become a bagel? analyzing the remaining mistakes on imagenet."

[2] Beyer, Lucas, et al. "Are we done with imagenet?."

**Weaknesses:**

1. **Limited novelty.**

Given the results from the prior works that study label mistakes and ambiguity in ImageNet dataset like [1] and [2], the novel insights from this paper seem limited.
I would suggest that the authors explicitly emphasize that the error categorization and definitions as in Figure 1 and most subsections in Section 3 (except for novel approach for fine-grained OOV) were proposed in prior works, and that they separate the background and results from prior works and their novel approaches and insights.

In particular, both [1] and [2] categorize mistakes in a similar fashion, and the categorization in [2] is the one this paper relies on. [1] focus on analyzing mistakes of a few large-scale models like ViT-3B and report the percentage of different error types and their severity in their Section 4.1 Table 2. Similarly to this paper they report mistakes separately for objects and organisms classes from ImageNet. They also propose ImageNet-M — a subset of only 68 examples where high-performing models still consistently make major mistakes.
[2] also discuss a similar error categorization but they focus on analyzing how scaling up models affects top-1 and multi-label accuracy.
This paper combines two types of analysis from [1] and [2] and studies how scaling up models affects the error types distribution, and concludes that for stronger performing models the percentage of major mistakes significantly decreases. While this experiment and analysis is new, the conclusions are somewhat straightforward given the results from [1] and [2]?

Could the authors please list the new observations and conclusions that were made in their analysis and that were not known in prior work?



2. **The extent to which the pipeline of error type classification is automatic seems to be overclaimed.**
The introduction and abstract of the work make an impression that the paper proposes a fully automatic way to classify model’s mistakes, without relying on manual annotation unlike prior works [1, 2].
However, after reading Section 3 it turns out that class overlap (Sec 3.1), multi-object cases (Sec 3.2), and spurious correlations (Sec 3.6) rely on manually obtained labels from [1] and [2], the definition of non-prototypical examples (Sec 3.5) relies on the ViT model’s mistakes which were analyzed in [1], and fine-grained errors (Sec 3.3) rely on manual grouping of classes into super-classes (see another comment re this case in Questions).
The only case of automatic error categorization is the fine-grained out-of-vocabulary error in Section 3.4 which involves WordNet tree and open world CLIP classifier.
 I suggest that the authors adjust their claims accordingly since such a pipeline can’t be called automatic.

In particular, if the proposed error classification pipeline was indeed automatic and didn’t have limitations discussed in lines 45-47 in terms of relying on human labellers, how would one apply the proposed pipeline to another large-scale dataset where we don’t have these labels from [1] and [2]? If it is possible, it would be helpful to see the application of such error classification to another dataset.


3. **The design choices for detection of fine-grained out-of-vocabulary mistakes is not fully clear.**
The only automatic and novel part of the error classification pipeline is detection of fine-grained OOV mistakes: (1) first we find 10 most visually similar images in ImageNet train set in terms of the CLIP embedding, (2) check that at least one label belongs to the same superclass as model’s prediction, and (3) measure CLIP similarity with the correct class and WordNet in-vocabulary and OOV sibling classes related to both correct class and model's prediction.

First, for the last step the description of the label proposal set is a bit confusing, and it would be helpful if the authors could clarify the exact set of proposals they are considering.
Second, it would be helpful to have more examples like in Figure 10 to illustrate step-by-step fine-grained OOV mistakes identification and show for each example what were the 10 closest images in embeddings space and which labels they had, and then which label proposal set CLIP scored (in particular, where those proposals were taken from and what were the CLIP scores exactly). It would be helpful to have more intuition for each step of this process.

4. [Minor] While the paper is generally easy to follow, I highlight a few points below which were not as clear:

-  For Figures like 3, 5 and other trends, please clearly define what exactly is shown on y axis. I am assuming it is (# of mistakes of type X ) / (# of all mistakes) for each model.
- On the same Figures, does each marker correspond to one model? Is it correct that each model is shown twice: once for objects and once for organisms?

- Please fix captions for Figures 7, 12 and 14.


Typos:

- Line 54: and -> an
- Line 99: find
- Line 113: need to
- Line 189: the

**References**

[1] Vasudevan, Vijay, et al. "When does dough become a bagel? analyzing the remaining mistakes on imagenet."

[2] Beyer, Lucas, et al. "Are we done with imagenet?."

**Questions:**

1. Line 210: “we manually reviewed all 1000 ImageNet classes and defined semantically similar superclasses guided by the question of whether an untrained human could reasonably confuse them”
Why did you choose to manually review and group classes instead of using WordNet tree?
This also doesn’t match well with the claims about the error classification pipeline being automatic.

2. What are the examples of spurious correlations that you identified in Section 3.6? How are they related to the findings from prior works studying spurious correlations in ImageNet like e.g. Salient ImageNet [3] and Hard ImageNet [4]?

3. Line 370: “Arguably, we observe vision transformers achieving lower model failure rates on artifacts at very high MLA when compared to ConvNeXts.”
Does this observation hold even when we condition on the size of pre-training dataset?

**References**


[3] Singla, Sahil, and Soheil Feizi. "Salient ImageNet: How to discover spurious features in Deep Learning?."

[4] Moayeri, Mazda, Sahil Singla, and Soheil Feizi. "Hard imagenet: Segmentations for objects with strong spurious cues."

**Limitations:**

The authors adequately addressed the limitations.

---

> ### Author Rebuttal · Authors · 2023-08-10
>
> We thank Reviewer $\RG$ for their detailed review, insightful feedback, helpful suggestions, and interesting questions. We will fix the mentioned typos and update the Figure captions to provide further clarity. Below we answer the reviewer’s remaining questions.
>
> **Q: Can you highlight the main novelties of this work given prior investigations?**
> Yes, please see our reply to Q1 in our General Response.
>
> **Q: Is the error classification pipeline truly automatic? How would it be adapted to other large-scale datasets?**
> Our pipeline is automated in the sense that it can be continuously run on a large number of models, where previous approaches required the manual review of each new incorrect model prediction, which is infeasible for all but the most accurate models. This automatization even for relatively inaccurate models is what enables us to study how error distributions evolve, over meaningful ranges of (top-1) error rates.
>
> For a discussion of how to adapt it to other datasets, please see Q3 in our General Response.
>
> **Q: Can you clarify the exact design choices for fine-grained OOV mistakes?**
> Intuitively, we consider an error to be a fine-grained OOV mistake if it satisfies two requirements: first, the object that the model aims to classify is not included in the ImageNet label set, i.e., it is indeed out-of-vocabulary, and second, for this object, the predicted class is (only) a fine-grained error. To this end, we first retrieve images that show visually similar objects and check if these belong to the same superclass as the predicted label, thus indicating that the prediction is only a fine-grained error. To confirm that the true label is indeed OOV, we collect a set of label proposals (discussed in detail below) and check whether an open-world classifier prefers a label not in the ImageNet label set. We collect this set of label proposals as the union of the following: All labels in the same superclass as the model’s prediction, all direct WordNet siblings of the model’s prediction, and all WordNet ancestors of the model’s prediction up to but excluding the first common ancestor with the ImageNet label. Intuitively, this provides the open-world model the choice between the most closely related in-vocabulary classes (same superclass) and out-of-vocabulary classes (direct siblings and ancestors).
>
> **Q: In Figures 3 & 5 what is shown on y axis? Is it (# of mistakes of type X ) / (# of all mistakes) for each model?**
> The proportions are computed individually for each group, i.e., for a given model Y and group Z (organisms or artifacts) it shows the (# of mistakes of type X for model Y with an ImageNet label belonging to group Z)/(# of mistakes for model Y with an ImageNet multi-label belonging to group Z). We will clarify this.
>
> **Q: In Figures 3 & 5, does each marker correspond to one model? Is each model shown twice, once for objects and once for organisms?**
> Yes, each model is shown with one marker per group. We will clarify this.
>
> **Q: What are the examples of spurious correlations that you identified in Section 3.6? How do they relate to SalientImaget [2] and HardImageNet [3]?**
> As we outline in L312-317, we base our labels on Beyer et al. [1] and we obtain 965 pairs of classes that may be spurious correlations. Out of these 965 class pairs we observed 749 as actual spurious correlation errors and have included example images in the PDF attached to the main response.
> We thank the reviewer for highlighting the works on SalientImaget [2] and HardImageNet [3] and are happy to discuss them in our related work section. For each class in ImageNet, Singla and Feizi [2] collect 5 salient or core (visual) features. We checked all 965 of our co-occurrence pairs, considered candidates for spurious correlation, and found 301 contain two classes that share at least one salient visual feature. This substantial overlap has an almost 0 chance of occurring at random (less than $10^{-200}$), showcasing the alignment between the two methods. Further, we will extend our discussion on the detection of spurious correlations by considering overlapping core features as a criterion. HardImageNet [3] extends the analysis of Singla and Feizi [2] by considering the 15 class pairs with the strongest spurious features and turning them into a segmentation task.
>
> **Q: Do vision transformers outperform ConvNeXts even when conditioned on the size of the pre-training datasets?**
> Conditioning on the same multi-label accuracy and same pre-training dataset, we observe that ViTs exhibit significantly fewer severe model failures on artifacts than ConvNeXts. Interestingly this is not the case for organisms. There, ConvNeXts perform either similarly or better than ViTs, sometimes even outperforming ViTs pretrained on a larger corpus (at slightly lower MLAs). We speculate that this might be due to the ConvNeXts convolutional backbone leveraging textures more heavily [4], which could be particularly helpful for organisms, while ViTs are better at dealing with the clutter often present in artifact scenes. We believe this to be an interesting example of the type of trends that can only be discovered using a large-scale, automated analysis such as ours.
>
> **References**
> [1] Beyer et al. "Are we done with imagenet?." arXiv
> [2] Singla and Feizi. "Salient ImageNet: How to discover spurious features in Deep Learning?."
> [3] Moayeri, Mazda, Sahil Singla, and Soheil Feizi. "Hard imagenet: Segmentations for objects with strong spurious cues"
> [4] ImageNet-trained CNNs are biased towards texture; increasing shape bias improves accuracy and robustness. Geirhos et al., ICLR 2019

---

> > ### Comment · Reviewer_GgcH · 2023-08-19
> > **Thank you for your response!**
> >
> > I thank the authors for their response and clarifications! I read the rebuttal, as well as the other reviews and discussions.
> >
> > I strongly encourage the authors to make it more clear in the next revision of the paper that their error categorization relies on *manually obtained* multi-label annotations from [1, 2, 3]. The motivation of this work is to make error categorization scalable unlike the manual error review which relies on human experts (e.g. Introduction in lines 42-47 discusses the limitations of manual labelling), at the same time the core assumption of this pipeline is that we have multi-label annotations which require the same extent of manual review.
> > I think applying the proposed pipeline to other datasets beyond ImageNet, potentially using ReLabel approach as mentioned in the rebuttal, could be interesting and more novel.
> >
> > Given that insights made in this paper overlap with those of prior works [1-3], my vote is still leaning towards rejecting the paper, but to recognize clarifications from the authors' response and partially addressed concerns, I raise my score to 4.
> >
> >
> > [1]  Shanka, et al. "Evaluating machine accuracy on ImageNet."
> >
> > [2] Vasudevan, Vijay, et al. "When does dough become a bagel? analyzing the remaining mistakes on ImageNet."
> >
> > [3] Beyer, Lucas, et al. "Are we done with ImageNet?"

---

### Author Rebuttal · Authors · 2023-08-10

$\newcommand{RG}{\textcolor{green}{GgcH}}$
$\newcommand{Rr}{\textcolor{blue}{rfV8}}$
$\newcommand{RH}{\textcolor{purple}{HYyY}}$
$\newcommand{Rk}{\textcolor{orange}{kbqQ}}$
$\newcommand{Rh}{\textcolor{teal}{hvnS}}$

We thank all reviewers for their work and are delighted that they appreciate the structure and quality of the presentation ($\Rr$, $\Rh$, $\Rk$), connection to prior work ($\Rr$,$\RH$,$\Rk$) and design choices in building the automatic pipeline ($\Rr$,$\RH$,$\Rk$). Here we address questions raised by multiple reviewers and reply to individual concerns below.

**Q1: Can you highlight the main novelties of this work? ($\RG$, $\Rh$)**
The novelty and contribution of our work lies mostly in the automatization of the analysis done by Vasudevan et al. and the large-scale analysis enabled by this automatization. More concretely, our automated analysis of over 100 models (extended to over 900 by the time of this rebuttal) allows us to make much more general and well-supported observations than the manual or much coarser investigations performed before:
* While prior works have investigated the relation of multi-label (MLA) and top-1 accuracies, to the best of our knowledge, we are the first to go beyond this and perform a large-scale study of *the distributions of the different error types*.
* Organism and artifact classes exhibit significantly different trends in terms of error distributions. In particular, the main source of errors for organism classes are fine-grained classification mistakes, while artifacts suffer much more from spurious correlations and out-of-vocabulary errors. While this observation agrees well with our intuition, we believe we are the first to confirm it empirically.
* The *portion* of errors that we categorize as severe model failures decreases significantly with increasing top-1 and multi-label accuracy, suggesting that even top-1 accuracy remains a meaningful measurement of model improvement and recent progress might in fact have been under- instead of over-reported.

**Q2: Why did you manually review and group classes into superclasses (L210) instead of using WordNet tree? ($\RG$, $\Rk$)**
Our superclasses were designed to group classes that humans perceive to be very similar to each other such that, e.g., an untrained human could easily confuse them. The ImageNet labels are nodes from WordNet which is itself organized in a hierarchy of synonym sets (synsets), each expressing a distinct concept. The WordNet hierarchy implies a natural subset relationship between the synsets which indeed forms the basis for our superclass organization. However, this hierarchy is suboptimal for our purposes.
For example, the direct WordNet parent of ``cock`` and ``hen`` is ``bird``, which has 59 ImageNet labels as successors, including small birds, birds of prey, parrots, aquatic birds, etc., which are substantially different from ``cock`` and ``hen``. In contrast, we manually grouped ``cock`` and ``hen`` together with ``gallinaceous birds``, resulting in a more refined superset with just 9 classes. Further, some visually similar ImageNet classes have large very broad super classes, such as ``physical entity``.

**Q3: How would the error analysis pipeline be adapted to other large-scale datasets (where multi-label or re-annoations are not present)? ($\RG$, $\Rh$)**
We believe extending this work to new datasets as outlines below, to be an interesting item for future work. While less precise than using a human annotated gold-standard, multi-labels could be generated using a Re-Label [1] approach, allowing (given sufficient scale) spurious correlation pairs to be extracted from co-occurrence frequencies. After defining superclasses manually as required by their very definition, this would allow all error types except for the rare non-prototypical instances to be categorized.

**References**
[1] Re-Labeling ImageNet: From Single to Multi-Labels, From Global to Localized Labels. Yun et al., CVPR 2021

---

### Author Response · Authors · 2023-08-21
**Wrap-Up and Thank you for the Discussion!**

We thank all reviewers for the lively discussion and are encouraged by the score increases.
Further, we want to briefly highlight the new results on the GreedySoups model (introduced in the discussion with $\Rh$), which allow us to further validate our automatic pipeline against human expert annotations.

Thank you again for the discussion! We will take all comments, clarifications, and suggestions into account for the next revision.

---

### Decision · Program_Chairs · 2023-09-21

**Decision:**

Accept (poster)

**Comment:**

The paper proposes an automatic error classification pipeline for ImageNet models, they break down the various error types into overlapping classes, multi-object images, fine-grained and fine-grained with out-of-vocabulary (OOV), non-prototypical, the examples influenced by spurious correlations and unexplained mistakes. While the reviewers do cite that there are other works that do things in this direction and the impact could be limited since this is one dataset, the reviewers generally agree the contribution is valuable and the experimental work is thorough. Thus the decision will be to accept.